# Online Conformal Prediction via Universal Portfolio Algorithms

**Tuo Liu** [1]  **Edgar Dobriban** [2]  **Francesco Orabona** [1]

## Abstract

Online conformal prediction (OCP) seeks prediction intervals that achieve long-run $1-\alpha$ coverage for arbitrary (possibly adversarial) data streams, while remaining as informative as possible. Existing OCP methods often require manual learning-rate tuning to work well, and may also require algorithm-specific analyses. Here, we develop a general regret-to-coverage theory for interval-valued OCP based on the $(1-\alpha)$-pinball loss. Our first contribution is to identify *linearized regret* as a key notion, showing that controlling it implies coverage bounds for any online algorithm. This relies on a black-box reduction that depends only on the Fenchel conjugate of an upper bound on the linearized regret. Building on this theory, we propose UP-OCP, a parameter-free method for OCP, via a reduction to a two-asset portfolio selection problem, leveraging universal portfolio algorithms. We show strong finite-time bounds on the miscoverage of UP-OCP, even for polynomially growing predictions. Extensive experiments support that UP-OCP delivers consistently better size/coverage trade-offs than prior online conformal baselines. The code is available at https://github.com/AdaMLLab/up-for-ocp.

## 1. Introduction

Reliable uncertainty quantification is a central goal in modern statistical learning, especially when predictions must be accompanied by measures of confidence. A popular way to express such uncertainty is through *prediction sets*, which return a set of predicted outcomes—rather than a single prediction—for a test point. The key challenge is to construct these sets so that they are both informative (small) and valid (they achieve a prescribed coverage level) without

[1]King Abdullah University of Science and Technology, Saudi Arabia [2]University of Pennsylvania, USA. Correspondence to: Tuo Liu <tuo.liu@kaust.edu.sa>.

*Proceedings of the 43$^{rd}$ International Conference on Machine Learning*, Seoul, South Korea. PMLR 306, 2026. Copyright 2026 by the author(s).

relying on strong distributional assumptions. Conformal Prediction (CP) has emerged as a prominent methodology for constructing prediction sets with finite-sample statistical validity (see, e.g., Saunders et al., 1999; Vovk et al., 1999; Papadopoulos et al., 2002; Vovk et al., 2005; Vovk, 2013; Chernozhukov et al., 2018; Lei et al., 2013; 2018b; Guan, 2023; Romano et al., 2020). CP can wrap around any predictive model to produce sets that contain the true label with a user-specified probability of at least $1 - \alpha$. This framework has been used in various settings, including regression, classification, and structured prediction.

In its most basic form, CP provides coverage if the data points are *exchangeable*. Since data streams are often not exchangeable, conformal methods have been developed that can account for various distribution shifts, such as covariate shift (Tibshirani et al., 2019; Qiu et al., 2023; Park et al., 2022), label shift (Podkopaev & Ramdas, 2021; Si et al., 2024), more general distribution shifts (Barber et al., 2023; Gauthier et al., 2025), and time series models (Xu & Xie, 2021; Zaffran et al., 2022).

A different line of work, sometimes called Online Conformal Prediction (OCP), aims to avoid making assumptions about the data entirely, and instead considers *deterministic* and *adversarial* data (see, e.g., Gibbs & Candes, 2021; Zaffran et al., 2022; Bastani et al., 2022; Gibbs & Candès, 2024; Angelopoulos et al., 2023; Podkopaev et al., 2024). In this setting, one aims to achieve coverage $1 - \alpha$ averaged over time (and any possible algorithmic randomness). This work can be viewed as belonging to the setting of *online learning* (Cesa-Bianchi & Lugosi, 2006; Hazan, 2016; Orabona, 2019).

More formally, the observed data $((X_t, Y_t))_{t \geq 1}$ arrive over time $t = 1, 2, \ldots$. At each time point (or round) $t$, a prediction set $\hat{C}_t$ is constructed for $Y_t$ using all the previously observed data points $((X_i, Y_i))_{i \leq t-1}$, as well as the current features $X_t$. Let $\hat{Y}_t$ represent a point prediction given by a model $\hat{f}_t$ trained using all the information available before the true response $Y_t$. We are interested in regression problems, focusing on the perhaps most popular form of centered prediction sets (see, e.g., Lei et al., 2018a): $\hat{C}_t(b) := [\hat{Y}_t - b, \hat{Y}_t + b]$, defined as empty if $b < 0$. The goal is to design a conformal predictor whose observed long-term miscoverage rate is close to the nominal level, denoted

as $\alpha \in (0, 1)$. Formally, we aim to construct a sequence of radii $(b_t)_{t \geq 1}$ so that, as the time horizon $T$ grows, the corresponding prediction sets satisfy

$$\lim_{T \to \infty} \left| \frac{1}{T} \sum_{t=1}^{T} \mathbf{1} \left\{ Y_t \notin \hat{C}_t(b_t) \right\} - \alpha \right| = 0 . \quad (1)$$

**Related Work.** Adaptive Conformal Inference (ACI) (Gibbs & Candes, 2021) maintains a quantile threshold $\alpha_t$ over time $t = 1, 2, \ldots$, and updates it via Online Subgradient Descent (OSD) on the pinball (aka quantile) loss. ACI achieves long-term coverage close to the target $1 - \alpha$ level. However, its performance depends heavily on the step size. A small step size results in slow adaptation and miscoverage after a large distribution shift, while a large step size induces high variance and instability in the prediction set widths (Gibbs & Candès, 2024; Angelopoulos et al., 2023).

To address this limitation, later work introduces alternatives. Multi-valid Conformal Prediction (MVP) (Bastani et al., 2022) selects a threshold with the best historical coverage from a discretized grid. MVP guarantees long-term and threshold-calibrated coverage at multiple levels, but lacks rapid adaptivity to abrupt changes (Bastani et al., 2022).

However, the above approaches require tuning a step size, which is challenging in the online setting. Because the sequences are adversarial, we cannot rely, for example, on train-test validation to tune hyperparameters. To avoid this problem, Zaffran et al. (2022) propose Aggregated ACI, which uses online expert aggregation, running multiple copies of ACI with various step sizes and forming a weighted ensemble. Relatedly, Dynamically-Tuned ACI (DtACI) (Gibbs & Candès, 2024) re-weights experts in order to emphasize recent data; in effect it tunes ACI's step size online by minimizing a quantile loss. However, as noted by Angelopoulos et al. (2023), ACI-based methods can sometimes output infinite or null prediction sets, when $\alpha_t$ drifts below zero or above unity, respectively. Strongly Adaptive OCP (Bhatnagar et al., 2023) aggregates a number of scale-free base algorithms (Orabona & Pál, 2018) using the algorithm in Jun et al. (2017) to guarantee the worst-case regret over each subinterval, but it requires uniformly bounded predictions and knowledge of the maximum prediction range; both assumptions often fail to hold in practice. Zhang et al. (2024) and Podkopaev et al. (2024) instead avoid the use of a step size by using "parameter-free" online learning algorithms (Orabona, 2014; Orabona & Pál, 2016; Cutkosky & Orabona, 2018; Orabona & Pál, 2021; Chen et al., 2022b;a; Jacobsen & Cutkosky, 2022; Zhang et al., 2022).

A different approach has been recently proposed by Srinivas (2026), where one directly addresses the optimal trade-off between coverage and size of the confidence sets in a competitive ratio framework. Notably, their analysis confirms

that robust coverage in the worst-case setting fundamentally necessitates larger prediction sets. However, their algorithm, like the one by Bhatnagar et al. (2023), requires uniformly bounded predictions and knowledge of the maximum prediction range.

A complementary line of work frames online conformal calibration as a feedback-control problem and proposes conformal P/PI/PID controllers (Angelopoulos et al., 2023). Unlike these controller-based schemes—which introduce gain hyperparameters and typically require controller-specific analyses—we are interested in parameter-free approaches, more suited to the online setting.

Moreover, prior work used specialized analyses to prove that the algorithms can guarantee asymptotic coverage. To date, the precise connection between online learning and OCP is unclear. In online learning, the central goal is to obtain sublinear *regret*, i.e., the difference between the cumulative loss of the algorithm and that of the best fixed predictor chosen in hindsight. Algorithms that satisfy this property are said to be *no-regret*. However, Angelopoulos et al. (2025) also show that achieving coverage is, in general, completely distinct from achieving sublinear regret. Similarly, as we discuss later, the notion of proximal regret (Cai et al., 2024) also implies coverage, but has been established only for gradient descent. This makes it unclear when one can port methods from online learning for OCP.

**Contributions.** We answer the following questions:

*Is there a form of regret that implies coverage (see (1))?*

In Section 3, we show that the general notion of *linearized regret* directly implies coverage. Since linearized-regret bounds have been established for several online learning methods, this enables us to directly obtain coverage guarantees for these methods. In particular, we answer this question by making a connection to the regret-reward duality from online learning (Orabona, 2019). Once a bound on the miscoverage is established, it is natural to ask about optimality:

*Is it possible to construct an optimal online algorithm (in terms of regret and coverage) for OCP?*

This remained unexplored in the past literature. In Section 4, we design a parameter-free strategy that guarantees the best known finite-time coverage guarantee. Following classical work in parameter-free online learning (see, e.g., Orabona, 2019), this is achieved by observing the equivalence between the OCP problem and a gambling problem, then using universal portfolio algorithms (Cover & Ordentlich, 2002) to optimally solve it. We will also show that our algorithm guarantees online coverage with any polynomial growth of the nonconformity scores. We call the resulting algorithm Universal Portfolio OCP (UP-OCP).

Finally, we introduce a way to empirically quantify the *trade-off* between size and coverage, for a wide range of values of $\alpha$. In our extensive experiments on real and simulated datasets, UP-OCP achieves the best such trade-off among a number of strong baselines (Section 5).

## 2. Notation and Problem Setup

In this section, we formally introduce our notation and the problem setup.

**Notation.** We define here some basic concepts and tools from convex analysis (see, e.g., Rockafellar, 1970). For a function $f : \mathbb{R} \to \mathbb{R}$, we define a *subgradient* of $f$ at $x \in \mathbb{R}$ as $g \in \mathbb{R}$ that satisfies $f(y) \geq f(x) + g(y - x)$, $\forall y \in \mathbb{R}$. The set of subgradients of $f$ at $x$ is called the *subdifferential set* and we denote it by $\partial f(x)$. The *indicator function of the set* $\mathcal{V}$, $\mathbf{1}_{\mathcal{V}} : \mathbb{R} \to (-\infty, +\infty]$, has value 0 for $x \in \mathcal{V}$ and $+\infty$ otherwise. For a function $f : \mathbb{R} \to [-\infty, \infty]$, we define the *Fenchel conjugate* $f^{\star} : \mathbb{R} \to [-\infty, \infty]$ as $f^{\star}(\theta) = \sup_{x \in \mathbb{R}} (\theta x - f(x))$. The Fenchel conjugate is always well-defined and convex.

**Problem Setup.** We consider the problem of OCP, for arbitrary data streams, even adversarially generated ones, as introduced in Section 1. Let $S_t \geq 0$ denote the radius of the smallest prediction set that contains the true response $Y_t$, i.e., $S_t := \inf\{b \in [0, \infty) : Y_t \in \hat{C}_t(b)\} = |Y_t - \hat{Y}_t|$. We will also refer to $S_t$ as the nonconformity score (Vovk et al., 2005). Here, we use scores measured in absolute residual (centered intervals) for a concrete demonstration, but our methodology is not restricted to this choice: it applies to any nested prediction sets controlled by a scalar threshold, including asymmetric intervals in general. We defer these details to Appendix E for brevity. In terms of $S_t$, the target property (1) is equivalent to

$$\lim_{T \to \infty} \left| \frac{1}{T} \sum_{t=1}^{T} \mathbf{1}\{b_t < S_t\} - \alpha \right| = 0 .$$

This can be viewed as the problem of sequentially learning the $(1 - \alpha)$-th quantile of the nonconformity scores $(S_t)_{t \geq 1}$.

A standard approach (see, e.g., Gibbs & Candes, 2021; Podkopaev et al., 2024) to learning this quantile is to use a proper scoring rule (see, e.g., Gneiting & Raftery, 2007), namely the *pinball (or quantile) loss*, defined as

$$\ell^{(1-\alpha)}(b, S) := \max\{(1-\alpha)(S-b), \alpha(b-S)\}, \quad (2)$$

where $S$ is the nonconformity score and $b$ is the radius of the prediction interval. This loss is convex and $L$-Lipschitz in the first argument, where $L := \max\{1 - \alpha, \alpha\}$. These two properties make it online learnable (see, e.g., Hazan, 2016; Orabona, 2019; Cesa-Bianchi & Orabona, 2021). For $t \geq 1$, let $b_t$ be the prediction, and $\ell_t(b_t) := \ell^{(1-\alpha)}(b_t, S_t)$ be the

loss at round $t$. The *regret* of the algorithm with respect to any fixed comparator $u \in \mathbb{R}$ is defined as

$$\text{Regret}_T(u) := \sum_{t=1}^{T} \ell_t(b_t) - \sum_{t=1}^{T} \ell_t(u) . \quad (3)$$

The subdifferential set of $\ell_t$ is

$$\partial \ell_t(b) = \begin{cases} \{\mathbf{1}\{b \geq S_t\} - (1 - \alpha)\}, & b \neq S_t \\ [-(1-\alpha), \alpha], & b = S_t . \end{cases}$$

At $b = S_t$, there is an infinite set of subgradients. Throughout, we adopt the convention of selecting the right subgradient $g_t = \alpha$ when $b_t = S_t$, so that $g_t \in \{-(1-\alpha), \alpha\}$ for all $t$. Other choices are possible and essentially equivalent. Thus, an online learning algorithm predicting $b_t$ and receiving the pinball loss $\ell_t$ will receive the subgradient

$$g_t = \mathbf{1}\{b_t \geq S_t\} - (1 - \alpha) . \quad (4)$$

As explained by Gibbs & Candes (2021); Angelopoulos et al. (2025), the miscoverage error is closely related to the observed subgradients, because

$$\begin{aligned} \text{MisCov}_T &:= \left| \frac{1}{T} \sum_{t=1}^{T} \mathbf{1}\{b_t \geq S_t\} - (1 - \alpha) \right| \\ &= \frac{1}{T} \left| \sum_{t=1}^{T} g_t \right| . \end{aligned} \quad (5)$$

## 3. Coverage Guarantees for No-Regret Algorithms

In this section, we describe our main result providing a coverage guarantee for any online algorithm controlling an appropriate form of linearized regret.

Consider an online learning algorithm that in each round $t = 1, 2, \ldots$ produces an action $b_t \in \mathbb{R}$, and let $g_t \in \partial \ell_t(b_t)$ denote a subgradient of the loss at round $t$. We consider the *linearized regret* (Gordon, 1999; Zinkevich, 2003) of the algorithm on this sequence at the action $u \in \mathbb{R}$, defined as

$$\text{LinRegret}_T(u) := \sum_{t=1}^{T} g_t(b_t - u) . \quad (6)$$

In contrast to the standard notion of regret from (3), this quantity sums the linearizations $g_t(b_t - u)$ of the loss differences $\ell_t(b_t) - \ell_t(u)$ around $b_t$. By the definition of subgradients, we have that $\text{Regret}_T(u) \leq \text{LinRegret}_T(u)$ for all $u$. Thus, *any algorithm that controls the linearized regret also controls the usual regret*. However, an algorithm may control regret but not linearized regret. Crucially, *our analysis shows that controlling the linearized regret suffices to ensure coverage*.

Specifically, we have the following result which bounds the range of the sum of the gradients $\sum_{t=1}^{T} g_t$ via the Fenchel conjugate of a bound on the regret function (proof in Appendix A). Due to (4), this immediately implies a bound on the coverage.

**Theorem 3.1.** *For an online learning algorithm, let $F_T : \mathbb{R} \to \mathbb{R}$ such that $\mathrm{LinRegret}_T(u) \leq F_T(u)$ on the pinball losses $(\ell_t)_{1 \leq t \leq T}$. Then,*

$$-\sum_{t=1}^{T} g_t \in \left\{ z \in \mathbb{R} : F_T^{\star}(z) \leq (1-\alpha) \sum_{t=1}^{T} S_t \right\}, \quad (7)$$

*where $F_T^{\star}(\cdot)$ is the Fenchel conjugate of $F_T(\cdot)$.*

*Proof sketch (full proof in Appendix A).* Two ingredients. (i) Lemma A.1 shows $-g_t b_t \leq (1-\alpha)S_t$ for every $t$ using the piecewise-linear structure of the pinball loss, giving $-\sum_t g_t b_t \leq D_T := (1-\alpha) \sum_t S_t$. (ii) From the regret bound $\sum_t g_t(b_t - u) \leq F_T(u)$, rearranging and taking $\sup_u$ yields $F_T^{\star}(-\sum_t g_t) \leq D_T$. Chaining the two yields (7). □

Since $F_T^{\star}(|\theta|)$ is convex and grows with $|\theta|$, this constrains $|\sum_t g_t|$ (and hence miscoverage) to be small whenever $F_T$ is sublinear. Further, because conjugation is order-reversing, this reduction is *monotone*: a uniformly tighter linearized-regret bound provably shrinks the certified range of $\sum_{t=1}^{T} g_t$, hence the miscoverage (see Proposition B.2 in Appendix B for a formal statement).

In Appendix D, we show that a simpler asymptotic coverage result can be obtained more directly from our theory, without requiring the machinery of Fenchel conjugates. Moreover, since, as we discussed, coverage can be achieved in trivial ways, it is also important to have other correctness guarantees. For this reason, in the standard conformal prediction setting of i.i.d. scores, we show in Appendix F that any no-regret algorithm ensures that the averaged thresholds $b_t$ converge to the optimal one. This provides an additional desired correctness guarantee in our framework.

*Remark* 3.2. Angelopoulos et al. (2025, Example 1) show that sublinear regret does not imply coverage. Their example reduces to the following: consider $S_t = S > 0$ for all $t$, and an online learning algorithm that predicts $S + 1/\sqrt{t}$. While the regret with respect to the optimal prediction $S$ is sublinear, $\alpha \sum_{t=1}^{T} 1/\sqrt{t} \leq 2\alpha\sqrt{T}$, we have that $g_t = \alpha$ for all $t$, so the coverage error does not vanish. This is not a contradiction. Specifically, as already discussed, the linearized regret is stronger than the standard one. By taking $u = S - \varepsilon$, where $\varepsilon > 0$, as the competitor in $\mathrm{LinRegret}_T(u)$, we see that the linearized regret is *not controlled*, growing at least as $\alpha\varepsilon T$. Hence, $b_t = S + 1/\sqrt{t}$ does not control the linearized regret.

*Remark* 3.3. Cai et al. (2024) derive coverage guarantees for OSD by showing that it minimizes a notion called *proximal regret* for the specific class of linear functions. However, they only prove this property for OSD, while we handle any online algorithm with a suitable linearized regret. Angelopoulos et al. (2025) discuss another notion, *no-move regret*, as a special case of proximal regret, and show a corresponding asymptotic coverage result for smooth losses. Our guarantees are instead derived for algorithms minimizing the non-smooth pinball loss.

*Remark* 3.4. Ramalingam et al. (2025) also studies the relationship between no-regret learning and online conformal prediction, and explicitly observe that standard (external) regret on the pinball loss is *insufficient* to ensure marginal coverage in adversarial settings. They showed equivalence between *swap regret* and multi-valid coverage under mild smoothness assumptions on the score distribution and, extended the ACI bound of Gibbs & Candes (2021) to FTRL via a direct iterate-magnitude analysis. However, we identify *linearized regret*, a mild strengthening of external regret as the right notion that, on its own, implies fully distribution-independent marginal coverage for *any* no-regret algorithm on the pinball loss.

**Warm-up: Coverage Guarantee for KT and OSD.** To show the generality of our result, we first present a coverage analysis for the KT approach in Podkopaev et al. (2024). This consists of the parameter-free KT algorithm of Orabona & Pál (2016), applied to the sequence of pinball losses. Podkopaev et al. (2024) only provide the asymptotic guarantee (1), while here we obtain finite-time convergence using Theorem 3.1.

As proved by Orabona & Pál (2016), a valid choice for the Fenchel conjugate $F_T^{\star}$ for the KT algorithm is

$$F_T^{\star}(\theta) = (24T)^{-1/2} \exp(\theta^2/(2T)) - 1 . \quad (8)$$

We now use the mild assumption that the growth rate of $S_t$ is bounded polynomially: let $D > 0$ and $q \geq 0$, and assume $S_t \leq Dt^q$ for all $t$. This assumption is strictly weaker than the standard boundedness condition ($S_t \leq C$) or i.i.d. assumptions typically required in prior work. It allows our guarantees to hold even in non-stationary environments where the scale of nonconformity scores expands over time. As we demonstrate in Appendix K, this polynomial growth model is a more realistic representation of real-world dynamics than a static bound.

Now, using Theorem 3.1, we immediately obtain

$$\left| -\sum_{t=1}^{T} g_t \right| \leq \sqrt{2T \ln\left( \frac{\sqrt{24}D(1-\alpha)}{q+1} T^{3/2+q} + \sqrt{24T} \right)}.$$

By (4), the miscoverage is $\mathrm{MisCov}_T = \mathcal{O}(\sqrt{\ln(DT)/T})$, regardless of the growth rate exponent $q$.

To show the full generality of our approach based on the conjugate of the linearized regret, in Appendix C, we also show a minor variant of Theorem 3.1 specialized for OSD with step size $\eta$ and $b_1 = 0$, which, under the same assumptions on $S_t$, gives the following bound: $\text{MisCov}_T = (1/T)\left|\sum_{t=1}^{T} g_t\right| \leq (1/T)\left(DT^q/\eta + 1\right)$. With $q = 0$, this rate matches the one in Gibbs & Candes (2021), extending their analysis to the case where the scores $S_t$ can grow over time. We discuss the sharpness of this reduction at the end of Appendix A and analyze potential source of looseness.

# 4. A Universal-Portfolio Based Strategy

Thanks to Theorem 3.1, we now have a direct relationship between the linearized regret of an online algorithm and its miscoverage error, through the Fenchel conjugate $F_T^\star$. Since a tighter (smaller) regret bound $F_T$ corresponds to a larger, steeper $F_T^\star$, minimizing regret leads to better coverage bounds. Thus, for a fast bound on coverage, it is desirable to use an online algorithm with optimal regret.

It is known that the optimal linearized regret in unconstrained online learning is achieved only by parameter-free algorithms, such as those in Zhang et al. (2024) and Podkopaev et al. (2024); see the lower bound in Orabona (2019, Section 5.3). However, these algorithms are not fully optimal for OCP because they implicitly assume a degree of symmetry (as explained below), whereas the coverage problem is inherently asymmetric: the target miscoverage rate $\alpha$ is typically chosen to be small (e.g., $\alpha = 0.05$ or $0.01$), implying that the positive and negative subgradients of the losses are very different.

Instead, we propose reducing our problem to a portfolio selection problem, and leveraging the *Universal Portfolio* (UP) algorithm (Cover & Ordentlich, 2002) for OCP. In the following, we explain how UP methods lead to optimal solutions to online learning problems with asymmetric subgradients. For our reduction, considering pinball losses, we construct a market with two synthetic stocks, whose market gains are driven by the observed miscoverage.

**Definition 4.1** (The Conformal Market). For a miscoverage rate $\alpha \in (0, 1)$, given the subgradient $g_t \in \{-(1 - \alpha), \alpha\}$, $t \geq 1$ defined in (4), we define the vector of *returns* $\boldsymbol{w}_t = (w_{t,1}, w_{t,2})^\top \in \mathbb{R}^2$ of two synthetic stocks as

$$w_{t,1} = -g_t/\alpha + 1, \quad w_{t,2} = 1 + g_t/(1 - \alpha) .$$

The returns are the coordinates of a *market gain vector* $\boldsymbol{w}_t$ representing the ratio of the closing price to the opening price for the two stocks.

We have $w_{t,1}, w_{t,2} \geq 0$, because $-g_t \in \{-\alpha, 1 - \alpha\}$. Stock 1 yields high returns when coverage is lost ($g_t < 0$), while Stock 2 yields moderate returns when coverage is maintained ($g_t > 0$).

We can now formally define the wealth of an algorithm operating in this market.

**Definition 4.2** (Wealth Process). Consider an online algorithm that, at each round $t$, chooses a portfolio weight $\lambda_t \in [0, 1]$ representing the fraction of capital invested in Stock 1. The *wealth* $W_t$ is defined as $W_0 = 1$ and

$$W_t = W_{t-1} \cdot (\lambda_t w_{t,1} + (1 - \lambda_t)w_{t,2}) . \tag{9}$$

The Universal Portfolio algorithm computes the weight $\lambda_t$ as the wealth-weighted average over the simplex of all possible constant portfolios. Let $W_{t-1}(\lambda) = \prod_{i=1}^{t-1}(\lambda w_{i,1} + (1 - \lambda)w_{i,2})$ be the wealth of a constant portfolio $\lambda$. Given a prior $\mu$ over $\lambda \in [0, 1]$, the prediction is

$$\lambda_t = \frac{\int_0^1 \lambda \cdot W_{t-1}(\lambda)d\mu(\lambda)}{\int_0^1 W_{t-1}(\lambda)d\mu(\lambda)} . \tag{10}$$

Choosing $d\mu(\lambda) = 1/\left[\pi\sqrt{\lambda(1 - \lambda)}\right]d\lambda$, Cover & Ordentlich (2002) proved that the log wealth of the algorithm is at least the log wealth, Cover & Ordentlich (2002) proved that the log wealth of the algorithm is at least the log wealth of the best constant $\lambda$ in each round, up to a slack of $\frac{1}{2}\ln(\pi(T + 1))$. This regret guarantee is optimal up to constant additive factors (Cover & Ordentlich, 2002).

The critical insight formalized in the following theorem—which follows from Orabona & Jun (2023, Lemma 1) and the standard reduction of OCO to coin betting (Orabona, 2019)—is that *maximizing the logarithmic growth of this wealth is equivalent to minimizing the linearized regret on the pinball loss*. This allows us to translate the wealth guarantees of portfolio algorithms to coverage through (7). The proof is in Appendix G.

**Theorem 4.3** (Regret of UP-OCP). *Let $\mathcal{A}$ be the universal portfolio algorithm with $d\mu(\lambda) = 1/\left[\pi\sqrt{\lambda(1 - \lambda)}\right]d\lambda$ that outputs weights $(\lambda_t)_{t\geq 1}$ on the conformal market returns $\boldsymbol{w}_t \in \mathbb{R}^2$. Define $b_t$ via*

$$b_t = W_{t-1} \cdot \left[-(1 - \alpha)^{-1} + \lambda_t/(\alpha(1 - \alpha))\right] . \tag{11}$$

*Then, the resulting sequence of actions $(b_t)_{t\geq 1}$ achieves linearized regret $\text{LinRegret}_T(u) \leq F_T(|u|)$ for all $u \in \mathbb{R}$ in online quantile loss minimization, where $F_T(|u|)$ is*

$$\max\left\{|u|\sqrt{2T\alpha(1 - \alpha)\ln(4\alpha(1 - \alpha)(T + 1)^{3/2}u^2 + 1)},\right.$$
$$\left.\frac{4}{3}|u|\left(\ln(3|u|\sqrt{T + 1}) - 1\right)\right\} .$$

*Proof sketch (full proof in Appendix G).* The proof constructs a two-stock *conformal market* (Definition (4.1)) whose returns encode the pinball loss subgradients: $w_{t,1} = 1 + g_t/\alpha$ and $w_{t,2} = 1 + g_t/(1 - \alpha)$. Lemma (G.1) shows

that any portfolio strategy on this market is *equivalent* to a coin-betting strategy with bet $b_t = \beta_t W_{t-1}$, where $\beta_t = (\lambda_t - \alpha)/(\alpha(1-\alpha))$ and $W_{t-1}$ is the accumulated wealth. By coin-betting/Fenchel duality, $W_T - 1 \geq \Psi_T(\sum_t c_t)$ implies $\text{LinRegret}_T(u) \leq \Psi_T^\star(u)$. The best constant rebalanced portfolio gives $\ln W_T^\star = T \cdot \text{KL}(\alpha + \sum_{t=1}^T c_t/T \| \alpha)$; combining the log-wealth regret of Cover's universal portfolio $\mathcal{R}_T = (1/2)\ln(\pi(T+1))$ (Cover & Ordentlich, 2002) with the Bernstein-type bound of Lemma H.1 and taking the Fenchel conjugate yields the stated $F_T$. $\qquad\square$

In the worst case, the regret is of order $\mathcal{O}(\sqrt{T \ln T})$. However, when $\alpha \to 1$, the bound is of order $\ln T$. In Appendix H, we also show that the regret upper bound for UP-OCP is better than the one for the KT approach from Podkopaev et al. (2024). We also discuss the sharpness of this portfolio-to-regret reduction at the end of Appendix G.

### 4.1. Coverage Guarantee for UP-OCP

We now provide a coverage guarantee for the proposed UP-OCP strategy. We also quantify its advantage over the Krichevsky-Trofimov (KT) bettor, particularly in the regime of small $\alpha$. The argument relies on a second-order expansion of the optimal wealth, which reveals that *UP adapts to the variance of the gradients*, whereas KT implicitly assumes a worst-case symmetric variance.

**Theorem 4.4** (Coverage bound for UP-OCP). *Let $\alpha \in (0,1)$ and let $(g_t)_{t=1}^T$ be the sequence of subgradients observed by UP-OCP, with $g_t \in \{-(1-\alpha), \alpha\}$. Let $D > 0$ and $q \geq 0$, and assume that $S_t \leq Dt^q$ for all $t$. For every integer $T \geq 1$, define*

$$\varepsilon_T := \frac{1}{T}\left[\ln\left(1 + \frac{(1-\alpha)D(T+1)^{q+1}}{q+1}\right) + \frac{1}{2}\ln(\pi(T+1))\right].$$

*Then,*

$$\text{MisCov}_T = \frac{1}{T}\left|\sum_{t=1}^T g_t\right| \leq \varepsilon_T + \sqrt{2\alpha(1-\alpha)\varepsilon_T}.$$

**Comparison with Previous OCP Algorithms.** Now, we compare the coverage rates of UP-OCP with existing bounds. While we only compare upper bounds and not actual coverage, we will show in Section 5 that the bounds are consistent with the empirical results. To make the dependence on the problem parameters explicit, Table 1 instantiates the regret-to-coverage reduction of Theorem 3.1 for three representative algorithms, translating each linearized-regret bound into a miscoverage rate.

First, consider the coverage guarantee for OSD methods. In OSD, one has to choose a step size $\eta$. The OSD bound decreases with $\eta$, so one might be tempted to set $\eta$ to be

large. However, in that case, OSD would predict zero on the first round and, if the scores were bounded by $D$, values larger than $D$ later, until predicting a non-positive number. The cycle would then repeat. This behavior would not be informative for uncertainty quantification.

A more meaningful choice of $\eta$ is the one that minimizes the worst-case regret, i.e., $\eta = \sqrt{DT^{q-1}}/\max(\alpha, 1-\alpha)$. This gives a coverage bound of $\sqrt{DT^q}\max(\alpha, 1-\alpha)/\sqrt{T} + 1/T$. In contrast to the bound we derived for the KT strategy, this rate deteriorates with $q$. However, things are even worse: the choice of $\eta$ that depends on $q$ and $D$ cannot be used, because $D$ and $q$ are not available to the algorithm. In this situation one can only use the step size $\eta = c/[\max(\alpha, 1-\alpha)\sqrt{T}]$ where $c > 0$ is a hyperparameter. With this choice the coverage will converge at the worse rate of $DT^{q-1/2}\max(\alpha, 1-\alpha)/c + 1/T$.

Overall, we can see that OSD, whether tuned with oracle knowledge of $q$ and $D$ or untuned, has worse dependence on $T$ if $q > 0$. Moreover, when $\alpha \to 1$ or $\alpha \to 0$, the coverage rate of UP-OCP approaches $\ln T/T$ while that of OSD cannot be faster than $1/\sqrt{T}$.

Next, consider the KT strategy. UP-OCP and the KT strategy have the same dependence on $T$, $D$, and $q$. However, KT is designed to be minimax optimal for coin-betting games with symmetric outcomes, that is, with $\alpha = 1/2$. So, as for OSD, we see that the rate of the KT strategy does not improve when $\alpha \to 1$ or $\alpha \to 0$. A similar rate was shown for a parameter-free algorithm in Zhang et al. (2024), but only in an asymptotic sense. Moreover, the betting strategy implicit in the algorithm in Zhang et al. (2024) is provably inferior to that of universal portfolio methods, because it matches only the leading term of the growth rate of the best rebalanced portfolio.

### 4.2. Closed-Form Update for Universal Portfolios

A direct evaluation of (10) can be implemented with cumulative time complexity up to time $t$ of $\mathcal{O}(t^2)$ (Cover & Ordentlich, 2002). However, for our specific Conformal Market defined in Definition 4.1, there is a simple closed-form update $\lambda_t = t^{-1}(\sum_{i=1}^{t-1} \mathbf{1}\{g_i = -(1-\alpha)\} + 1/2)$, see Appendix I. Substituting this into (11) yields a parameter-free update rule for the conformal radius $b_t$ that adapts to the asymmetry of the gradients. See Algorithm 1 for the complete pseudocode.

The mapping from wealth to radius in (11) can produce negative values. So, in Algorithm 1, we clip the radius to zero. In Appendix H.2, we show that the regret of the truncated sequence is upper bounded by the regret of the original sequence, and that the subgradients remain unchanged, preserving the theoretical guarantees.

Note the multiplicative nature of the algorithm, characteris-

*Table 1.* The regret-to-coverage reduction of Theorem 3.1 instantiated for three algorithms, under polynomial growth $S_t \leq Dt^q$ ($D > 0$, $q \geq 0$). The last three columns indicate whether the method is parameter-free, convergence rate degrades with polynomially growing scores ($q > 0$), and improves as $\alpha \to 0$. Here $\Lambda_s := \ln[(1-\alpha)DT^s]$. Note that UP-OCP improves to $\mathcal{O}(\ln T/T)$ as $\alpha \to 0$.

| Algorithm | Linearized regret $F_T(|u|)$ | Miscoverage rate $\mathrm{MisCov}_T$ | Parameter-free | $S_t \to \infty$ | $\alpha \to 0$ |
|---|---|---|---|---|---|
| OSD | $u^2/(2\eta) + \eta T/2$ | $\mathcal{O}\big(T^{q-1}/\eta + 1/T\big)$ | ✗ | ✗ | ✗ |
| KT bettor | $|u|\sqrt{T\ln(24T^2u^2+1)}+1$ | $\mathcal{O}\big(\sqrt{\Lambda_{q+3/2}/T}\big)$ | ✓ | ✓ | ✗ |
| **UP-OCP (ours)** | See Theorem 4.3 | $\mathcal{O}\big(\sqrt{\alpha(1-\alpha)\Lambda_{q+1}/T} + \Lambda_{q+1}/T\big)$ | ✓ | ✓ | ✓ |

---

**Algorithm 1** Universal Portfolio for OCP (UP-OCP)

---

**Input:** Target miscoverage rate $\alpha \in (0,1)$
**Initialize:** Wealth $W_0 \leftarrow 1$, miscoverage count $N \leftarrow 0$
**for** $t = 1, 2, \ldots$ **do**
   $\lambda_t \leftarrow \dfrac{N + 1/2}{t}$; $b_t \leftarrow \max\left(0, W_{t-1} \cdot \dfrac{\lambda_t - \alpha}{\alpha(1-\alpha)}\right)$
   Output prediction set $\hat{C}_t \leftarrow [\hat{Y}_t - b_t, \hat{Y}_t + b_t]$
   Observe true label $Y_t$
   Compute nonconformity score $S_t \leftarrow |Y_t - \hat{Y}_t|$
   **if** $S_t > b_t$ **then**
     $N \leftarrow N + 1$; $W_t \leftarrow W_{t-1} \cdot \lambda_t/\alpha$
   **else**
     $W_t \leftarrow W_{t-1} \cdot (1 - \lambda_t)/(1 - \alpha)$
   **end if**
**end for**

---

tic of all parameter-free algorithms, which allows it to adapt to any polynomially growing sequence of $S_t$.

## 5. Experiments

We support our theoretical findings through evaluations on both synthetic data and empirical time series. Our experiments cover the finance and energy domains, where data can be highly non-stationary. Across these settings, we compare the proposed UP-OCP method (Algorithm 1) with state-of-the-art parameter-free and tuned baselines.

**Datasets and Models.** It is notoriously difficult to test algorithms in the online setting, because it would require constructing adversarial sequences for each algorithm. Hence, we follow the OCP literature (Gibbs & Candes, 2021; Angelopoulos et al., 2023; Gibbs & Candès, 2024; Podkopaev et al., 2024) and choose standard non-adversarial benchmark datasets. We consider three categories of data: (1) daily opening prices (log-scale) of four major US stocks (American Express, Apple, Amazon, and Google) from 2008 to 2018 (Nguyen, 2018); (2) electricity demand records from New South Wales (Harries, 1999); and (3) synthetic environments. For the base predictive models, we employ domain-standard choices:

- Stock prices: We use `Prophet` (Taylor & Letham,

  2018), re-fitted daily.
- Electricity demand: We use a standard autoregressive (AR) model with lag 3.
- Synthetic data: Following the protocol in Angelopoulos et al. (2023, Appendix F.5), we bypass the prediction step and directly simulate the nonconformity scores $S_t$. We generate three distinct patterns: a sinusoidal wave with Gaussian noise, and two trends (constant and quadratic) with random sparse bumps. Full details are provided in Appendix N.

The synthetic tests simulate a forecaster with residuals exhibiting specific challenging patterns, isolating the OCP method's behavior from the base model's dynamics. The synthetic nature also allows us to use multiple trials to generate error bars.

In all cases, the nonconformity scores are defined as the absolute residuals $S_t = |Y_t - \hat{Y}_t|$. Due to space constraints, we focus our main analysis on the American Express (AXP) dataset and the synthetic sinusoidal environment. Full results for synthetic data, electricity demand, and other stock tickers are deferred to the Appendix.

**Baselines.** We compare UP-OCP with previous OCP algorithms that do not rely on prior knowledge of a uniform upper bound on the nonconformity scores:

- Krichevsky-Trofimov (KT) (Podkopaev et al., 2024).
- Dynamically-tuned Adaptive Conformal Inference (DtACI) (Gibbs & Candès, 2024).
- Scale-Free Online Gradient Descent (SF-OGD): A variant of OSD that adapts to the scale of the gradients (Orabona & Pál, 2018).
- Conformal P/PI Control (P/PI Ctrl) (Angelopoulos et al., 2023).

For the parameterized baselines, we performed a grid search to select the best hyperparameters based on ex-post performance. In contrast, UP-OCP, KT, and DtACI are fully parameter-free and require no tuning. Crucially, tuning baselines on ex-post data grants them an oracle advantage. This means that we may overestimate their performance. Also, we emphasize that reasonable performance for parameterized baselines depends heavily on fine-tuned hyperparameter choices. As we demonstrate in Appendix O, without

*Table 2.* Quantitative comparison on the AXP dataset. Performance metrics for UP-OCP versus parameter-free (KT, DtACI) baselines and tuned SF-OGD (lr=25).

| Metric | UP-OCP | KT | DtACI | SF-OGD |
|---|---|---|---|---|
| Marginal Coverage | 0.932 | 0.920 | 0.956 | 0.948 |
| Longest Err. Seq. | **4** | 15 | 6 | **3** |
| Avg. Set Size | **14.8** | 16.9 | ∞ | **16.4** |
| Median Set Size | **11.5** | 12.9 | 12.6 | 13.8 |
| 75% Quantile Size | **18.8** | 24.9 | 21.8 | 21.5 |
| 90% Quantile Size | **32.3** | 32.5 | ∞ | **32.3** |
| 95% Quantile Size | 38 | 36.1 | ∞ | 36.1 |

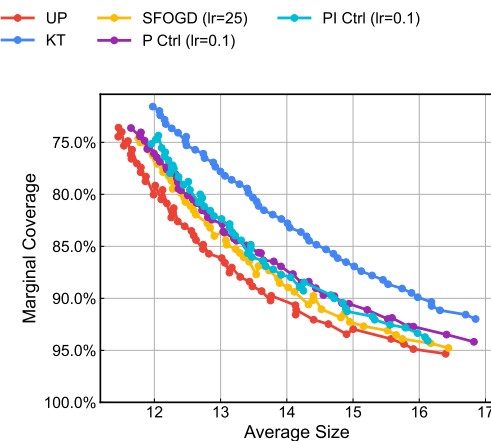

*Figure 1.* Pareto frontiers for average prediction set size on the AXP dataset, for 50 target miscoverage rates $\alpha$ uniformly from 0.05 to 0.25. Better performance is closer to the bottom-left corner.

careful tuning, they can fail to maintain valid coverage, showing poor behavior locally and globally. Detailed update rules and hyperparameter search grids are provided in Appendix J.

**Metrics.** From what we said above, it should be clear that the coverage alone is a meaningless metric. Indeed, we show a trivial predictor in Appendix J that achieves coverage without any informative prediction set. So, here *we heavily focus on the trade-off between coverage and prediction set sizes*, by proposing the use of Pareto frontier plots.

### 5.1. Results for AXP

We begin by evaluating the efficiency-coverage trade-off on the American Express (AXP) dataset, setting $\alpha = 0.05$. An initial warm-up period of 100 days is used for training the initial base forecaster. Table 2 reports the performance metrics. Due to space limitations, the full results for control-based methods are in Appendix M.

Our UP-OCP method achieves valid coverage; although slightly lower (around 93%) than the target, such differences may typically be ignorable in terms of practical importance (Podkopaev et al., 2024). Appendix L explains this minor gap and further proposes a principled heuristic that empirically closes the gap observed in finite samples without changing the Pareto frontier. Furthermore, UP-OCP limits consecutive miscoverage events to just four days, matching the best tuned baseline (SF-OGD), indicating a rapid correction mechanism. In contrast, the KT strategy suffers from a long error sequence of 15 days. This supports that the symmetric betting strategy of KT is too conservative for small $\alpha$, failing to expand intervals sufficiently fast during extreme events. UP-OCP also demonstrates competitive efficiency compared to KT across most size metrics. Specifically, UP-OCP achieves smaller set sizes across average, median, and 75% quantile metrics, while maintaining higher marginal coverage (0.932 versus 0.920). It is only in the extreme upper tails (95% quantiles) that UP-OCP produces larger sets, a necessary behavior to correct for miscoverage during high-volatility events.

Notably, the degeneracy of DtACI is clear from the table despite its valid marginal rate—the 90% quantile size is infinite. This supports that DtACI does not just produce large sets occasionally; it relies on trivial predictions to achieve coverage in at least 10% of the rounds. Instead, UP-OCP achieves coverage without generating infinite sets.

**Local Coverage and Efficiency.** Global coverage can be insufficient if the errors are grouped in time. To complement Table 2, we provide detailed 1-vs-1 comparisons of local adaptivity between UP-OCP and the baselines in Appendix M. These plots show that UP-OCP maintains local coverage tightly around the 95% target with no significant swings, while other methods exhibit marked volatility.

**Takeaway.** UP-OCP matches the performance of the oracle tuned SF-OGD, the best tuned baseline. Both methods achieve minimal error clustering and similar average set sizes. However, SF-OGD was ex-post tuned to select the optimal learning rate, whereas UP-OCP is parameter-free.

**Pareto Frontiers.** In general, higher coverage requires larger sets; however this trade-off varies across algorithms. Here, we prioritize the *average* prediction set size as our primary metric, as it reflects the cumulative cost of uncertainty in downstream tasks. Unlike the quantiles of the length, the average better captures unreasonably large prediction sets. This distinction is critical for separating stable algorithms from those that trivially satisfy coverage by outputting vacuous sets during volatility. In Appendix M, we also show that UP-OCP remains dominant across quantiles of the length.

Figure 1 illustrates that UP-OCP (red) empirically achieves the best Pareto trade-off. UP-OCP consistently achieves the smallest average set size for any given target coverage within the range $[0.75, 0.95]$. The parameter-free KT baseline (blue) is strictly suboptimal.[1]

---

[1]DtACI is excluded here, because it produces prediction sets

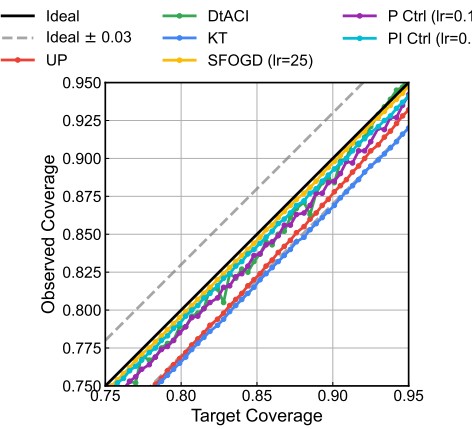

*Figure 2.* Realized versus target coverage.

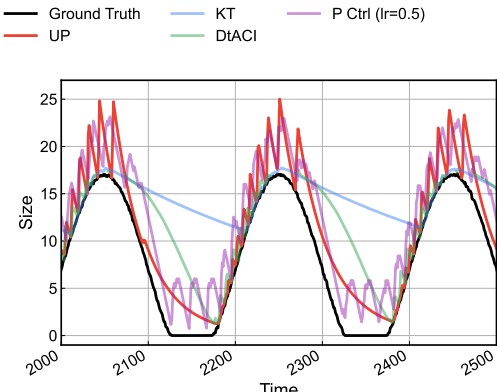

*Figure 3.* Interval widths over a representative window of the synthetic sinusoid dataset. The black curve shows the nonconformity score and the colored curves show the radii selected by each method.

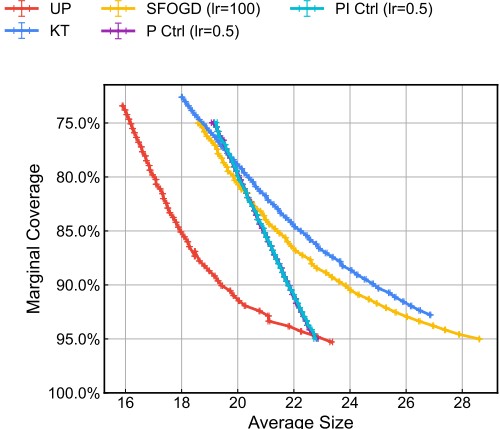

*Figure 4.* Plot of the realized marginal coverage against the average prediction set size, averaged over 10 independent random seeds. Error bars indicate the standard error of the mean along both axes.

**Target Calibration.** Beyond efficiency, a reliable OCP algorithm must also track the user-specified target $1-\alpha$. Figure 2 plots the realized versus target coverage for $\alpha \in [0.05, 0.25]$. The results confirm that UP-OCP and the baselines maintain calibration within a tight $\pm 0.03$ tolerance band (dashed lines) over all targets. In Appendix L, we also provide a heuristic to improve the tracking of any OCP algorithm.

### 5.2. Results for Synthetic Sinusoid

We analyze the performance on a synthetic dataset designed to test adaptivity to periodic volatility. Following Angelopoulos et al. (2023, Appendix F.5), we generate nonconformity scores $S_t$ as a sinusoid with Gaussian noise: $S_t = \max(0, [\sin((2\pi t)/P) + 0.5]S_{\mathrm{mag}} + S_{\min} + \varepsilon_t)$, where $\varepsilon_t \overset{\text{i.i.d.}}{\sim} \mathcal{N}(0, \sigma^2)$. We fix the period $P = 200$, magnitude $S_{\mathrm{mag}} = 10$, minimum offset $S_{\min} = 2$, and noise scale $\sigma = 0.3$. The total sequence length is $T = 3000$, and we report results averaged over 10 independent trials.

**Tracking Dynamics.** Figure 3 visualizes the evolution of interval widths over a representative window $t \in [2000, 2500]$. The ground truth (black) exhibits a clear periodic pattern. UP-OCP (red) tracks this, expanding rapidly during high-volatility phases ($t \approx 2050$) to guarantee coverage, then shrinking as the noise variance decreases. In contrast, KT (blue) shows significant lag and fails to reduce interval widths sufficiently during low-noise periods. DtACI (green) shows strong instability; the trace terminates after $t > 2400$ (marked by missing values in the plot), indicating the algorithm has diverged and is outputting *infinite* sets to reach coverage. Again, UP-OCP matches the best tuned baseline, P-Control (purple), without hyperparameter tuning. Other parameterized baselines behave similarly to the P controller; see Appendix N.5.

---

with infinite radius ($b_t = \infty$) to satisfy coverage; see Table 2. We observed this at all target levels, making the mean set size infinite.

**Pareto Frontiers.** Figure 4 quantifies the stability of these findings by showing error bars over random repetitions. Consistent with the AXP results, UP-OCP (red) dominates the baselines. The vertical error bars are negligible, confirming that all methods satisfy validity. In contrast, the size metrics show larger oscillations.

## 6. Discussion

This paper shows that coverage guarantees for online conformal prediction (OCP) can be derived from linearized regret bounds. Then it proposes UP-OCP, a parameter-free strategy for OCP using Universal Portfolio methods.

Open directions include extending the framework toward conditional/feature-dependent validity (Bastani et al., 2022), and local coverage guarantees from strongly adaptive regret (Bhatnagar et al., 2023).

## Acknowledgements

We thank the Frontiers of Statistical Inference 2025 Workshop, organized by Mladen Kolar at MBZUAI, for allowing FO and ED to meet and discuss the initial ideas for this paper. ED's work was supported in part by the US NSF, the ARO, AFOSR, the ARO, and the Sloan Foundation.

## Impact Statement

This paper presents work whose goal is to advance the field of machine learning. There are many potential societal consequences of our work, none of which we feel must be specifically highlighted here.

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

## A. Proof of Theorem 3.1

Our main results leverage the following simple but fundamental lemma. This lemma allows us to lower bound the term $g_t b_t$ which arises in the linearized regret, without proving a bound on the iterates of the algorithm.

**Lemma A.1.** *Let $b_t \in \mathbb{R}$, $t \geq 1$ be generated by any online learning algorithm and let $g_t \in \partial \ell_t(b_t)$ denote a subgradient of the pinball loss $\ell^{(1-\alpha)}(b, S_t)$ at $b = b_t$, for all $t$. Then, we have*

$$-g_t b_t \leq (1-\alpha)S_t, \qquad \forall t . \tag{12}$$

*Proof.* We consider cases. If $b_t < 0$, then $b_t < S_t$ since $S_t \geq 0$, hence $g_t = -(1-\alpha)$ and so $-g_t b_t = (1-\alpha)b_t \leq 0 \leq (1-\alpha)S_t$. If $0 \leq b_t < S_t$, then again $g_t = -(1-\alpha)$ and thus $-g_t b_t = (1-\alpha)b_t \leq (1-\alpha)S_t$. Finally, if $b_t \geq S_t$, then (including the tie case $b_t = S_t$) we have $g_t = \alpha$, so $-g_t b_t = -\alpha b_t \leq 0 \leq (1-\alpha)S_t$. $\quad\square$

We can now prove our main result.

*Proof of Theorem 3.1.* By the assumed linearized-regret bound, for every $u \in \mathbb{R}$ we have

$$\sum_{t=1}^{T} g_t(b_t - u) \leq F_T(u) .$$

Rearranging yields

$$\left(-\sum_{t=1}^{T} g_t\right) u - F_T(u) \leq -\sum_{t=1}^{T} g_t b_t .$$

Taking the supremum over $u \in \mathbb{R}$ gives

$$F_T^\star\left(-\sum_{t=1}^{T} g_t\right) \leq -\sum_{t=1}^{T} g_t b_t .$$

Finally, Lemma A.1 implies $-g_t b_t \leq (1-\alpha)S_t$ for all $t$, hence $-\sum_{t=1}^{T} g_t b_t \leq (1-\alpha)\sum_{t=1}^{T} S_t$. Combining the two inequalities completes the proof. $\quad\square$

**Sharpness of the reduction.** The reduction of Theorem 3.1 rests on exactly two steps, and it is useful to ask where, if anywhere, it loses tightness. *(i) The pointwise bound.* Lemma A.1 bounds $-g_t b_t$ by $(1-\alpha)S_t$ for *arbitrary* algorithms. This generic bound can be loose for specific algorithms, for instance, for OSD with $q = 0$ it yields an $\mathcal{O}(1/\sqrt{T})$ rate, whereas the sharp rate is $\mathcal{O}(1/T)$. Crucially, this looseness is not inherent to the framework but only to the generality of Lemma A.1: whenever algorithm-specific structure is available, one can substitute a tighter bound for $-\sum_t g_t b_t$ into the *same* proof template and recover the sharp rate. We make this concrete in Appendix C, where the bounded-iterate property of OSD (Lemma C.1) replaces Lemma A.1 and restores the $\mathcal{O}(1/T)$ rate of Gibbs & Candes (2021). *(ii) The Fenchel step.* The duality step itself is tight: the inequality $\left(-\sum_t g_t\right)u - F_T(u) \leq -\sum_t g_t b_t$ becomes the *equality* $F_T^\star\left(-\sum_t g_t\right) = -\sum_t g_t b_t$ at any $u$ attaining the supremum in the conjugate, so no slack is introduced here. We are not aware of lower bounds for the online miscoverage rate, so we cannot rule out that a better rate is achievable by an analysis that does not pass through the linearized regret; while we doubt this, settling it is an interesting open problem.

## B. Tighter Linearized Regret Implies Smaller Miscoverage

From the equivalence (5), the miscoverage at horizon $T$ is small precisely when $\left|\sum_{t=1}^{T} g_t\right|$ is small. We make precise here the sense in which Theorem 3.1 turns a tighter *linearized-regret* bound into a smaller *sum of subgradients*, and hence a smaller miscoverage. The mechanism is the order-reversing nature of the Fenchel conjugate.

**Lemma B.1** (Conjugation is order-reversing). *Let $f, g : \mathbb{R} \to (-\infty, +\infty]$ with $f(u) \leq g(u)$ for all $u \in \mathbb{R}$. Then $f^\star(z) \geq g^\star(z)$ for all $z \in \mathbb{R}$.*

*Proof.* For every $z$, $f^\star(z) = \sup_u (zu - f(u)) \geq \sup_u (zu - g(u)) = g^\star(z)$, since $-f(u) \geq -g(u)$ pointwise. $\quad\square$

Fix a loss sequence $(\ell_t)_{1 \leq t \leq T}$, let $g_t \in \partial \ell_t(b_t)$ be the realized subgradients, and write $D_T := (1 - \alpha) \sum_{t=1}^{T} S_t \geq 0$. For a valid linearized-regret bound $F_T$ (i.e., $\mathrm{LinRegret}_T(u) \leq F_T(u)$ for all $u$ along the algorithm's trajectory), define the *certificate set* and the *certified deviation*

$$\mathcal{Z}(F_T) := \{z \in \mathbb{R} : F_T^\star(z) \leq D_T\}, \qquad r(F_T) := \sup_{z \in \mathcal{Z}(F_T)} |z| \in [0, +\infty].$$

**Proposition B.2** (Monotone regret-to-miscoverage transfer). *With the notation above:*

(a) (Certificate.) $-\sum_{t=1}^{T} g_t \in \mathcal{Z}(F_T)$, *and therefore* $\mathrm{MisCov}_T = \frac{1}{T} \left| \sum_{t=1}^{T} g_t \right| \leq \frac{r(F_T)}{T}$.

(b) (Monotonicity.) *If* $F_T^A(u) \leq F_T^B(u)$ *for all* $u \in \mathbb{R}$, *then* $\mathcal{Z}(F_T^A) \subseteq \mathcal{Z}(F_T^B)$ *and* $r(F_T^A) \leq r(F_T^B)$. *That is, a uniformly tighter linearized-regret bound certifies a uniformly smaller range for* $\sum_{t=1}^{T} g_t$, *hence a smaller guaranteed miscoverage.*

*Proof.* **(a)** This is a restatement of Theorem 3.1: that theorem gives $F_T^\star(-\sum_t g_t) \leq D_T$, i.e., $-\sum_t g_t \in \mathcal{Z}(F_T)$; the displayed inequality then follows from $|-\sum_t g_t| \leq r(F_T)$ and (5).

**(b)** By Lemma B.1, $F_T^A \leq F_T^B$ pointwise implies $(F_T^A)^\star(z) \geq (F_T^B)^\star(z)$ for all $z$. Hence, if $z \in \mathcal{Z}(F_T^A)$, then $(F_T^B)^\star(z) \leq (F_T^A)^\star(z) \leq D_T$, so $z \in \mathcal{Z}(F_T^B)$. This proves $\mathcal{Z}(F_T^A) \subseteq \mathcal{Z}(F_T^B)$, and taking suprema of $|z|$ over the two nested sets gives $r(F_T^A) \leq r(F_T^B)$. $\qquad\square$

Parts (a)–(b) formalize the intuition: *any* improvement of the linearized-regret bound shrinks the certificate set $\mathcal{Z}(F_T)$ that localizes $\sum_t g_t$, so it can only decrease the guaranteed miscoverage.

## C. Proof of Coverage of OSD

We show that our analysis can recover the known coverage bound for online subgradient descent (Gibbs & Candes, 2021). From the well-known regret guarantee of online subgradient descent—OSD—see, e.g., Orabona (2019), for all $u \in \mathbb{R}$, we have the following bound on the linearized regret:

$$\mathrm{LinRegret}_T(u) \leq \frac{u^2}{2\eta} + \frac{\eta}{2} \sum_{t=1}^{T} g_t^2 =: F_T(u), \tag{13}$$

where $g_t$ are defined in (4). It follows that $F_T^\star(\theta) = \sup_{u \in \mathbb{R}} \theta u - F_T(u) = \frac{\eta \theta^2}{2} - \frac{\eta}{2} \sum_{t=1}^{T} g_t^2$. Let $D > 0$ and $q \geq 0$. Assuming that $S_t \leq Dt^q$ for all $t$, by Lemma A.1 and Theorem 3.1, the miscoverage can be bounded as

$$\frac{1}{T} \left| \sum_{t=1}^{T} g_t \right| \leq \frac{1}{\sqrt{T}} \sqrt{\frac{2(1 - \alpha)D}{(1 + q)\eta} T^q + \max(\alpha^2, (1 - \alpha)^2)}.$$

For the case of $q = 0$, Gibbs & Candes (2021) proved the better bound of $\mathcal{O}(1/(\eta T) + 1/T)$. The reason is that Lemma A.1 is not sharp in this specific case. Indeed, we can prove the following improved bound for OSD.

**Lemma C.1** (Improved Guarantee for OSD). *Let* $D > 0$ *and* $q \geq 0$, *and assume that* $S_t \leq Dt^q$ *for all* $t$. *The OSD algorithm with fixed step size* $\eta$ *and* $b_1 = 0$ *over a sequence of pinball losses* $\ell_t$ *guarantees, with* $g_t \in \partial \ell_t(b_t)$,

$$-\sum_{t=1}^{T} g_t b_t \leq \frac{(DT^q + \eta)^2}{2\eta} - \frac{\eta}{2} \sum_{t=1}^{T} g_t^2.$$

*Proof.* The (unconstrained) OSD update is $b_{t+1} = b_t - \eta g_t$. Expanding gives $b_{t+1}^2 = b_t^2 - 2\eta g_t b_t + \eta^2 g_t^2$. Summing from $t = 1$ to $T$ and using $b_1 = 0$ yields

$$\frac{b_{T+1}^2}{2\eta} = -\sum_{t=1}^{T} g_t b_t + \frac{\eta}{2} \sum_{t=1}^{T} g_t^2.$$

Thus, it suffices to show $|b_{T+1}| \leq DT^q + \eta$. Similarly to Gibbs & Candes (2021), we will show a boundedness property for the iterates $b_t$. We prove by induction that $|b_t| \leq D(t-1)^q + \eta$ for all $t$. The base case $b_1 = 0$ is immediate. Assume $|b_t| \leq D(t-1)^q + \eta$ and let's prove that $|b_{t+1}| \leq Dt^q + \eta$. If $0 \leq b_t < Dt^q$, then $|b_{t+1}| = |b_t - \eta g_t| \leq |b_t| + \eta|g_t| \leq Dt^q + \eta$ since $|g_t| \leq 1$. If $b_t \geq Dt^q$, then $b_t \geq S_t$ because $S_t \leq Dt^q$, hence $g_t = \alpha$ and $Dt^q - \eta \leq b_{t+1} \leq b_t \leq D(t-1)^q + \eta$. Finally, if $b_t < 0$, then $b_t < S_t$ and $g_t = -(1-\alpha)$, so $\eta \geq b_{t+1} \geq b_t \geq -D(t-1)^q - \eta$. In all cases, $|b_{t+1}| \leq Dt^q + \eta$, completing the induction.

Substituting $b_{T+1}^2 \leq (DT^q + \eta)^2$ into the identity yields the claim. $\qquad\square$

Using this improved guarantee instead of Lemma A.1 in the proof of Theorem 3.1, we have that

$$\frac{1}{T} \left| \sum_{t=1}^{T} g_t \right| \leq \frac{1}{T} \left( \frac{DT^q}{\eta} + 1 \right) .$$

## D. Asymptotic Coverage from Sublinear Regret

Now, we show an asymptotic coverage result, for any algorithm whose linearized regret grows sufficiently slowly, as a function of time $t$ and at the radius $|r| = t^p$, when the growth rate of the scores $S_t$ is not too rapid.

**Theorem D.1** (No-linearized-regret implies coverage). *Consider an online learning algorithm that in each round $t = 1, 2, \ldots$ produces $b_t \in \mathbb{R}$. Assume that for all rounds $T \geq 1$ there exists a function $F_T : [0, \infty) \to [0, \infty)$ such that the linearized regret from (6) on the pinball losses $(\ell_t)_{1 \leq t \leq T}$ is bounded as*

$$\text{LinRegret}_T(r) \leq F_T(|r|), \quad \forall r \in \mathbb{R} . \tag{14}$$

*Moreover, assume that the nonconformity scores have a bounded growth: for some $D > 0$, $q \geq 0$, we have $0 \leq S_t \leq Dt^q$ for all $t \geq 1$. Finally, assume that there exists $q < p < 1$ such that[2]*

$$\lim_{t \to \infty} \frac{F_t(t^p)}{t} = 0 . \tag{15}$$

*Then, the algorithm satisfies the long-term coverage guarantee (1).*

*Proof.* Rearranging (14) gives, for all real $r$

$$-\sum_{t=1}^{T} g_t b_t \geq \left( -\sum_{t=1}^{T} g_t \right) r - F_T(|r|) .$$

Hence, by Lemma A.1, we have

$$\left( -\sum_{t=1}^{T} g_t \right) r - F_T(|r|) \leq (1-\alpha) \sum_{t=1}^{T} S_t \leq \frac{(1-\alpha)D(T+1)^{q+1}}{q+1} . \tag{16}$$

Now, set $r = r_T = -\text{sign}\left( \sum_{t=1}^{T} g_t \right) \cdot T^p$, where we define $\text{sign}(0) = 0$. Invoking (15) yields

$$\frac{(1-\alpha)D(T+1)^{q+1}}{q+1} \geq T^p \left| \sum_{t=1}^{T} g_t \right| - F_T(T^p) , \tag{17}$$

which implies that

$$\frac{1}{T} \left| \sum_{t=1}^{T} g_t \right| \leq \frac{F_T(T^p)}{T^{p+1}} + \frac{(1-\alpha)D(T+1)^{q+1}}{(q+1)T^{p+1}} .$$

Now, taking the lim sup of the first term on the r.h.s., we have

$$\limsup_{T \to \infty} \frac{F_T(T^p)}{T^{p+1}} = \limsup_{T \to \infty} \frac{F_T(T^p)}{T} T^{-p} = 0 .$$

---

[2]These two conditions can be summarized into $\max\{\text{LinRegret}_t(t^p), \text{LinRegret}_t(-t^p)\} = o(t)$.

For the second term, given that $q < p$, we have that

$$\limsup_{T \to \infty} \frac{(1 - \alpha)D(T + 1)^{q+1}}{(q + 1)T^{p+1}} = 0 \, .$$

We conclude by the equivalence (5). □

## E. Beyond Symmetric Intervals: Asymmetric and CQR-Style Prediction Sets

Our regret-to-coverage theory does not rely on the prediction set being a *symmetric* interval around a point forecast. Theorem (3.1) by itself does not assume symmetric intervals. Indeed, the proof of Theorem 3.1 uses only two facts: (i) the correspondence between the pinball-loss subgradient $g_t$ and the coverage event $\mathbf{1}\{b_t \geq S_t\}$ in (4), and (ii) the pointwise bound on $-g_t b_t$ of Lemma A.1. Neither references the shape of $\hat{C}_t$. Consequently, any nested family of prediction sets controlled by a scalar threshold, together with a (possibly signed) score whose sublevel sets match the coverage events, can be plugged into the same pipeline: the online learning layer remains the same, and leads to the same results. We illustrate this with two-sided, possibly asymmetric intervals, which cover the conformalized quantile regression (CQR) construction of Romano et al. (2019) as a special case.

Our results were stated for non-negative scores $S_t \geq 0$, but a bounded *signed* score reduces to this case by a simple shift, with *no* change to any coverage event.

**Lemma E.1** (Shift reduction). *Let $(S_t)_{t \geq 1} \subseteq \mathbb{R}$ be a (signed) score sequence with a known lower bound $S_t \geq -M$, $M \geq 0$. Define the shifted score $\tilde{S}_t := S_t + M \geq 0$. If an online algorithm produces thresholds $\tilde{b}_t$ and we set $b_t := \tilde{b}_t - M$, then for every $t$*

$$\mathbf{1}\{b_t \geq S_t\} = \mathbf{1}\left\{\tilde{b}_t \geq \tilde{S}_t\right\}, \qquad \partial\ell^{(1-\alpha)}(\tilde{b}_t, \tilde{S}_t) = \partial\ell^{(1-\alpha)}(b_t, S_t) \, .$$

*Hence Theorem 3.1 applied to $(\tilde{S}_t, \tilde{b}_t)$ yields the identical coverage guarantee for the original $(S_t, b_t)$, with $\sum_t \tilde{S}_t = \sum_t S_t + MT$.*

*Proof.* Both displayed identities follow from $\tilde{b}_t - \tilde{S}_t = b_t - S_t$: the indicator depends only on the sign of $b_t - S_t$, and the pinball subdifferential at $(b, S)$ depends only on $b - S$ (it is $\{-(1 - \alpha)\}$ for $b < S$, $\{\alpha\}$ for $b > S$, and $[-(1 - \alpha), \alpha]$ at $b = S$). Since $\tilde{S}_t \geq 0$, Lemma A.1 and Theorem 3.1 apply verbatim to $(\tilde{S}_t, \tilde{b}_t)$. □

**Two-sided asymmetric intervals.**   Let the base model output a lower and an upper functional $\hat{Y}_t^-$ and $\hat{Y}_t^+$ (e.g., conditional quantile regressors at levels $\alpha/2$ and $1 - \alpha/2$ as in CQR (Romano et al., 2019); taking $\hat{Y}_t^- = \hat{Y}_t^+ = \hat{Y}_t$ recovers the centered case). For thresholds $b_t^-, b_t^+ \in \mathbb{R}$, consider the (possibly asymmetric) prediction set

$$\hat{C}_t = \left[\hat{Y}_t^- - b_t^-, \hat{Y}_t^+ + b_t^+\right] \, ,$$

which we assume non-degenerate, i.e., $\hat{Y}_t^- - b_t^- \leq \hat{Y}_t^+ + b_t^+$. Define the two *one-sided signed scores*

$$S_t^- := \hat{Y}_t^- - Y_t, \qquad S_t^+ := Y_t - \hat{Y}_t^+ \, .$$

A direct computation shows that the lower- and upper-tail misses are exactly the sublevel complements of these scores:

$$\left\{Y_t < \hat{Y}_t^- - b_t^-\right\} = \left\{S_t^- > b_t^-\right\}, \qquad \left\{Y_t > \hat{Y}_t^+ + b_t^+\right\} = \left\{S_t^+ > b_t^+\right\} \, ,$$

and, by non-degeneracy, these two events are mutually exclusive. Therefore, the miscoverage indicator splits exactly:

$$\mathbf{1}\left\{Y_t \notin \hat{C}_t\right\} = \mathbf{1}\left\{S_t^- > b_t^-\right\} + \mathbf{1}\left\{S_t^+ > b_t^+\right\} \, . \tag{18}$$

**Two independent instances.**   Split the budget $\alpha = \alpha^- + \alpha^+$ (the symmetric choice $\alpha^- = \alpha^+ = \alpha/2$ is the usual default) and run *two independent* online instances: instance "$-$" minimizes the linearized regret of the $(1 - \alpha^-)$-pinball loss on the (shifted, via Lemma E.1) sequence $S_t^-$ to output $b_t^-$, and instance "$+$" does the same with $\alpha^+$ on $S_t^+$ to output $b_t^+$. Let $g_t^\bullet = \mathbf{1}\{b_t^\bullet \geq S_t^\bullet\} - (1 - \alpha^\bullet)$ for $\bullet \in \{-, +\}$, and let $F_T^\bullet$ be a valid linearized-regret bound for instance $\bullet$.

**Proposition E.2** (Coverage for asymmetric intervals). *Assume each one-sided score is bounded, $S_t^\bullet \in [-M, M]$, and the interval is non-degenerate for all $t$. Then each instance satisfies, via Theorem 3.1 and Lemma E.1,*

$$(F_T^\bullet)^\star \left( -\sum_{t=1}^T g_t^\bullet \right) \leq (1 - \alpha^\bullet) \sum_{t=1}^T (S_t^\bullet + M), \qquad \bullet \in \{-, +\},$$

*and the overall miscoverage of $\hat{C}_t$ is controlled by the sum of the two one-sided errors:*

$$\left| \frac{1}{T} \sum_{t=1}^T \mathbf{1}\left\{ Y_t \notin \hat{C}_t \right\} - \alpha \right| \leq \text{MisCov}_T^- + \text{MisCov}_T^+, \qquad \text{MisCov}_T^\bullet := \frac{1}{T} \left| \sum_{t=1}^T g_t^\bullet \right|.$$

*In particular, instantiating each instance with UP-OCP yields the bound of Theorem 4.4 with $\alpha$ replaced by $\alpha^-$ and $\alpha^+$, respectively, summed over the two tails.*

*Proof.* The per-instance inequality is Theorem 3.1 applied to the shifted non-negative scores $\tilde{S}_t^\bullet = S_t^\bullet + M$ (Lemma E.1), which leaves the subgradients $g_t^\bullet$ and coverage events unchanged. Averaging the exact decomposition (18) and subtracting $\alpha = \alpha^- + \alpha^+$,

$$(1/T) \sum_{t=1}^T \mathbf{1}\left\{ Y_t \notin \hat{C}_t \right\} - \alpha = \underbrace{\left( \frac{1}{T} \sum_t \mathbf{1}\left\{ S_t^- > b_t^- \right\} - \alpha^- \right)}_{=-(1/T) \sum_t g_t^-} + \underbrace{\left( \frac{1}{T} \sum_t \mathbf{1}\left\{ S_t^+ > b_t^+ \right\} - \alpha^+ \right)}_{=-(1/T) \sum_t g_t^+},$$

and the triangle inequality gives the stated bound. The final claim follows by using the UP-OCP linearized-regret bound of Theorem 4.3 for each $F_T^\bullet$ and invoking Theorem 4.4 per tail. $\square$

Thus asymmetry is handled at the level of the *score*, while the online-learning layer, and all of its parameter-free, polynomial-growth guarantees, can be reused unchanged, once per tail.

# F. The Average Size in Stochastic Settings

While the previous sections established coverage guarantees under adversarial settings, characterizing the *efficiency*, i.e., the size of the prediction sets relative to the optimum, seems to require additional assumptions on the data-generating process. In a fully adversarial setting, a globally optimal interval width is ill-defined, as there is no guarantee that the $Y_t$ share any common behavior over the course of the $T$ rounds.

Therefore, we consider a stochastic setting where the nonconformity scores $S_t$ are sampled i.i.d. from a fixed distribution $\mathcal{D}$. In this setting, the optimal fixed prediction interval of the form $[0, b^\star]$ is achieved when $b^\star$ is any $(1 - \alpha)$-th quantile of the distribution. Our goal is to show that regret-minimizing algorithms do not merely satisfy coverage constraints, but also converge to this optimal radius $b^\star$ at a rate characterized by their regret.

We define the expected loss $L(b)$ with respect to $\mathcal{D}$ as $L(b) := \mathbb{E}_{S \sim \mathcal{D}} \left[ \ell^{(1-\alpha)}(b, S) \right]$. We denote the optimal radius as $b^\star := \arg\min_{b \geq 0} L(b)$; our conditions below will ensure that this is uniquely defined. To rigorously connect the regret $R_T$ to the convergence of the distance $|b_t - b^\star|$, we require mild regularity conditions on $\mathcal{D}$.

**Assumption F.1** (Regularity of $\mathcal{D}$). The nonconformity scores $S_1, S_2, \ldots$ are i.i.d. draws from a distribution $\mathcal{D}$ supported on a compact interval $\mathcal{B} \subseteq [0, \infty)$, such that:

1. $\mathcal{D}$ admits a probability density function (PDF) $\phi$ and a cumulative distribution function (CDF) $\tilde{\Phi}$;
2. The density $\phi$ is uniformly bounded from below, so there is a constant $\kappa > 0$ such that $\phi(b) \geq \kappa, \ \forall b \in \mathcal{B}$.

In the above notation, $b^\star := \tilde{\Phi}^{-1}(1 - \alpha)$.

*Remark* F.2. Clearly, the continuity of $\phi$ is not necessary for the existence of the derivatives of $L$. The expected loss $L(b) = \mathbb{E}[\ell^{(1-\alpha)}(b, S)]$ is a convolution of the continuous pinball loss with the measure $\mathcal{D}$. This acts as a smoothing operator: as long as $\mathcal{D}$ has no point masses, $L(b)$ is continuously differentiable (see, e.g., Joshi et al., 2026). Assumption F.1.2 is imposed primarily to ensure *strong convexity* (positive curvature), which allows us to convert the regret bound into a variance bound.

Under these conditions, we prove that a sublinear regret implies the convergence of the average radius $\bar{b}_T = \frac{1}{T}\sum_{t=1}^{T} b_t$ to the optimal oracle radius $b^\star$.

**Theorem F.3** (Width Convergence via Regret). *Let an online algorithm generate radii $b_1, b_2, \ldots, b_T$ with $b_t \in \mathcal{B}$ for all $t$ such that the expected regret is bounded by $R_T$:*

$$\mathbb{E}\left[\sum_{t=1}^{T} \ell^{(1-\alpha)}(b_t, S_t) - \sum_{t=1}^{T} \ell^{(1-\alpha)}(b^\star, S_t)\right] \leq R_T . \tag{19}$$

*Under Assumption F.1, the squared distance between the average radius $\bar{b}_T = \frac{1}{T}\sum_{t=1}^{T} b_t$ and the true $(1-\alpha)$-quantile is bounded by $\mathbb{E}\left[(\bar{b}_T - b^\star)^2\right] \leq \frac{2}{\kappa} \cdot \frac{R_T}{T}$ .*

*Proof.* The proof proceeds in two parts: a first- and second-order analysis of the expected loss, and applying an online-to-batch conversion.

**First and second-order analysis.** This part reviews well-known results used either explicitly or implicitly in a number of prior works, see, e.g., Gibbs & Candes (2021); Joshi et al. (2026); we include it here only for the sake of being self-contained. Recall the $(1-\alpha)$-pinball loss: $\ell^{(1-\alpha)}(b, S) = \max\{(1-\alpha)(S-b), \alpha(b-S)\}$. The expected loss is given by

$$L(b) = \int_0^b \alpha(b-s)\phi(s)\, ds + \int_b^\infty (1-\alpha)(s-b)\phi(s)\, ds .$$

Differentiating with respect to $b$ yields

$$L'(b) = \alpha\tilde{\Phi}(b) - (1-\alpha)(1 - \tilde{\Phi}(b)) = \tilde{\Phi}(b) - (1-\alpha) .$$

Setting $L'(b) = 0$, we confirm that the minimizer is unique and satisfies $\tilde{\Phi}(b^\star) = 1 - \alpha$. Thus, $b^\star$ is exactly the $(1-\alpha)$-quantile of $\mathcal{D}$. Differentiating $L'(b)$ again, we obtain the Hessian of the expected loss: $L''(b) = \frac{d}{db}(\tilde{\Phi}(b) - 1 + \alpha) = \phi(b)$ . By Assumption F.1, we have $L''(b) = \phi(b) \geq \kappa$ for all $b \in \mathcal{B}$, hence $L$ is $\kappa$-strongly convex on $\mathcal{B}$. In particular,

$$L(b) - L(b^\star) \geq \frac{\kappa}{2}(b - b^\star)^2, \quad \forall b \in \mathcal{B} . \tag{20}$$

**Online-to-batch conversion.** Since the loss function $L(b)$ is convex, Jensen's inequality implies $L(\bar{b}_T) \leq \frac{1}{T}\sum_{t=1}^{T} L(b_t)$. Using standard online-to-batch conversion results (see, e.g., Orabona, 2019, Theorem 3.1), the average regret upper bounds the excess risk:

$$\mathbb{E}[L(\bar{b}_T)] - L(b^\star) \leq \mathbb{E}\left[\frac{1}{T}\sum_{t=1}^{T} L(b_t) - L(b^\star)\right] \leq \frac{R_T}{T} .$$

Combining this with the strong convexity bound in (20), we have

$$\frac{\kappa}{2}\mathbb{E}\left[(\bar{b}_T - b^\star)^2\right] \leq \mathbb{E}[L(\bar{b}_T)] - L(b^\star) \leq \frac{R_T}{T} .$$

Rearranging the terms yields the claim. $\square$

Theorem F.3 implies that *any* algorithm minimizing pinball loss regret *automatically* converges to the statistically efficient oracle width $b^\star$. These results are different in nature from those of Srinivas (2026). Srinivas (2026) frame the problem as direct length optimization, aiming to compete with the best fixed interval length in hindsight. In contrast, our framework specifically leverages the pinball loss as a proper scoring rule (Gneiting & Raftery, 2007).

Recent work by Areces et al. (2025) also investigates the efficiency of online conformal prediction in stochastic settings. Their main result (Theorem 6.1) establishes the convergence of the *last iterate* to the optimal parameter. However, obtaining this guarantee requires a specific decaying schedule for the learning rate ($\eta_t \propto t^{-c}$), which creates an explicit trade-off: faster decay improves efficiency but renders the adversarial coverage bounds vacuous (Areces et al., 2025, Section 6). In contrast, our width convergence result (Theorem F.3) is more general, showing that efficiency is an automatic consequence of regret minimization for *any* algorithm, regardless of its specific update rule. Furthermore, because UP-OCP is parameter-free, it naturally achieves this efficiency through low regret without requiring manual step-size tuning or sacrificing robust coverage guarantees under adversarial distribution shifts.

# G. Proof of Theorem 4.3

This appendix details the reduction used in Section 4. The main point is that, once we express the pinball subgradients as an asymmetric coin sequence, our two-stock conformal market in Definition 4.1 is exactly a two-asset encoding of an asymmetric coin-betting game. Standard coin-betting duality then translates wealth guarantees into linearized-regret guarantees, which in turn control the pinball-loss regret.

## G.1. From pinball subgradients to an asymmetric coin

Recall that for the pinball loss $\ell_t(b) = \ell^{(1-\alpha)}(b, S_t)$ we use the subgradient $g_t = \mathbf{1}\{b_t \geq S_t\} - (1 - \alpha)$, as in (4). Hence, with our tie-breaking convention at $b_t = S_t$, we have $g_t \in \{-(1-\alpha), \alpha\}$.

It is convenient to flip signs and work with the bounded outcome of a coin:

$$c_t := -g_t \in \{-\alpha, 1 - \alpha\} . \tag{21}$$

Under our tie-breaking, $c_t$ takes only the two values: $c_t = 1 - \alpha$ if $b_t < S_t$ (miscoverage) and $c_t = -\alpha$ if $b_t \geq S_t$ (coverage).

Consider now the following asymmetric coin-betting game. At each round $t$, a bettor chooses a *signed betting fraction* $\beta_t$ and then the outcome $c_t$ is revealed. We define the wealth process as

$$W_t := W_{t-1}(1 + \beta_t c_t), \qquad W_0 := 1 . \tag{22}$$

To ensure *no bankruptcy* for all outcomes $c_t \in [-\alpha, 1 - \alpha]$, the betting fraction must satisfy

$$\beta_t \in \left[ -\frac{1}{1 - \alpha}, \frac{1}{\alpha} \right] . \tag{23}$$

Indeed, the condition $1 + \beta_t c_t \geq 0$ for all $c_t \in \{-\alpha, 1 - \alpha\}$ is equivalent to simultaneously requiring $1 - \alpha\beta_t \geq 0$ and $1 + (1 - \alpha)\beta_t \geq 0$, which is exactly (23).

We also define the *bet amount*

$$b_t := \beta_t W_{t-1} . \tag{24}$$

Plugging (24) into (22) yields the additive form

$$W_t = W_{t-1} + c_t b_t = W_{t-1} - g_t b_t . \tag{25}$$

## G.2. The conformal market is a two-stock encoding of coin betting

We now show that the conformal market in Definition 4.1 is precisely a two-asset representation of the asymmetric coin game above.

Using $c_t = -g_t$, Definition 4.1 can be rewritten as

$$w_{t,1} = \frac{c_t + \alpha}{\alpha}, \qquad w_{t,2} = \frac{(1 - \alpha) - c_t}{1 - \alpha} . \tag{26}$$

These returns are nonnegative for every $c_t \in [-\alpha, 1 - \alpha]$. Moreover, when $c_t = 1 - \alpha$ (miscoverage), we have $w_{t,1} = 1/\alpha$ and $w_{t,2} = 0$, while when $c_t = -\alpha$ (coverage), we have $w_{t,1} = 0$ and $w_{t,2} = 1/(1 - \alpha)$.

Given a portfolio weight $\lambda_t \in [0, 1]$ (fraction of capital invested in Stock 1), the wealth update in Definition 4.2 is $W_t = W_{t-1}(\lambda_t w_{t,1} + (1 - \lambda_t)w_{t,2})$. The next lemma shows that this is exactly (22) under an affine reparameterization of $\lambda_t$.

**Lemma G.1** (Portfolio-to-coin equivalence). *Fix $\alpha \in (0, 1)$ and let $c_t \in [-\alpha, 1 - \alpha]$. Define the two-stock returns by (26). For any $\lambda_t \in [0, 1]$, define*

$$\beta_t := -\frac{1}{1 - \alpha} + \frac{\lambda_t}{\alpha(1 - \alpha)} = \frac{\lambda_t - \alpha}{\alpha(1 - \alpha)} . \tag{27}$$

*Then,*

$$\lambda_t w_{t,1} + (1 - \lambda_t) w_{t,2} = 1 + \beta_t c_t . \tag{28}$$

*Consequently, the wealth recursion of Definition 4.2 is identical to the coin-betting recursion (22), and the bet amount* $b_t = \beta_t W_{t-1}$ *equals the mapping (11).*

*Proof.* Using (26),

$$
\begin{aligned}
\lambda_t w_{t,1} + (1 - \lambda_t) w_{t,2} &= \lambda_t \left( \frac{c_t + \alpha}{\alpha} \right) + (1 - \lambda_t) \left( \frac{(1 - \alpha) - c_t}{1 - \alpha} \right) \\
&= \lambda_t \left( 1 + \frac{c_t}{\alpha} \right) + (1 - \lambda_t) \left( 1 - \frac{c_t}{1 - \alpha} \right) \\
&= 1 + c_t \left( \frac{\lambda_t}{\alpha} - \frac{1 - \lambda_t}{1 - \alpha} \right) .
\end{aligned}
$$

The coefficient of $c_t$ simplifies as $\frac{\lambda_t}{\alpha} - \frac{1-\lambda_t}{1-\alpha} = -\frac{1}{1-\alpha} + \frac{\lambda_t}{\alpha(1-\alpha)} = \beta_t$, which proves (28). Substituting into the portfolio recursion gives $W_t = W_{t-1}(1 + \beta_t c_t)$, and $b_t = \beta_t W_{t-1}$ is exactly (11). □

Two immediate consequences are worth recording. First, since $\lambda_t \in [0, 1]$, the induced $\beta_t$ in (27) always lies in the safe interval (23). Second, the one-dimensional action $b_t$ in (11) is simply the coin-betting bet amount associated with the portfolio choice $\lambda_t$.

### G.3. From wealth lower bounds to linearized-regret bounds

We now connect wealth to the linearized regret on the pinball loss. Recall that linearized regret is $\mathrm{LinRegret}_T(u) = \sum_{t=1}^{T} g_t(b_t - u)$. Using $c_t = -g_t$ from (21), this can be rewritten as

$$\mathrm{LinRegret}_T(u) = \sum_{t=1}^{T} c_t(u - b_t) . \tag{29}$$

Moreover, by telescoping (25), we have

$$W_T = 1 + \sum_{t=1}^{T} c_t b_t . \tag{30}$$

The standard coin-betting duality is that a *lower bound* on the achievable wealth as a function of the cumulative outcome sum implies an *upper bound* on linearized regret via Fenchel conjugacy (see, e.g., Orabona, 2019).

**Lemma G.2** (Wealth lower bound ⇒ linearized regret bound). *Let* $\Psi_T : \mathbb{R} \to (-\infty, +\infty]$ *be a function. Assume that an algorithm produces* $(b_t)_{t=1}^{T}$ *and wealth* $W_T$ *satisfying*

$$W_T - 1 \geq \Psi_T \left( \sum_{t=1}^{T} c_t \right) \tag{31}$$

*for every sequence* $(c_t)_{t=1}^{T} \subseteq [-\alpha, 1 - \alpha]$*. Then, for every comparator* $u \in \mathbb{R}$*, its linearized regret satisfies*

$$\mathrm{LinRegret}_T(u) \leq \Psi_T^{\star}(u), \tag{32}$$

*where* $\Psi_T^{\star}$ *is the Fenchel conjugate of* $\Psi_T$*.*

*Proof.* Let $\theta_T := \sum_{t=1}^{T} c_t$. By Fenchel-Young duality, for every $u$ we have $u\theta_T \leq \Psi_T(\theta_T) + \Psi_T^{\star}(u)$. By the assumption (31) and the identity (30), we have $\sum_{t=1}^{T} c_t b_t = W_T - 1 \geq \Psi_T(\theta_T)$. Therefore,

$$\sum_{t=1}^{T} c_t(u - b_t) = u\theta_T - \sum_{t=1}^{T} c_t b_t \leq u\theta_T - \Psi_T(\theta_T) \leq \Psi_T^{\star}(u),$$

where in the last equality we used Fenchel-Young inequality. This is exactly (32) using (29). □

Lemma G.2 is the precise mathematical sense in which "maximizing wealth" (in the coin-betting/portfolio game) corresponds to "minimizing linearized regret" (in online convex optimization). The only remaining ingredient is to identify an explicit lower bound $\Psi_T$ for $W_T - 1$ and calculate (an upper bound to) its Fenchel conjugate.

## G.4. Instantiating the potential via the best constant rebalanced portfolio

We now connect the portfolio regret guarantee to a wealth lower bound of the form (31).

Let $W_T^\star$ denote the wealth of the best constant rebalanced portfolio in the conformal market, i.e., $W_T^\star = \max_{\lambda \in [0,1]} \prod_{t=1}^T (\lambda w_{t,1} + (1 - \lambda) w_{t,2})$. By Lemma G.1, this is equivalently the best constant betting fraction in the asymmetric coin game: $W_T^\star = \max_{\beta \in [-1/(1-\alpha), 1/\alpha]} \prod_{t=1}^T (1 + \beta c_t)$.

When we use the extreme subgradient convention (so $c_t \in \{-\alpha, 1 - \alpha\}$), $W_T^\star$ depends on the data only through the number of miscoverages. Let $M_T := \sum_{t=1}^T \mathbf{1}\{c_t = 1 - \alpha\}$ and $C_T := T - M_T$. Then, for a fixed $\lambda \in [0, 1]$, we have

$$W_T(\lambda) = \left(\frac{\lambda}{\alpha}\right)^{M_T} \left(\frac{1 - \lambda}{1 - \alpha}\right)^{C_T} .$$

Maximizing over $\lambda$ yields $\lambda^\star = M_T/T$, and hence

$$W_T^\star = \left(\frac{M_T}{\alpha T}\right)^{M_T} \left(\frac{C_T}{(1 - \alpha)T}\right)^{C_T} = \exp\left(T \cdot \mathrm{KL}\left(\frac{M_T}{T} \middle\| \alpha\right)\right), \tag{33}$$

where $\mathrm{KL}(p \| q) = p \log \frac{p}{q} + (1 - p) \log \frac{1-p}{1-q}$ is the Bernoulli KL divergence.

Moreover, $M_T$ can be expressed directly in terms of $\theta_T = \sum_{t=1}^T c_t$: since $c_t$ equals $1 - \alpha$ on miscoverage and $-\alpha$ on coverage, we have $\theta_T = (1 - \alpha)M_T - \alpha C_T = M_T - \alpha T$, hence $M_T = \alpha T + \theta_T$. Substituting into (33) gives an explicit function of $\theta_T$:

$$W_T^\star = \exp\left(T \cdot \mathrm{KL}\left(\alpha + \frac{\theta_T}{T} \middle\| \alpha\right)\right) . \tag{34}$$

From the assumption that the portfolio algorithm guarantees $\log W_T \geq \log W_T^\star - \mathcal{R}_T$, exponentiating yields

$$W_T \geq \exp\left(-\mathcal{R}_T\right) W_T^\star = \exp\left(-\mathcal{R}_T\right) \exp\left(T \cdot \mathrm{KL}\left(\alpha + \frac{\theta_T}{T} \middle\| \alpha\right)\right) := \Psi_T(\theta_T) + 1. \tag{35}$$

Using Lemma G.2, we need to calculate the Fenchel conjugate of $\Psi_T$. Unfortunately, it does not have a closed-form expression. Hence, we use Lemma H.1 to obtain an easier lower bound:

$$\exp\left(T \cdot \mathrm{KL}\left(\alpha + \frac{\theta_T}{T} \middle\| \alpha\right)\right) \geq \exp\left(T \cdot \frac{\theta_T^2/T^2}{2\alpha(1 - \alpha) + 2/3|\theta_T|/T}\right) \geq \min\left\{\exp\left(\frac{\theta_T^2}{4\alpha(1 - \alpha)}\right), \exp\left(\frac{3}{4}|\theta_T|\right)\right\} . \tag{36}$$

Now, observe that if $f(x) = \min\{h_1(x), h_2(x)\}$, then we have

$$f^\star(y) = \sup_x xy - f(x) = \sup_x xy - \min\{h_1(x), h_2(x)\} = \sup_x \max\{xy - h_1(x), xy - h_2(x)\}$$
$$\leq \max\{\sup_x xy - h_1(x), \sup_x xy - h_2(x)\} = \max\{h_1^\star(y), h_2^\star(y)\} .$$

Hence, it suffices to find the Fenchel conjugates (or upper bounds) of the two functions in the min in the right-hand side of (36). This follows immediately from Orabona (2019, Example 6.18, Lemma 6.24, and Theorems C.3 and C.4):

$$\Psi_T^\star(u) \leq \max\left\{|u|\sqrt{2T\alpha(1 - \alpha)\ln(2T\alpha(1 - \alpha)u^2 \exp(\mathcal{R}_T) + 1)}, \frac{4}{3}|u|\left(\ln \frac{4|u|\exp(\mathcal{R}_T)}{3} - 1\right)\right\} .$$

Using the regret of universal portfolio of $\frac{1}{2}\ln(\pi(T + 1))$ and overapproximating completes the proof.

**Sharpness of the reduction.** The portfolio-to-regret step here is an *equivalence*: the passage between the wealth of a coin-betting/portfolio algorithm and the linearized regret of the induced online linear optimization algorithm is via the regret–reward duality, which is a two-sided correspondence (Orabona, 2019, Theorem 10.6). Consequently, Lemmas G.1 and G.2 introduce no slackness, and the universal-portfolio regret $(1/2)\ln(\pi(T+1))$ is itself minimax optimal up to additive constants (Cover & Ordentlich, 2002). The only looseness occurs in the last step, the Fenchel-conjugate inversion of the exact potential $\exp\left(T\,\mathrm{KL}(\alpha + \theta_T/T\|\alpha)\right)$, whose conjugate has no closed form: we first relax the KL by the Bernstein-type bound of Lemma H.1 and then upper bound the conjugate of a minimum by a maximum of conjugates in (36). Both relaxations are of second order; they only affect constants and lower-order terms, so the two leading regimes of $F_T$, i.e., the $\sqrt{T\alpha(1-\alpha)\ln(\cdot)}$ regime and the logarithmic regime active as $\alpha \to 0$, are preserved.

# H. Regret Guarantee of UP-OCP versus KT

The proof of the regret guarantee of Podkopaev et al. (2024) follows exactly the same steps as the proof for UP-OCP; moreover, the KT approach can also be written as a universal portfolio algorithm, with the same prior. The only difference is the transformation of the subgradients into two stocks, which changes the wealth of the best constant rebalanced portfolio, $W_T^\star$.

The transformation for KT is the following one:

$$w_{t,1} = 1 - g_t, \quad w_{t,2} = 1 + g_t.$$

Given that $g_t \in \{\alpha - 1, \alpha\} \subset [-1, 1]$, we have $w_{t,1} \geq 0$ and $w_{t,2} \geq 0$. In the coin-betting view, this corresponds to the wealth process

$$W_t = W_{t-1}(1 - g_t\beta_t),$$

where we constrain $\beta_t \in [-1, 1]$. This means that $W_t^\star = \max_{\beta \in [-1,1]} \prod_{t=1}^T (1 - g_t\beta)$. Contrast this with the one we derived for UP-OCP:

$$\max_{\beta \in [-1/(1-\alpha), 1/\alpha]} \prod_{t=1}^T (1 - \beta g_t).$$

Given that $[-1, 1] \subset [-1/(1-\alpha), 1/\alpha]$, the wealth of the best constant rebalanced portfolio in the UP-OCP reduction is always at least that of the KT one, but it is potentially much larger.

## H.1. Missing proofs in Section 4

**Lemma H.1.** *Let $p, q \in [0, 1]$, then*

$$\mathrm{KL}(p\|q) = p\ln\frac{p}{q} + (1-p)\ln\frac{1-p}{1-q} \geq \frac{(p-q)^2}{2q(1-q) + \frac{2}{3}|p-q|}.$$

*Proof of Lemma H.1.* Define $h(x) = (1 + x)\ln(1 + x) - x$ for $x > -1$ and extend it by continuity in $x = -1$ with $h(-1) := 1$. Also, define $\Delta = p - q \in [-1, 1]$.

We have that

$$p\ln\frac{p}{q} = (q + \Delta)\ln\left(1 + \frac{\Delta}{q}\right) = q\left(h\left(\frac{\Delta}{q}\right) + \frac{\Delta}{q}\right) = q\,h\left(\frac{\Delta}{q}\right) + \Delta.$$

Similarly, we have

$$(1-p)\ln\frac{1-p}{1-q} = (1-q-\Delta)\ln\left(1 - \frac{\Delta}{1-q}\right) = (1-q)\left(1 - \frac{\Delta}{1-q}\right)\ln\left(1 - \frac{\Delta}{1-q}\right) = (1-q)\,h\left(-\frac{\Delta}{1-q}\right) - \Delta.$$

Hence, overall we have

$$\mathrm{KL}(p\|q) = q\,h\left(\frac{\Delta}{q}\right) + (1-q)\,h\left(-\frac{\Delta}{1-q}\right).$$

Observe that $\frac{\Delta}{q} \geq -1$ and $-\frac{\Delta}{1-q} \geq -1$. Hence, we use the elementary inequality $h(x) \geq \frac{x^2}{2 + \frac{2}{3}|x|}$ for $x \geq -1$, to have

$$\mathrm{KL}(p\|q) \geq \Delta^2\left(\frac{1}{2q + \frac{2}{3}|\Delta|} + \frac{1}{2(1-q) + \frac{2}{3}|\Delta|}\right) = \Delta^2\frac{2 + \frac{4}{3}|\Delta|}{4q(1-q) + \frac{4}{3}|\Delta| + \frac{4}{9}\Delta^2} \geq \frac{\Delta^2}{2q(1-q) + \frac{2}{3}|\Delta|}.$$

Using the value of $\Delta$ finishes the proof. $\qquad\square$

*Proof of Theorem 4.4.* Let $k := |\{t \in [T] : g_t < 0\}|$ and $\hat{p} := k/T$. Since $g_t = \alpha$ on coverage rounds and $g_t = -(1-\alpha)$ on miscoverage rounds,

$$\sum_{t=1}^{T} g_t = \alpha(T-k) - (1-\alpha)k = \alpha T - k, \qquad \frac{1}{T}\sum_{t=1}^{T} g_t = \alpha - \hat{p} .$$

**Best-constant wealth equals an exact KL term.** For a constant betting strategy $\lambda \in [0,1]$, the wealth is

$$W_T(\lambda) = \left(\frac{\lambda}{\alpha}\right)^k \left(\frac{1-\lambda}{1-\alpha}\right)^{T-k} .$$

The maximizer is $\lambda^\star = \hat{p}$, and substituting yields

$$\ln W_T^\star = k \ln \frac{\hat{p}}{\alpha} + (T-k) \ln \frac{1-\hat{p}}{1-\alpha}$$

$$= T\left(\hat{p}\ln\frac{\hat{p}}{\alpha} + (1-\hat{p})\ln\frac{1-\hat{p}}{1-\alpha}\right) = T \cdot \mathrm{KL}(\hat{p}\|\alpha) . \tag{37}$$

For two assets with Jeffreys prior, the Universal Portfolio wealth is the $\mathrm{Beta}(1/2, 1/2)$ mixture, whose regret is known (Cover & Ordentlich, 2002):

$$\ln W_T \geq \ln W_T^\star - \frac{1}{2}\ln(\pi(T+1)) . \tag{38}$$

Moreover, by Lemma A.1 and the assumption that $S_t \leq Dt^q$, we have that

$$W_T = 1 - \sum_{t=1}^{T} b_t g_t \leq 1 + \frac{(1-\alpha)D}{q+1}(T+1)^{q+1} .$$

Combining (37) and (38) gives

$$\mathrm{KL}(\hat{p}\|\alpha) \leq \varepsilon_T := \frac{1}{T}\left[\ln\left(1 + \frac{(1-\alpha)D(T+1)^{q+1}}{q+1}\right) + \frac{1}{2}\ln(\pi(T+1))\right] . \tag{39}$$

**Explicit inversion from KL to miscoverage deviation.** Let $\delta := |\hat{p} - \alpha|$. From Lemma H.1, we have

$$\mathrm{KL}(a\|b) \geq \frac{(a-b)^2}{2b(1-b) + \frac{2}{3}|a-b|} . \tag{40}$$

Applying (40) with $a = \hat{p}$ and $b = \alpha$ and using (39) yields

$$\varepsilon_T \geq \frac{\delta^2}{2\alpha(1-\alpha) + \frac{2}{3}\delta},$$

equivalently

$$\delta^2 - \frac{2\varepsilon_T}{3}\delta - 2\alpha(1-\alpha)\varepsilon_T \leq 0 .$$

Solving this quadratic inequality for the nonnegative root gives

$$\delta \leq \frac{\varepsilon_T}{3} + \sqrt{2\alpha(1-\alpha)\varepsilon_T + \frac{\varepsilon_T^2}{9}} \leq \varepsilon_T + \sqrt{2\alpha(1-\alpha)\varepsilon_T} .$$

Since $\delta = |\hat{p} - \alpha| = \left|\frac{1}{T}\sum_{t=1}^{T} g_t\right|$, this proves the stated bound. $\qquad\square$

## H.2. Radius Clipping

We slightly abuse notation here by letting $\tilde{b}_t$ be the *raw* output of the wealth mapping (11). In Algorithm 1, the prediction radius is truncated to be non-negative: $b_t \leftarrow \max(0, \tilde{b}_t)$. Here we show that this operation preserves the validity of the regret guarantees.

First, we examine the consistency of the gradients used for the wealth update. The algorithm updates the wealth based on whether coverage was attained:

- Case 1: If $\tilde{b}_t \geq 0$, then trivially the clipping step is not active and hence $b_t = \tilde{b}_t$. The subgradient $g_t$ is computed at the same point as in the unclipped case.

- Case 2: If $\tilde{b}_t < 0$, then $\tilde{b}_t < S_t$ (since $S_t \geq 0$) and hence the algorithm receives the subgradient $g_t = -(1 - \alpha)$. In terms of the clipped value $b_t = 0$, we also have $b_t \leq S_t$, yielding the same gradient $g_t = -(1 - \alpha)$.

We conclude that the clipping does not alter the subgradient sequence seen by the algorithm in the standard case.

Now, we compare the linearized regret terms. Let $\mathrm{LinRegret}_t(u) = g_t(b_t - u)$ and $\widetilde{\mathrm{LinRegret}}_t(u) = g_t(\tilde{b}_t - u)$. The difference is:

$$\mathrm{LinRegret}_t(u) - \widetilde{\mathrm{LinRegret}}_t(u) = g_t(b_t - u) - g_t(\tilde{b}_t - u) = g_t(b_t - \tilde{b}_t) \, .$$

Again, we do a case analysis:

- If $\tilde{b}_t \geq 0$, then $b_t = \tilde{b}_t$, so the difference is 0.

- If $\tilde{b}_t < 0$, then $b_t = 0$ and $g_t = -(1 - \alpha)$. The difference is:

$$-(1 - \alpha)(0 - \tilde{b}_t) = (1 - \alpha)\tilde{b}_t < 0 \, .$$

In all cases, $g_t(b_t - u) \leq \tilde{g}_t(\tilde{b}_t - u)$. Summing over $t$ confirms that the linearized regret of the truncated algorithm satisfies the same bound as the original wealth process.

# I. Closed-Form UP Update

**Theorem I.1** (Closed-Form Update (Jeffreys prior))**.** *For the universal portfolio update* (10) *with the Jeffreys prior on* $\Delta = [0, 1]$ *(equivalently a* $\mathrm{Beta}(1/2, 1/2)$ *prior in the two-asset case), the weights admit the closed-form update*

$$\lambda_t = \frac{1}{t} \left( \sum_{i=1}^{t-1} \mathbf{1} \left\{ g_i = -(1 - \alpha) \right\} + \frac{1}{2} \right) \, . \tag{41}$$

*Proof.* In the conformal market of Definition 4.1, at round $t$ the two synthetic stocks have gross returns

$$(w_{t,1}, w_{t,2}) = \left( 1 - \frac{g_t}{\alpha}, \, 1 + \frac{g_t}{1 - \alpha} \right) \, .$$

A constant-rebalanced portfolio that invests fraction $\lambda \in [0, 1]$ in stock 1 and $1 - \lambda$ in stock 2 achieves one-step gross return $\lambda \, w_{t,1} + (1 - \lambda) \, w_{t,2}$. Hence, the universal portfolio weight is

$$\lambda_{T+1} = \frac{1}{K} \int_0^1 \lambda \prod_{t=1}^T \left[ \lambda \left( 1 - \frac{g_t}{\alpha} \right) + (1 - \lambda) \left( 1 + \frac{g_t}{1 - \alpha} \right) \right] \mu(\lambda) \, d\lambda,$$

where

$$K = \int_0^1 \prod_{t=1}^T \left[ \lambda \left( 1 - \frac{g_t}{\alpha} \right) + (1 - \lambda) \left( 1 + \frac{g_t}{1 - \alpha} \right) \right] \mu(\lambda) \, d\lambda \, .$$

Now use that $g_t \in \{-(1-\alpha), \alpha\}$. Let $a$ be the number of rounds with $g_t = -(1-\alpha)$ (miscoverage), so $T - a$ is the number of rounds with $g_t = \alpha$. For a miscoverage round $g_t = -(1-\alpha)$, we have $1 - g_t/\alpha = 1 + (1-\alpha)/\alpha = 1/\alpha$ and $1 + g_t/(1-\alpha) = 1 - 1 = 0$, hence

$$\lambda \left(1 - \frac{g_t}{\alpha}\right) + (1 - \lambda) \left(1 + \frac{g_t}{1 - \alpha}\right) = \frac{\lambda}{\alpha} \ .$$

For a coverage round $g_t = \alpha$, we have $1 - g_t/\alpha = 0$ and $1 + g_t/(1-\alpha) = 1 + \alpha/(1-\alpha) = 1/(1-\alpha)$, hence

$$\lambda \left(1 - \frac{g_t}{\alpha}\right) + (1 - \lambda) \left(1 + \frac{g_t}{1 - \alpha}\right) = \frac{1 - \lambda}{1 - \alpha} \ .$$

Therefore,

$$\prod_{t=1}^{T} \left[\lambda \left(1 - \frac{g_t}{\alpha}\right) + (1 - \lambda) \left(1 + \frac{g_t}{1 - \alpha}\right)\right] = \left(\frac{\lambda}{\alpha}\right)^a \left(\frac{1 - \lambda}{1 - \alpha}\right)^{T-a} \ .$$

Plugging this into numerator and denominator gives

$$
\begin{aligned}
\lambda_{T+1} &= \frac{\int_0^1 \lambda \left(\frac{\lambda}{\alpha}\right)^a \left(\frac{1-\lambda}{1-\alpha}\right)^{T-a} \mu(\lambda) \, \mathrm{d}\lambda}{\int_0^1 \left(\frac{\lambda}{\alpha}\right)^a \left(\frac{1-\lambda}{1-\alpha}\right)^{T-a} \mu(\lambda) \, \mathrm{d}\lambda} \\
&= \frac{\int_0^1 \lambda^{a+1}(1 - \lambda)^{T-a} \mu(\lambda) \, \mathrm{d}\lambda}{\int_0^1 \lambda^a (1 - \lambda)^{T-a} \mu(\lambda) \, \mathrm{d}\lambda} \ ,
\end{aligned}
$$

since the constant factor $\alpha^{-a}(1 - \alpha)^{a-T}$ cancels.

Now, specialize it to the $\mathrm{Dirichlet}(1/2, 1/2)$ prior, whose marginal density is $\mu(\lambda) = \frac{\Gamma(1)}{\Gamma(1/2)^2} \lambda^{-1/2}(1 - \lambda)^{-1/2}$ on $(0, 1)$. Define, for $p, q > -1/2$,

$$I(p, q) = \int_0^1 \lambda^p (1 - \lambda)^q \mu(\lambda) \, \mathrm{d}\lambda \ .$$

Then,

$$I(p, q) = \frac{\Gamma(1)}{\Gamma(1/2)^2} \int_0^1 \lambda^{p-1/2}(1 - \lambda)^{q-1/2} \, \mathrm{d}\lambda = \frac{\Gamma(1)}{\Gamma(1/2)^2} \cdot \frac{\Gamma(p + 1/2)\Gamma(q + 1/2)}{\Gamma(p + q + 1)} \ .$$

Using $\Gamma(u + 1) = u\Gamma(u)$,

$$\frac{I(p + 1, q)}{I(p, q)} = \frac{\Gamma(p + 3/2)}{\Gamma(p + 1/2)} \cdot \frac{\Gamma(p + q + 1)}{\Gamma(p + q + 2)} = \frac{p + 1/2}{p + q + 1} \ .$$

Finally, taking $p = a$ and $q = T - a$ yields

$$\lambda_{T+1} = \frac{I(a + 1, T - a)}{I(a, T - a)} = \frac{a + 1/2}{T + 1} \ .$$

For a uniform distribution as the prior, a similar derivation yields $\lambda_{T+1} = \frac{a+1}{T+2}$ . $\qquad \square$

This result shows that for conformal prediction, the optimal Universal Portfolio strategy is computationally efficient, requiring only $\mathcal{O}(1)$ time per step. Alternatively, using a uniform prior yields the Laplace rule

$$\frac{\sum_{i=1}^{t-1} \mathbf{1}\{g_i = -(1 - \alpha)\} + 1}{t + 1},$$

which is known to have slightly higher regret (Orabona, 2019, Theorem 13.1).

# J. Baseline Update Rules

## J.1. Krichevsky-Trofimov

The Krichevsky-Trofimov (KT) bettor is a parameter-free approach for adaptive conformal inference that addresses the sensitivity of traditional methods to learning rate tuning (Podkopaev et al., 2024). By framing the selection of the conformal radius $b_t$ as a coin-betting game, the algorithm avoids the need for a manually specified learning rate. In this framework, the algorithm starts with an initial wealth $W_0 = 1$ and places bets on the outcome of coins $c_t \in [-1, 1]$, which are defined as the negative subgradients of the pinball loss: $c_t = -g_t$. The radius at each step $t$ is determined by the betting fraction $\beta_t$ and the current wealth:

$$b_t = \beta_t W_{t-1} . \tag{42}$$

The wealth is updated recursively based on the bet's success:

$$W_t = W_{t-1} + b_t c_t = W_{t-1} - g_t b_t . \tag{43}$$

The KT estimator provides a practical betting scheme that adapts to the observed sequence of subgradients by updating the betting fraction as follows:

$$\beta_{t+1} = \frac{t}{t+1}\beta_t - \frac{1}{t+1}g_t . \tag{44}$$

This strategy is proven to control long-term miscoverage frequency at the nominal level $\alpha$ provided the nonconformity scores are bounded (Podkopaev et al., 2024).

---

**Algorithm 2** KT-based Adaptive Conformal Predictor

---

**Initialize:** $\alpha \in (0, 1)$, $W_0 = 1$, $\beta_1 = 0$, $b_1 = 0$.
**for** $t = 1, 2, \ldots$ **do**
  Produce a forecast $\hat{Y}_t$ and output a set: $\hat{C}_t = [\hat{Y}_t - b_t, \hat{Y}_t + b_t]$
  Observe $Y_t$ and compute error: $S_t = \left| Y_t - \hat{Y}_t \right|$
  Compute $g_t \in \partial\ell^{(1-\alpha)}(b, S_t)|_{b=b_t}$ as per (4);
  Set $W_t = W_{t-1} - g_t b_t$
  Set $\beta_{t+1} = \frac{t}{t+1}\beta_t - \frac{1}{t+1}g_t$
  Set $b_{t+1} = \beta_{t+1}W_t$
**end for**

---

## J.2. Dynamically-tuned Adaptive Conformal Inference

Dynamically-tuned Adaptive Conformal Inference (DtACI) is an extension of ACI designed to eliminate the sensitivity to the fixed step-size parameter $\gamma$, which governs the adaptation rate to distribution shifts (Gibbs & Candès, 2024). Instead of relying on a single $\gamma$, DtACI maintains a set of expert ACI instances running in parallel, each using a different step size $\gamma_k$ from a candidate grid. At each time step $t$, the algorithm aggregates the predictions of these experts using an online learning procedure (e.g., exponentially weighted average) to produce a robust conformal radius $b_t$. The aggregation relies on the empirical quantiles of the current score, defined as $\beta_t := \hat{F}_{t-1}(S_t)$, where $\hat{F}_{t-1}$ is the empirical distribution of the past nonconformity scores. The experts are weighted based on their performance with respect to the pinball loss evaluated at these levels. Specifically, the loss for an expert proposing target level $\alpha_t^i$ is given by:

$$\ell(\beta_t, \alpha_t^i) = \begin{cases} \alpha(\beta_t - \alpha_t^i) & \text{if } \beta_t \geq \alpha_t^i \\ (\alpha - 1)(\beta_t - \alpha_t^i) & \text{if } \beta_t < \alpha_t^i. \end{cases}$$

---

**Algorithm 3** Dynamically-tuned Adaptive Conformal Inference

---

**Input:** Observed values $\{\beta_t\}_{1 \le t \le T}$, set of candidate $\gamma$ values $\{\gamma_i\}_{1 \le i \le k}$, starting points $\{\alpha_1^i\}_{1 \le i \le k}$, and parameters $\sigma$ and $\eta$.
**Initialize:** $w_1^i = 1, 1 \le i \le k$.
**for** $t = 1, 2, \dots, T$ **do**
   Define the probabilities $p_t^i = w_t^i / \sum_{1 \le j \le k} w_t^j, \forall 1 \le i \le k$
   Output $\overline{\alpha}_t = \sum_{1 \le i \le k} p_t^i \alpha_t^i$
   $\overline{w}_t^i = w_t^i \exp(-\eta \ell(\beta_t, \alpha_t^i)), \forall 1 \le i \le k$
   $\overline{W}_t = \sum_{1 \le i \le k} \overline{w}_t^i$
   $w_{t+1}^i = (1 - \sigma)\overline{w}_t^i + \overline{W}_t \sigma / k$
   $err_t^i = \mathbf{1} \left\{ Y_t \notin \hat{C}_t(\alpha_t^i) \right\}, \forall 1 \le i \le k$
   $err_t = \mathbf{1} \left\{ Y_t \notin \hat{C}_t(\overline{\alpha}_t) \right\}$
   $\alpha_{t+1}^i = \alpha_t^i + \gamma_i(\alpha - err_t^i), \forall 1 \le i \le k$
**end for**

---

The default hyperparameters for DtACI follow the experimental setup detailed in Section 4 of Gibbs & Candès (2024). The candidate step sizes are set to be on a logarithmic grid $\Gamma = \{0.001 \cdot 2^k\}_{k=0}^7$, ranging from 0.001 to 0.128, designed to cover a spectrum of regimes from stable to highly reactive. The fixed share parameter is set to $\sigma = 0.001$, and the learning rate $\eta$ for the expert algorithm is chosen to be $e$. The starting points $\{\alpha_1^i\}_{1 \le i \le k}$ are all initialized to the nominal miscoverage level $\alpha$.

## J.3. Scale-Free Online Gradient Descent

As mentioned in the main text, standard Online Gradient Descent (OGD) is sensitive to the scale of the nonconformity scores, often requiring careful tuning of the learning rate to ensure reasonable performance. To address this, Bhatnagar et al. (2023) adapted Scale-Free Online Gradient Descent (SF-OGD) (Orabona & Pál, 2018) for conformal prediction. SF-OGD dynamically adjusts the step size by normalizing the current gradient by the Euclidean norm of the history of past gradients. This normalization makes the algorithm robust to the magnitude of the scores without requiring prior knowledge of their bounds. The update rule for the conformal radius $b_t$ is given by:

$$b_{t+1} = b_t - \eta \frac{g_t}{\sqrt{\varepsilon + \sum_{i=1}^t g_i^2}}, \tag{45}$$

where $g_t$ is the subgradient of the pinball loss, $\eta$ is a scalar multiplier controlling the learning rate, and $\varepsilon$ is a small constant (e.g., $10^{-6}$) to ensure numerical stability.

---

**Algorithm 4** SF-OGD Adaptive Conformal Predictor

---

**Input:** Target miscoverage $\alpha \in (0, 1)$, learning rate $\eta > 0$.
**Initialize:** Radius $b_1 = 0$, sum of squared gradients $G_0 = 0$, $\varepsilon = 10^{-6}$.
**for** $t = 1, 2, \dots$ **do**
   Output set $\hat{C}_t = [\hat{Y}_t - b_t, \hat{Y}_t + b_t]$
   Observe $Y_t$ and compute score $S_t = \left| Y_t - \hat{Y}_t \right|$
   Compute gradient $g_t \in \partial \ell^{(1-\alpha)}(b, S_t)|_{b=b_t}$ as per (4);
   Update sum of squares: $G_t = G_{t-1} + g_t^2$
   Update radius: $b_{t+1} = b_t - \eta \frac{g_t}{\sqrt{G_t + \varepsilon}}$
**end for**

---

While robust to scale, SF-OGD still requires the selection of the global learning rate $\eta$. In our experiments, we select $\eta$ via grid search from the set $\{0.01, 0.1, 0.25, 1, 10, 25, 100\}$. These candidate values are adopted from the well-tuned choices reported in Podkopaev et al. (2024), ensuring the method can accommodate selected datasets.

## J.4. Conformal P/PI Control

Conformal P/PI Control frames the selection of the conformal radius $b_t$ as a feedback control problem, where the goal is to calibrate the coverage error $err_t = \mathbf{1}\left\{Y_t \notin \hat{C}_t\right\}$ to the set point $\alpha$ (Angelopoulos et al., 2023). The algorithm combines two components: a Proportional (P) controller, also known as Quantile Tracking, and an Integral (I) controller. The P controller updates the radius using online gradient descent on the quantile loss, adjusting $b_t$ proportional to the instantaneous error $err_t - \alpha$. While effective, P control can suffer from steady-state error. To mitigate this, the PI controller adds an integrator term $r_t(\cdot)$ that acts on the cumulative sum of past errors, $E_t = \sum_{i=1}^{t}(err_i - \alpha)$. The update rule for the radius at time $t + 1$ is given by combining these terms with the previous radius:

$$b_{t+1} = b_t + \eta(err_t - \alpha) + r_t(E_t), \tag{46}$$

where $\eta$ is the learning rate and $r_t$ is a saturation function (e.g., a tangent function) designed to stabilize coverage under arbitrary distribution shifts (Angelopoulos et al., 2023).

---

**Algorithm 5** Conformal P/PI Control

---

    **Input:** Target miscoverage $\alpha$, learning rate $\eta > 0$, integrator function $r_t(\cdot)$.
    **Note:** For P Control, set $r_t(x) = 0$.
    **Initialize:** Radius $b_1 = 0$, cumulative error $E_0 = 0$.
    **for** $t = 1, 2, \ldots$ **do**
        Receive input $x_t$ and forecast $\hat{Y}_t$
        Output set $\hat{C}_t = [\hat{Y}_t - b_t, \hat{Y}_t + b_t]$
        Observe $Y_t$ and compute error indicator $err_t = \mathbf{1}\{Y_t \notin \hat{C}_t\}$
        Update cumulative error: $E_t = E_{t-1} + (err_t - \alpha)$
        Compute P-step: $\delta_P = \eta(err_t - \alpha)$
        Compute I-step: $\delta_I = r_t(E_t)$
        Update radius: $b_{t+1} = b_t + \delta_P + \delta_I$
    **end for**

---

The hyperparameters for the P/PI controller are set following the heuristics provided in Appendix B of Angelopoulos et al. (2023). The proportional gain (learning rate) is typically set adaptively as $\eta = \lambda \hat{B}_t$, where $\hat{B}_t$ is the maximum score observed in a trailing window (or a global bound) and $\lambda \in (0, 1]$ is a scaling factor (0.1 was recommended as a good default). For the PI controller, the integrator is defined as $r_t(E) = K_I \tan(E \log(T)/(TC_{sat}))$. The constant $K_I$ aligns the integrator's output with the scale of the nonconformity scores; it is recommended to be set to a hypothesized upper bound on the scores (e.g., $K_I \approx \max S_t$). The parameter $C_{sat}$ controls the saturation point of the integrator and is derived from a theoretical guarantee to ensure the miscoverage does not exceed a tolerance $\delta$ by time $T$ (e.g., $C_{sat} \approx \frac{2}{\pi} \log(T\delta)$). In the original implementation, both $K_I$ and $C_{sat}$ are often pre-tuned or fixed heuristically for specific datasets to ensure stability.

Specifically in our experiments, the scalar multipliers $\lambda$ are selected from the fixed grid $\{0, 0.05, 0.1, 0.5, 1\}$ for all datasets.[3] For the PI controller, the integrator parameters are pre-tuned heuristically as described previously, while the same grid of $\lambda$ values is used for the proportional component.

Note that we do not compare against the full Conformal PID framework, specifically its derivative (D) component known as *Scorecasting* (Angelopoulos et al., 2023). Scorecasting introduces a secondary modeling layer that fundamentally alters the score distribution by predicting and residualizing errors, effectively transforming the problem into an easier one. To ensure a fair assessment of the conformal update rules themselves and to maintain consistency with the standard evaluation established in prior literature (Gibbs & Candes, 2021; Gibbs & Candès, 2024), we restrict our comparison to methods that adapt to the original sequence of nonconformity scores without modification.

## J.5. Trivial Predictor

The Trivial Predictor serves as a minimal baseline designed to verify the validity of coverage metrics. It guarantees that the empirical coverage exactly matches the target level $1 - \alpha$ at fixed periodic intervals, completely independent of the

---

[3]In fact, 0.1 was recommended by Angelopoulos et al. (2023) as a good default.

data distribution. The method approximates the target coverage probability as a rational fraction $K/N \approx 1 - \alpha$. It then generates a deterministic, periodic sequence of prediction sets consisting solely of infinite sets ($b_t = \infty$) and zero-radius sets ($b_t = 0$). By distributing the $K$ infinite sets as evenly as possible over each cycle of length $N$, the predictor ensures that the cumulative coverage error returns to exactly zero at the end of every cycle, providing perfect validity but practically useless set predictions.

---

**Algorithm 6** Trivial Predictor (Deterministic Cyclic Coverage)

---

**Input:** Target miscoverage $\alpha \in (0, 1)$.
**Initialize:** Rational approximation $1 - \alpha \approx \frac{K}{N}$ (e.g., via continued fractions).
**Initialize:** Time step $t = 0$.
**for** $t = 1, 2, \ldots$ **do**
    Determine current radius $b_t$ based on previous update.
    Output set $\hat{C}_t = [\hat{Y}_t - b_t, \hat{Y}_t + b_t]$
    Calculate position in cycle: $i = t \pmod{N}$
    Compute accumulated coverage credits:
        $acc_{curr} = \lfloor \frac{i \cdot K}{N} \rfloor$
        $acc_{next} = \lfloor \frac{(i+1) \cdot K}{N} \rfloor$
    **if** $acc_{next} > acc_{curr}$ **then**
        Set next radius: $b_{t+1} = \infty$ {Output Full Set}
    **else**
        Set next radius: $b_{t+1} = 0$ {Output Empty Set}
    **end if**
**end for**

---

## K. Score Growth Demonstration

In Section 3, we introduced the polynomial growth assumption $S_t \le Dt^q$, which relaxes the standard boundedness ($S_t \le C$) used in prior literature. Figure 5 illustrates the practical implication of this relaxation. To see why this matters in practice, consider the Apple (AAPL) dataset used in our experiments. The nonconformity scores follow a roughly quadratic trend ($q \approx 2$). Financial time series frequently exhibit volatility clustering and price drift that violate static bounds. Our analysis extends guarantees to the polynomial-growth regime, which provides a tighter fit for such non-stationary real-world data.

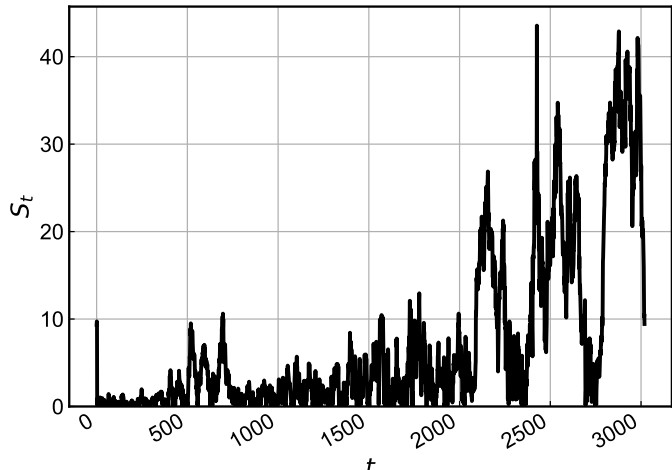

*Figure 5.* Observed nonconformity scores for AAPL stock returns.

## L. Slight Under-Coverage and $\alpha$-Correction

In our experiments, UP-OCP (and most baselines) exhibits a small finite-sample under-coverage, e.g., $\approx 0.93$ against a $0.95$ target on AXP data. In this section we explain why this is expected, and show how to minimally change *any* OCP algorithm to track the desired level more closely, *without* changing the Pareto-frontier curves or the rate of the theoretical guarantees.

**Why under-coverage arises.** There is a fundamental trade-off between marginal coverage and prediction set size in OCP. A trivial predictor achieves near-perfect coverage (Appendix J). Recent lower bounds Srinivas (2026) prove that any algorithm whose prediction sets have size at most $\mu$ times the optimal must incur $\mathcal{O}\left(\log(1/\varepsilon_{\min})/\log\mu \cdot \alpha T\right)$ coverage errors, where $\varepsilon_{\min}$ is the smallest relevant scale. So, the observed $93\%$ coverage for UP-OCP reflects its navigation of this Pareto frontier: it produces tighter prediction sets at the cost of slightly more miscoverage events than the target. Yet, it is guaranteed to reach the target coverage asymptotically.

Consequently, an algorithm that produces tight sets must necessarily sit slightly below the nominal level over any finite horizon: the observed $\approx 0.93$ for UP-OCP reflects exactly this navigation of the Pareto frontier, trading a small coverage deficit for substantially tighter intervals. This is not specific to UP-OCP: KT undercovers similarly ($\approx 0.92$ on AXP), while DtACI *overshoots* on other datasets (e.g., $\approx 0.96$ on the quadratic-trend data). The oracle-tuned baselines (e.g., SF-OGD with an ex-post-selected learning rate) incur a smaller miscoverage *precisely because* they benefit from ex-post choice that is unavailable in a genuine online setting. Yet, it is guaranteed to reach the target coverage asymptotically.

**A principled $\alpha$-correction.** Since coverage in the online setting cannot be guaranteed at any fixed time $t$ but only asymptotically, what we propose is a heuristic, but a theoretically principled one. As most algorithms tend to undercover, one runs the algorithm with a slightly inflated target, $\alpha' = \alpha + 1/\sqrt{T}$, and reports the resulting sets. Two properties make this correction safe. *(i) The convergence rate is preserved.* Write $\text{MisCov}_T(\alpha)$ for the miscoverage of the algorithm run at target $\alpha$. The realized coverage of the corrected run deviates from the original target $1 - \alpha$ by at most

$$|\text{Cov}_T - (1-\alpha)| \le \text{MisCov}_T(\alpha') + \underbrace{|\alpha' - \alpha|}_{=1/\sqrt{T}},$$

by the triangle inequality. If the algorithm guarantees $\text{MisCov}_T = \mathcal{O}(1/\sqrt{T})$ or slower, the added $k/\sqrt{T}$ term is of the same or lower order, so the asymptotic rate is unchanged; for UP-OCP, whose miscoverage decays as $\mathcal{O}(\ln T/T)$ (Theorem 4.4), the inflation is a strictly lower-order perturbation and the corrected run still converges to $1 - \alpha$. Thus the correction trades a vanishing, controlled bias for improved finite-sample tracking. *(ii) The Pareto frontier is invariant.* The Pareto frontier is, by construction, the set of realized (coverage, size) pairs obtained by sweeping the target level. Replacing $\alpha$ by $\alpha'$ is merely a different choice of target, i.e., a movement *along* the same frontier curve, not a new operating regime; hence the frontier itself is unchanged and no comparison in the paper is affected. Empirically, the correction closes the gap: on AXP it lifts UP-OCP from 0.932 to 0.953 coverage; see the second column (UP ($\alpha$-corr.)) of Table 3, where the set sizes enlarge exactly as predicted when moving up the frontier.

## M. Full Results for AXP Dataset

In this section, we provide a comprehensive analysis of the American Express (AXP) dataset, complementing the efficiency and calibration results presented in Section 5. Note that an initial warm-up period of 100 days is used for training the initial base forecaster. Additionally, for all plots presented in this section, we discard the first 50 days of the evaluation period as a burn-in.

**Local Adaptivity.** In the main text, we noted that aggregate metrics like marginal coverage can mask significant local failures, such as error clustering or instability. To visualize these behaviors, Figures 6 through 10 provide 1-vs-1 comparisons of local adaptivity between UP-OCP and all parameter-free and tuned baselines.

In each figure, the top left panel shows the local coverage computed over a rolling window of 100 days, with the dashed black line indicating the target coverage level ($1 - \alpha = 95\%$). The top right panel displays the local width of the prediction intervals ($2b_t$) using the same window size. The bottom panel illustrates the raw prediction sets $\hat{C}_t$ around the true observation (central black line).

We observe that UP-OCP maintains local coverage tightly around the $95\%$ target, with no significant swings. In contrast, KT exhibits volatility, where the local coverage drops below $75\%$ (e.g., during 2009 and 2015).

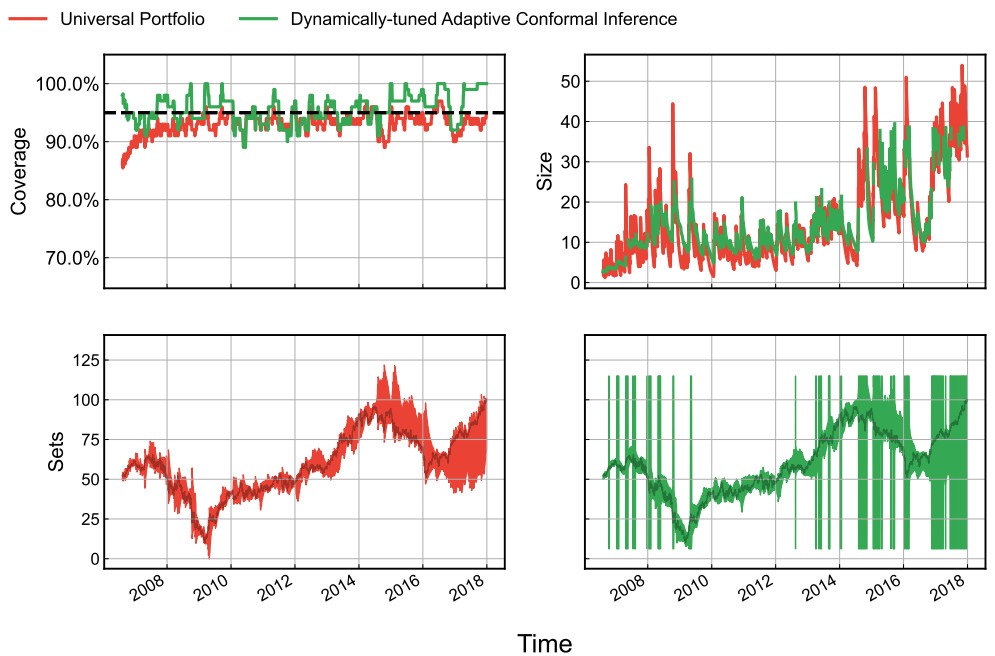

*Figure 6.* UP-OCP versus DtACI for forecasting AXP stock return.

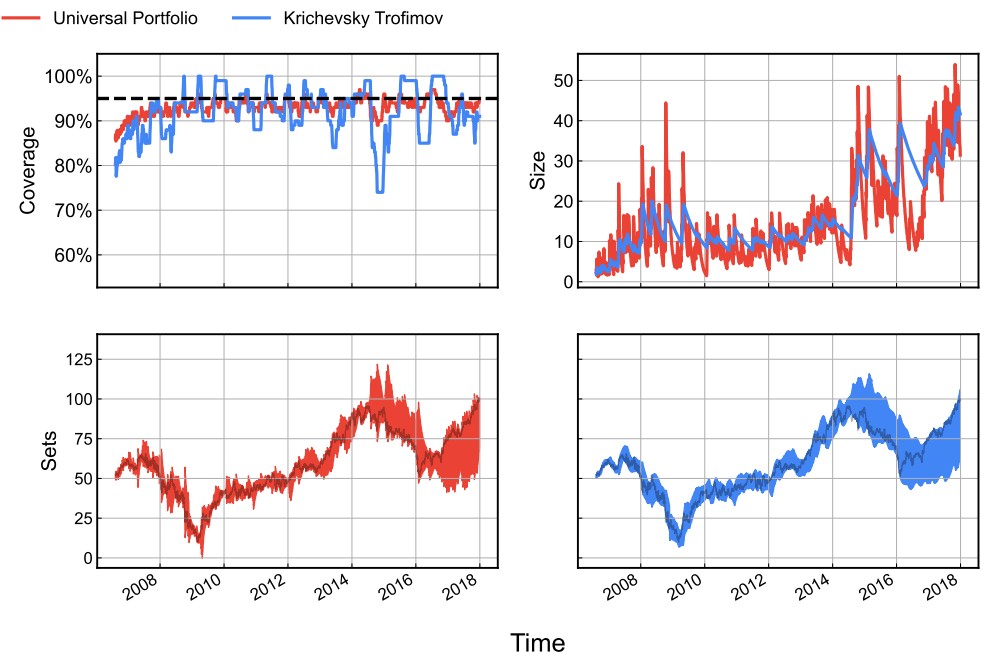

*Figure 7.* As in Figure 6, UP-OCP versus KT.

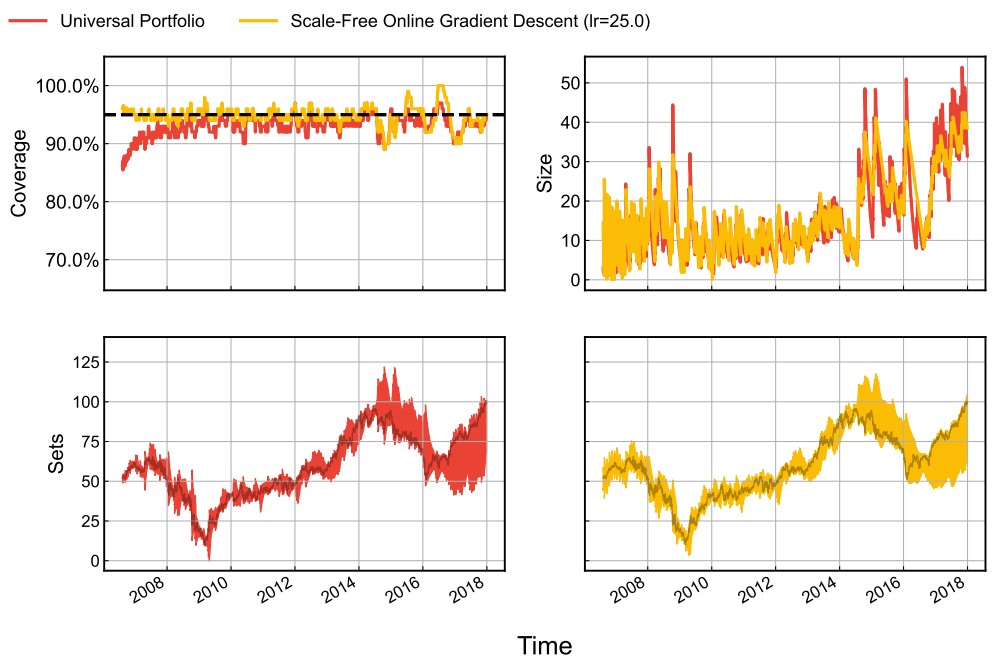

*Figure 8.* As in Figure 6, UP-OCP versus SFOGD (lr=25).

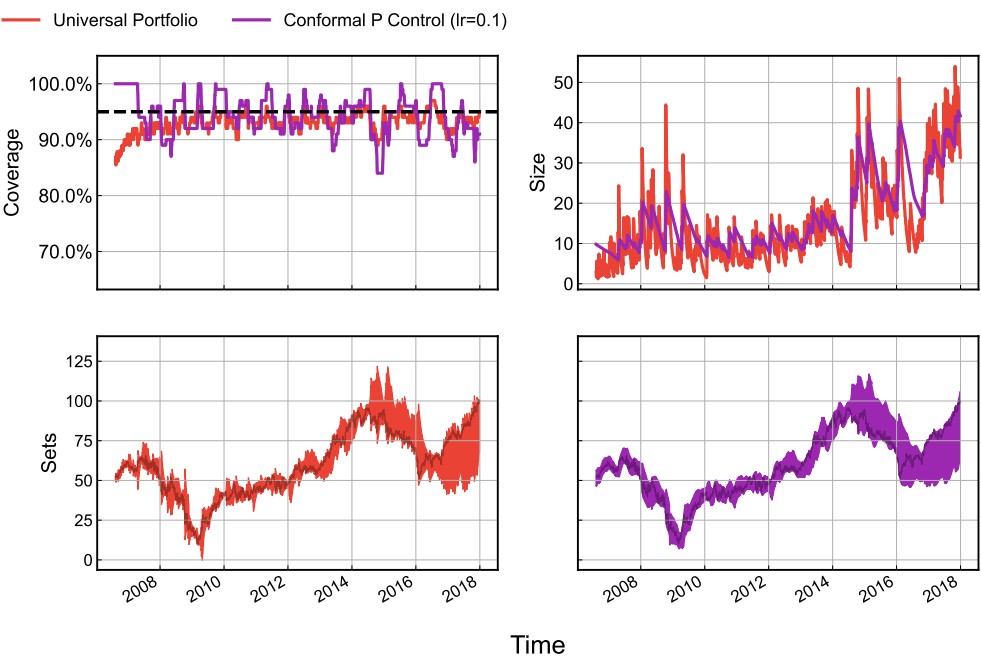

*Figure 9.* As in Figure 6, UP-OCP versus P Ctrl (lr=0.1).

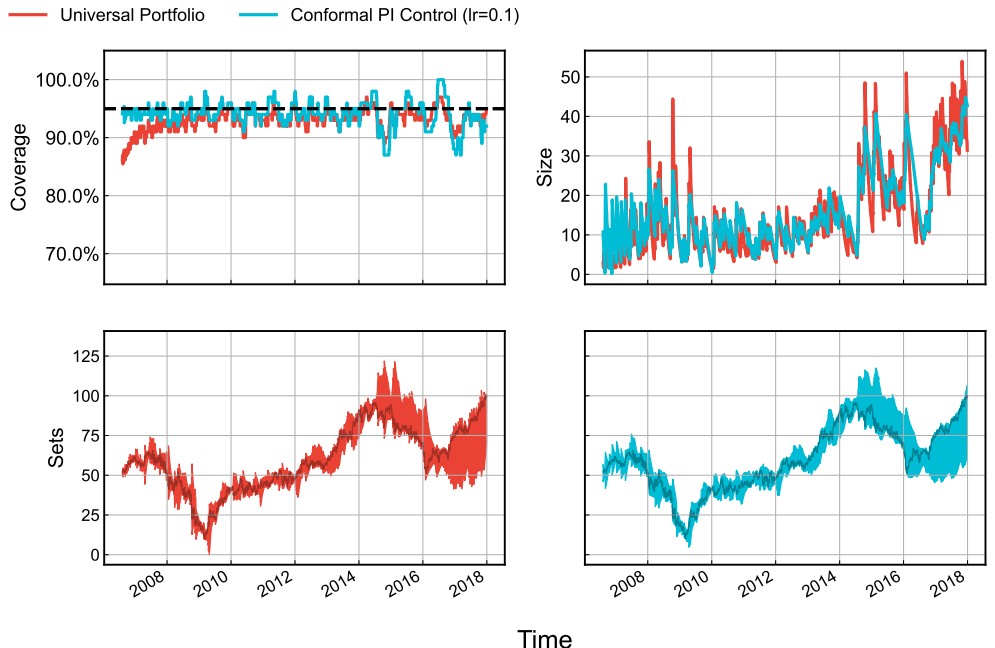

*Figure 10.* As in Figure 6, UP-OCP versus PI Ctrl (lr=0.1).

For completeness, Table 3 extends the results from the main text to include the heuristic conformal P and PI controllers (Angelopoulos et al., 2023). These methods were tuned via grid search to maximize performance.

*Table 3.* Quantitative comparison on the AXP dataset. The second column, UP ($\alpha$-corr.), applies the $\alpha$-correction of Appendix L to UP-OCP (target inflated by $1/\sqrt{T}$); all other columns are uncorrected. The correction lifts the marginal coverage from 0.932 to 0.953 (closest to the 0.95 target among all methods) while keeping the longest error sequence minimal; the prediction sets enlarge accordingly, as expected when moving along the Pareto frontier toward higher coverage.

|  | UP ($\alpha$-corr.) | UP | KT | DtACI | SFOGD (lr=25) | P Ctrl (lr=0.1) | PI Ctrl (lr=0.1) |
|---|---|---|---|---|---|---|---|
| Marginal coverage | 0.953 | 0.932 | 0.92 | 0.956 | 0.948 | 0.942 | 0.941 |
| Longest err sequence | **3** | **4** | 15 | 6 | **3** | 7 | **5** |
| Average set size | 16.4 | **14.8** | 16.9 | $\infty$ | 16.4 | 16.8 | 16.1 |
| Median set size | 13.0 | **11.5** | 12.9 | 12.6 | 13.8 | 13.1 | 13.2 |
| 75% quantile set size | 20.6 | **18.8** | 24.9 | 21.8 | 21.5 | 21.5 | 20.9 |
| 90% quantile set size | 34.3 | **32.3** | 32.5 | $\infty$ | 32.3 | 32.4 | **32** |
| 95% quantile set size | 41.7 | 38 | 36.1 | $\infty$ | 36.1 | 36.6 | 35.9 |

**More Pareto Frontiers and Target-level Tracking.** In Section 5, we prioritized the presentation of *average* prediction set size to penalize the infinite sets produced by algorithms like DtACI; however, robust statistics such as the median and 75% quantile provide insight into the typical performance of the algorithms, ignoring the heavy tails. We present additional Pareto frontiers for these metrics in Figures 11 through 13.

Figure 12 clarifies that DtACI is not inefficient on average days; its median performance overlaps with UP-OCP, and the inefficiency is driven entirely by its inability to handle tail events properly, which is shown by the divergence in the mean (Figure 11) and 75% quantile metrics (Figure 13); however, UP-OCP dominates or matches all baselines across both mean and robust metrics, proving it is both stable in the worst case and efficient in the typical case.

**Target-level Tracking.** Finally, Figure 14 illustrates the ability of the algorithms to track the user-specified target coverage across a spectrum of $\alpha$ values.

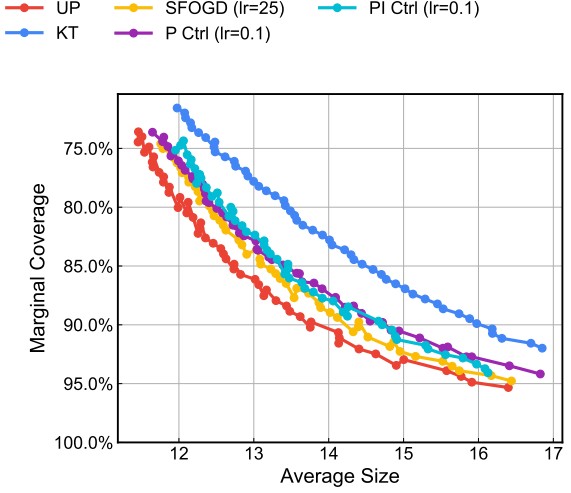

*Figure 11.* Mean prediction set sizes on AXP.

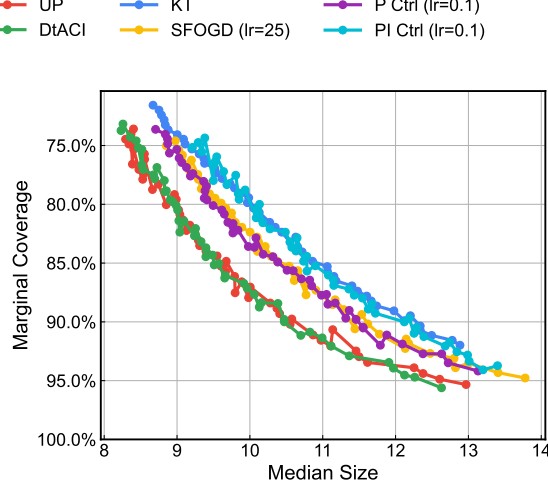

*Figure 12.* Median prediction set sizes on AXP.

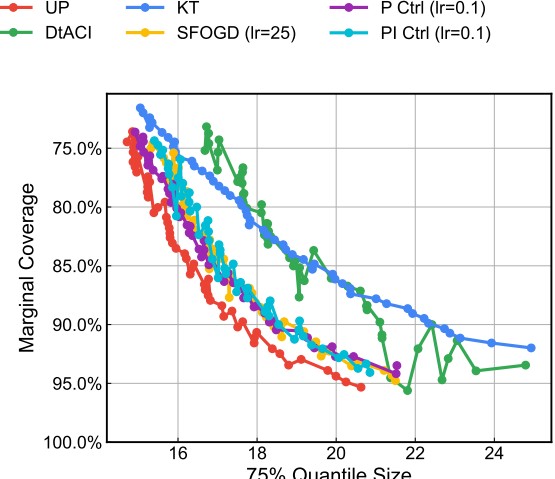

*Figure 13.* 75% quantile prediction set sizes on AXP.

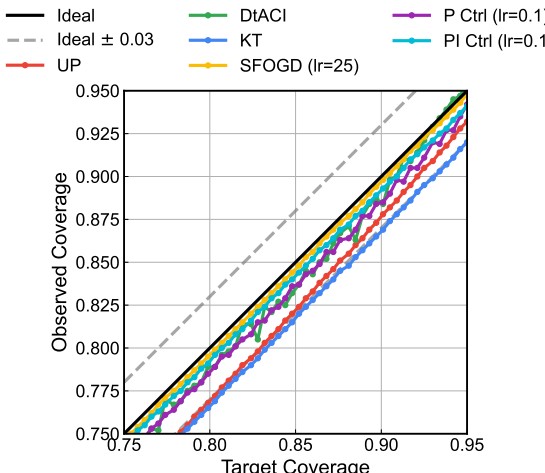

*Figure 14.* Realized versus target coverage on AXP data. Most methods track the diagonal within a small tolerance ($\pm$ 0.03).

# N. Additional Experiments

To demonstrate generalization of our method, we extend our evaluation to a diverse set of real-world and synthetic benchmarks. These include three additional major stocks (AAPL, AMZN, GOOGL), an electricity demand dataset (NSW), and three synthetic environments designed to test adaptivity (sinusoid, stationary random waves, and quadratic drift). The observed behaviors across these experiments remain qualitatively consistent with the findings from the AXP dataset. Therefore, we omit a detailed discussion to avoid redundancy and present the following figures and tables for completeness. We note interesting observations wherever applicable.

**More on Experimental Setup.** For the financial datasets (AAPL, AMZN, GOOGL), we follow the same protocol as the AXP experiment: we employ the Prophet model (Taylor & Letham, 2018) as the base forecaster, targeting a miscoverage rate of $\alpha = 0.05$ with an initial burn-in period of $T_{\text{burnin}} = 100$ days. For the electricity demand dataset, we utilize a standard Autoregressive (AR) model with a burn-in of $T_{\text{burnin}} = 300$ steps to capture the high-frequency intraday seasonality. The synthetic experiments generate nonconformity scores directly to isolate specific distributional shifts (periodicity, sparse spikes, and drift), also using a burn-in of 300 steps. To ensure statistical significance, all synthetic results are reported as averages over 10 independent random seeds, with error bars in figures denoting the standard error of the mean. In all comparisons, we evaluate UP-OCP against the full suite of parameter-free (KT, DtACI) and optimized baselines (SF-OGD, P-Control, PI-Control) described in Section 5.

**Takeaway.** It is worth noting that the optimal hyperparameters for the baseline methods vary across datasets. We illustrate this variability in Appendix O, highlighting that no single hyperparameter configuration yields consistent performance across all benchmarks. This sensitivity necessitates dataset-specific tuning, a requirement that our parameter-free UP-OCP method avoids.

## N.1. AAPL Dataset

**Local Adaptivity.**

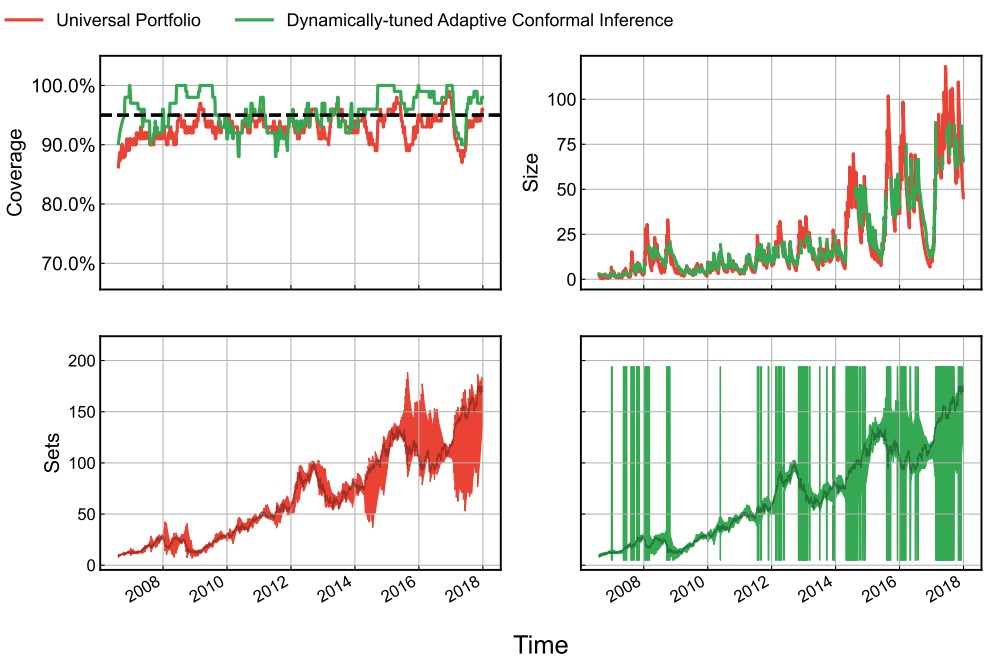

*Figure 15.* UP-OCP versus DtACI for forecasting AAPL stock return.

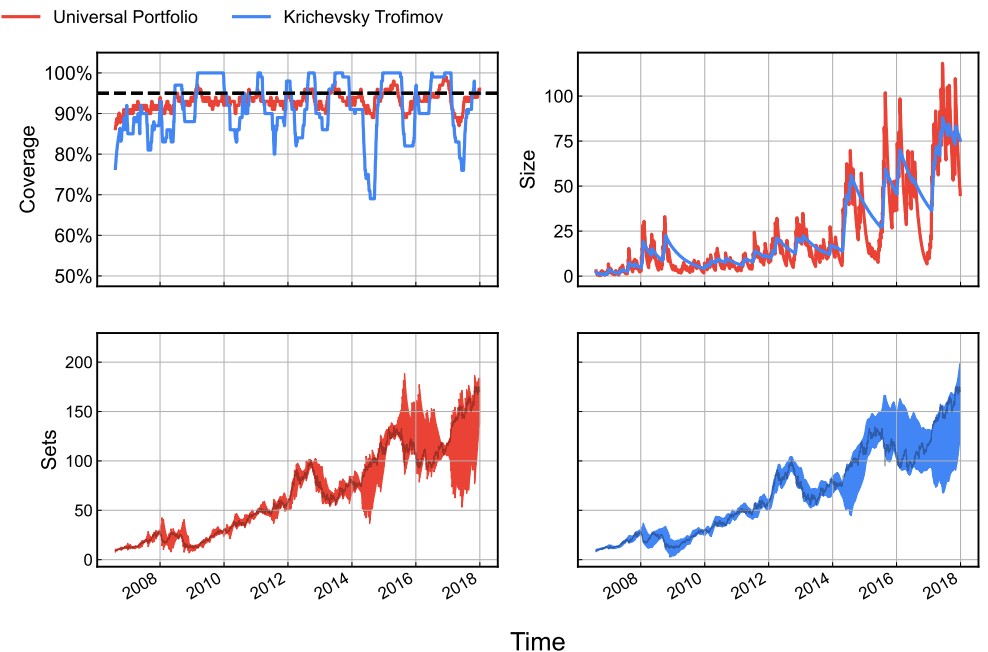

*Figure 16.* As in Figure 15, UP-OCP versus KT.

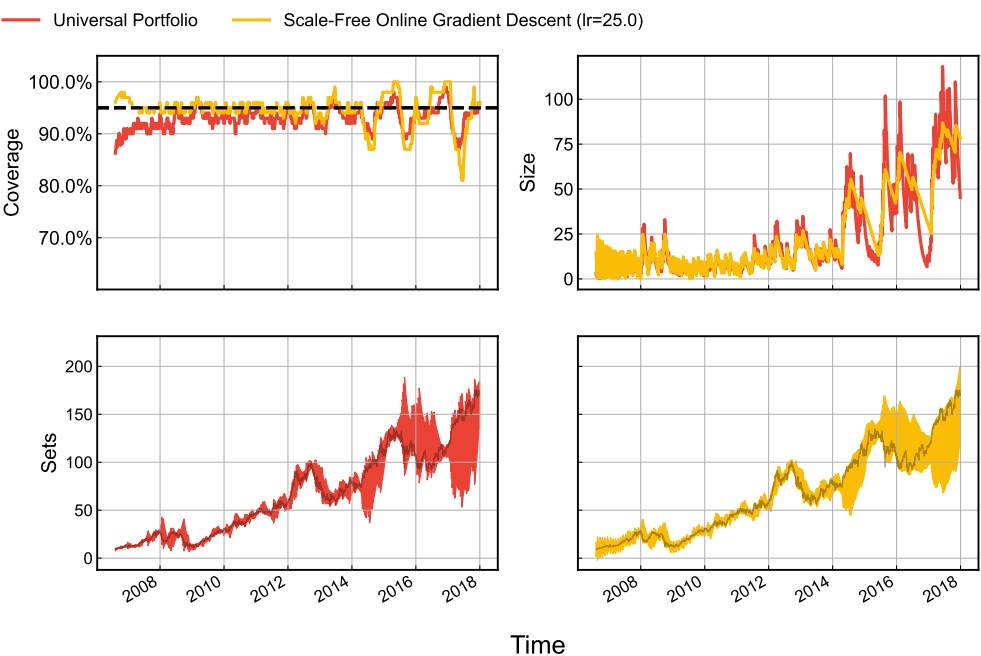

*Figure 17.* As in Figure 15, UP-OCP versus SFOGD (lr=25).

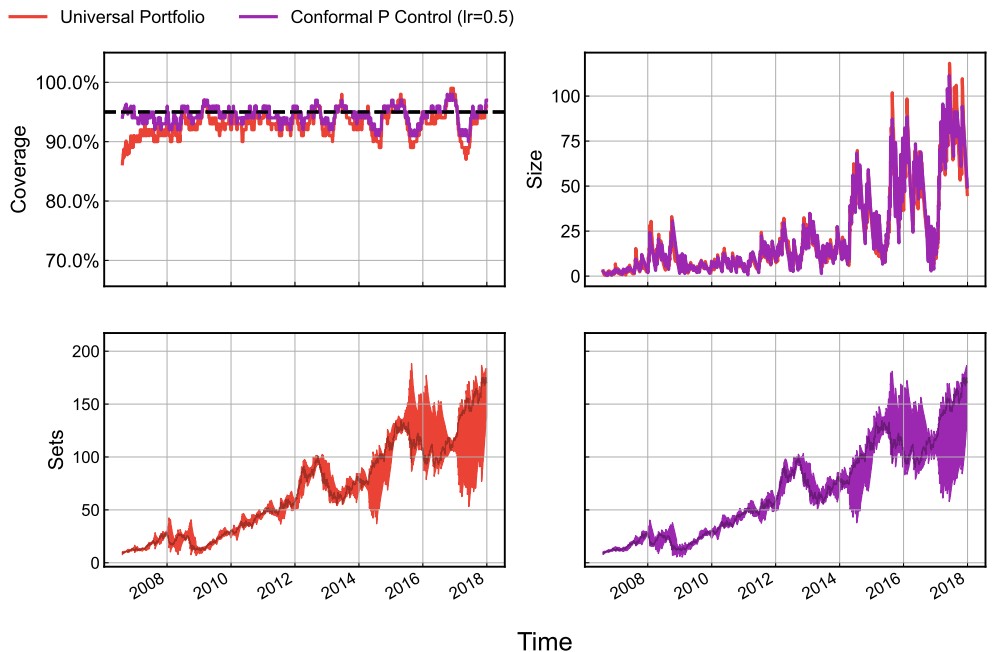

*Figure 18.* As in Figure 15, UP-OCP versus P Ctrl (lr=0.5).

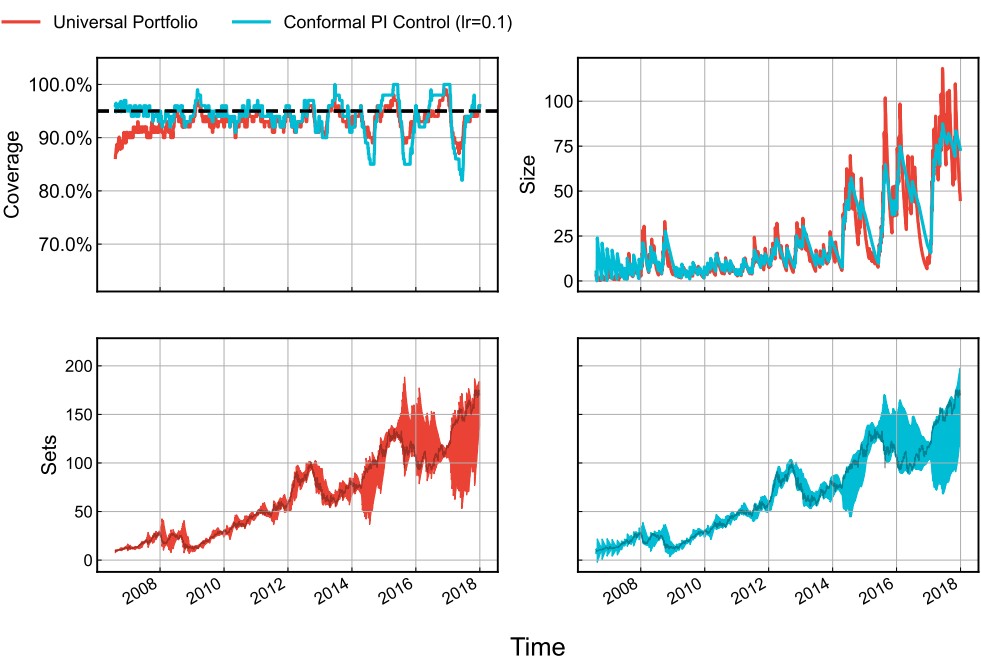

*Figure 19.* As in Figure 15, UP-OCP versus PI Ctrl (lr=0.1).

*Table 4.* Quantitative comparison on the AAPL dataset.

|  | UP | KT | DtACI | SFOGD (lr=25) | P Ctrl (lr=0.5) | PI Ctrl (lr=0.1) |
|---|---|---|---|---|---|---|
| Marginal coverage | 0.932 | 0.915 | 0.958 | 0.945 | 0.945 | 0.941 |
| Longest err sequence | **2** | 16 | 4 | 4 | **2** | 5 |
| Average set size | **21.6** | 24.4 | $\infty$ | 23.7 | 22.8 | 23.2 |
| Median set size | **11.7** | 14.5 | 14 | 13.8 | 13 | 13.7 |
| 75% quantile set size | **27.3** | 39.8 | 55.8 | 36.9 | 29.2 | 33.3 |
| 90% quantile set size | 60.1 | 58.8 | $\infty$ | 59 | 63.7 | 59.8 |
| 95% quantile set size | 75.9 | 75.9 | $\infty$ | 74 | 77.4 | 74.5 |

## More Pareto Frontiers and Target-level Tracking.

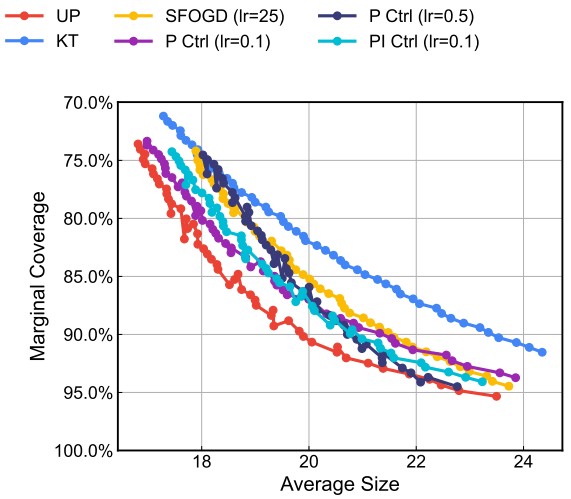

*Figure 20.* Mean prediction set sizes on AAPL.

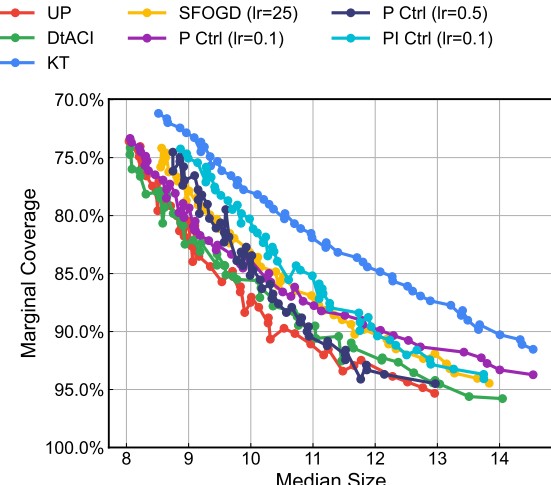

*Figure 21.* Median prediction set sizes on AAPL.

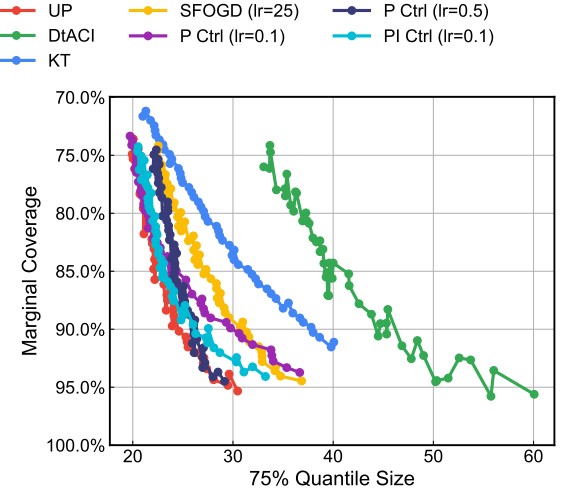

*Figure 22.* 75% quantile prediction set sizes on AAPL.

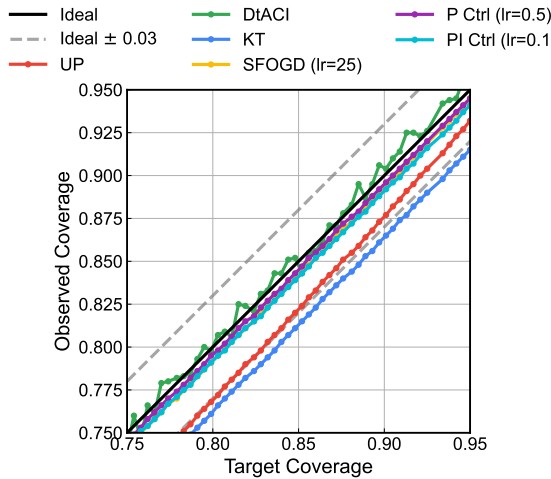

*Figure 23.* Realized versus target coverage on AAPL data. Most methods track the diagonal within a small tolerance ($\pm$ 0.03).

*Remark* N.1. The results here on all three metrics offer a compelling demonstration of adaptivity. UP-OCP (red) does not simply outperform a single baseline; rather, it effectively automates the hyperparameter selection process across the target levels.

Observe the behavior of the tuned P controllers (lr=0.1, purple): at moderate targets (75–85% coverage), the controller is optimal, while the higher-gain controller (lr=0.5, dark blue) is less efficient. Conversely, at high targets (90–95%), the high-gain controller becomes necessary to maintain tight sets, while the low-gain version falls behind. UP-OCP remarkably matches the performance of the *best-tuned* baseline in *each* specific regime. It is able to align with the purple curve at lower targets and the dark blue curve at higher targets, without requiring any manual tuning or gain scheduling.

## N.2. AMZN Dataset

**Local Adaptivity.**

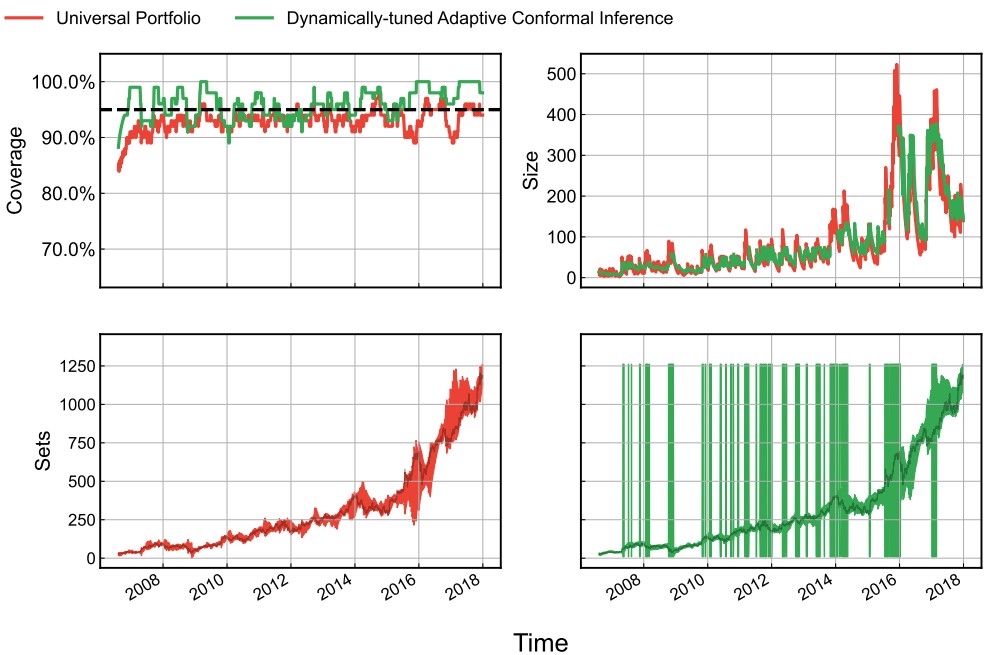

*Figure 24.* UP-OCP versus DtACI for forecasting AMZN stock return.

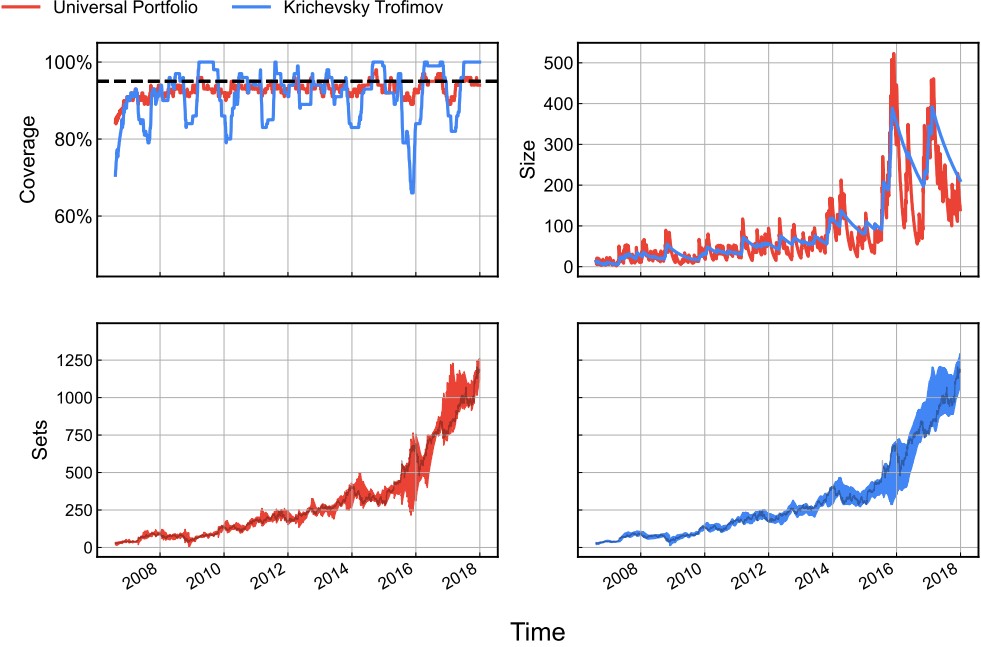

*Figure 25.* As in Figure 24, UP-OCP versus KT.

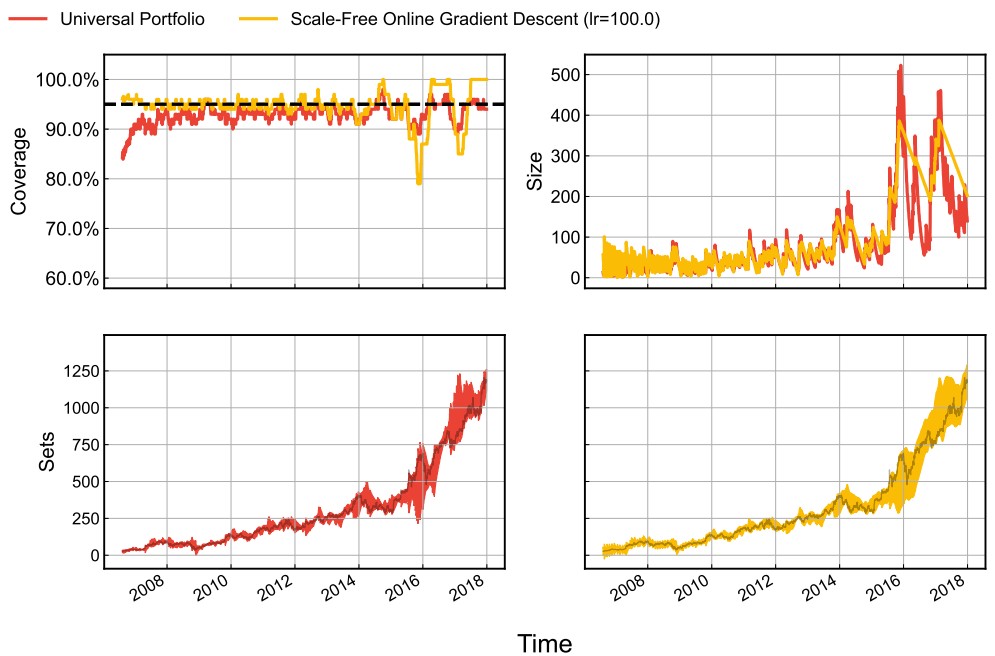

*Figure 26.* As in Figure 24, UP-OCP versus SFOGD (lr=100).

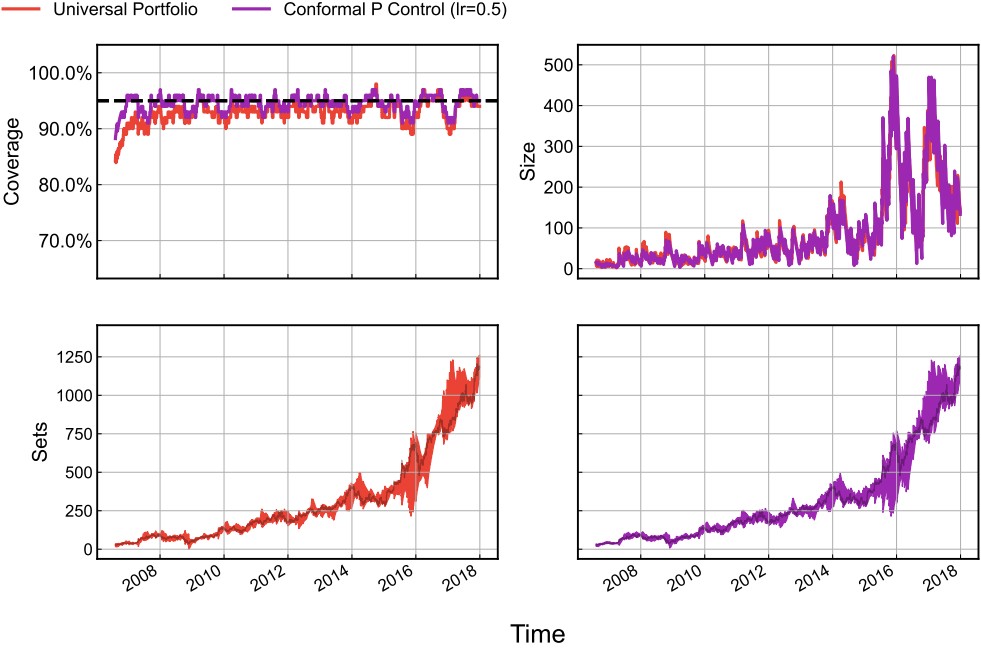

*Figure 27.* As in Figure 24, UP-OCP versus P Ctrl (lr=0.5).

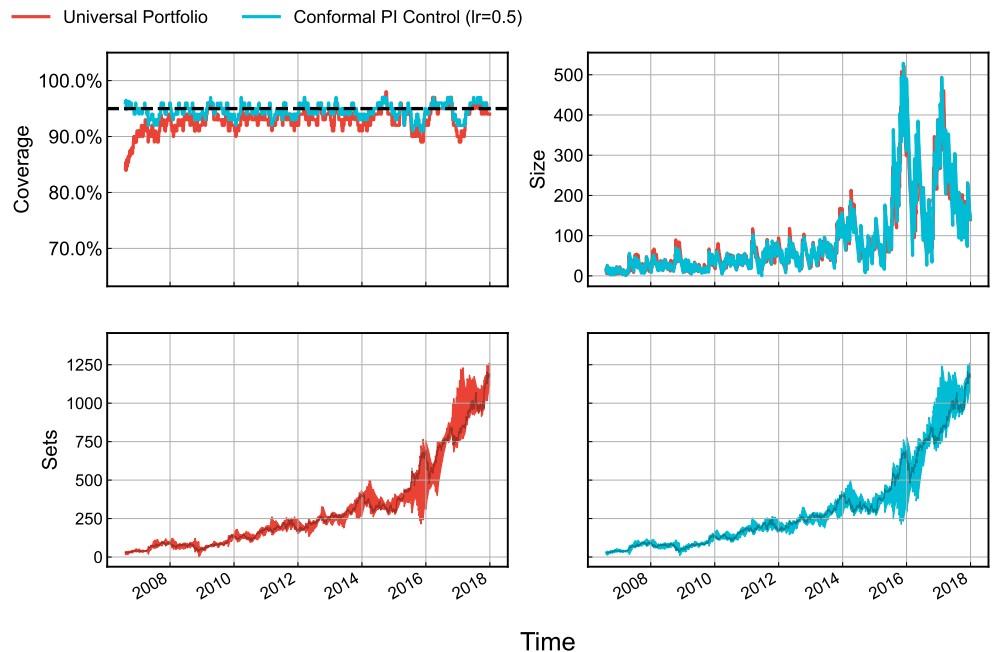

*Figure 28.* As in Figure 24, UP-OCP versus PI Ctrl (lr=0.5).

*Table 5.* Quantitative comparison on the AMZN dataset.

|  | UP | KT | DtACI | SFOGD (lr=100) | P Ctrl (lr=0.5) | PI Ctrl (lr=0.5) |
|---|---|---|---|---|---|---|
| Marginal coverage | 0.931 | 0.919 | 0.962 | 0.947 | 0.946 | 0.946 |
| Longest err sequence | **3** | 18 | 4 | 4 | **3** | **3** |
| Average set size | **82.8** | 99.4 | $\infty$ | 102 | 88.9 | 89.3 |
| Median set size | **49.1** | 57.2 | 60.4 | 57.5 | 51.5 | 51 |
| 75% quantile set size | **97.4** | 113 | 274 | 117 | 107 | 108 |
| 90% quantile set size | **208** | 276 | $\infty$ | 288 | 234 | 239 |
| 95% quantile set size | **292** | 330 | $\infty$ | 335 | 324 | 321 |

**More Pareto Frontiers and Target-level Tracking.**

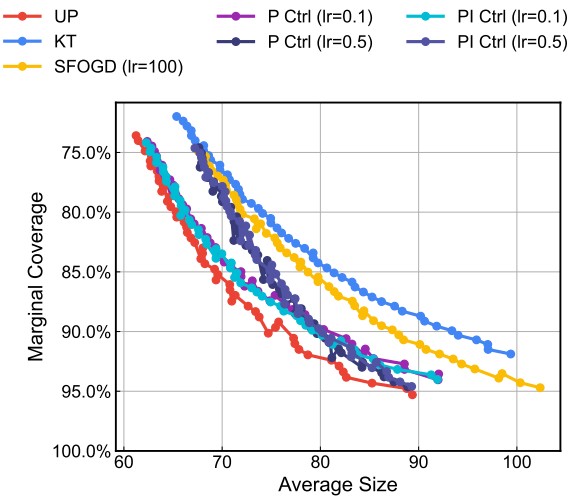

*Figure 29.* Mean prediction set sizes on AMZN.

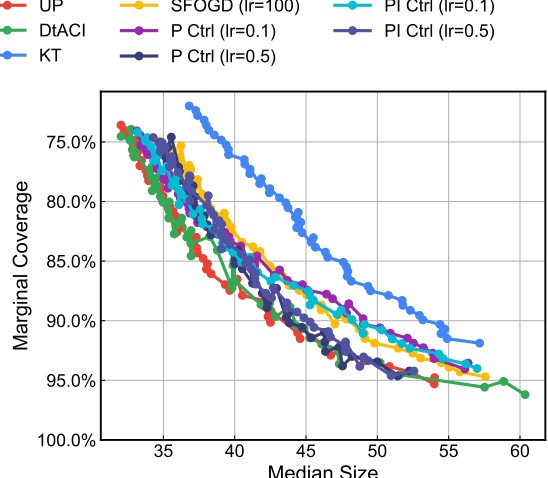

*Figure 30.* Median prediction set sizes on AMZN.

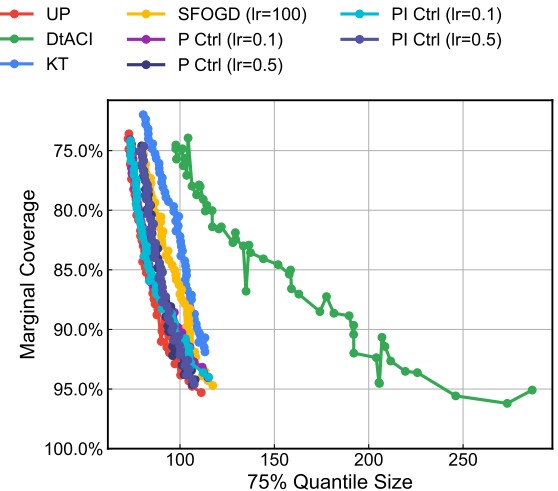

*Figure 31.* 75% quantile prediction set sizes on AMZN.

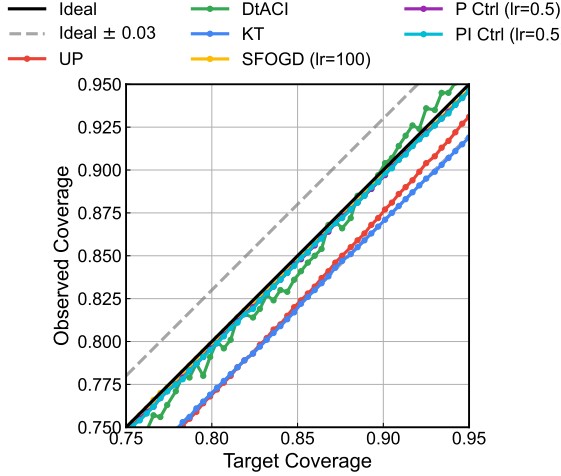

*Figure 32.* Realized versus target coverage on AMZN data. Most methods track the diagonal within a small tolerance ($\pm$ 0.03).

## N.3. GOOGL Dataset

**Local Adaptivity.**

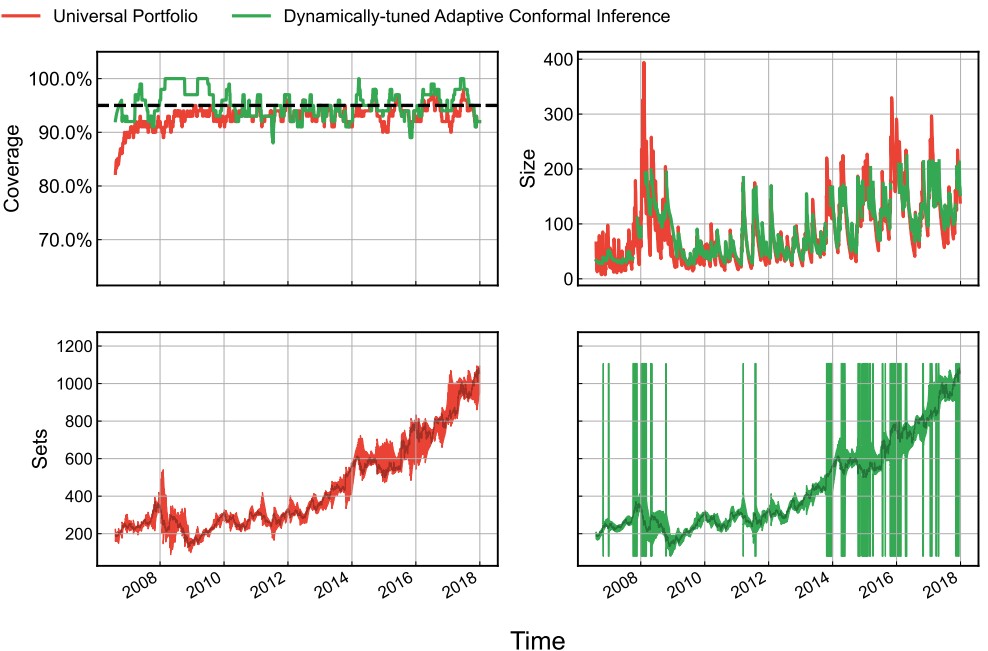

*Figure 33.* UP-OCP versus DtACI for forecasting GOOGL stock return.

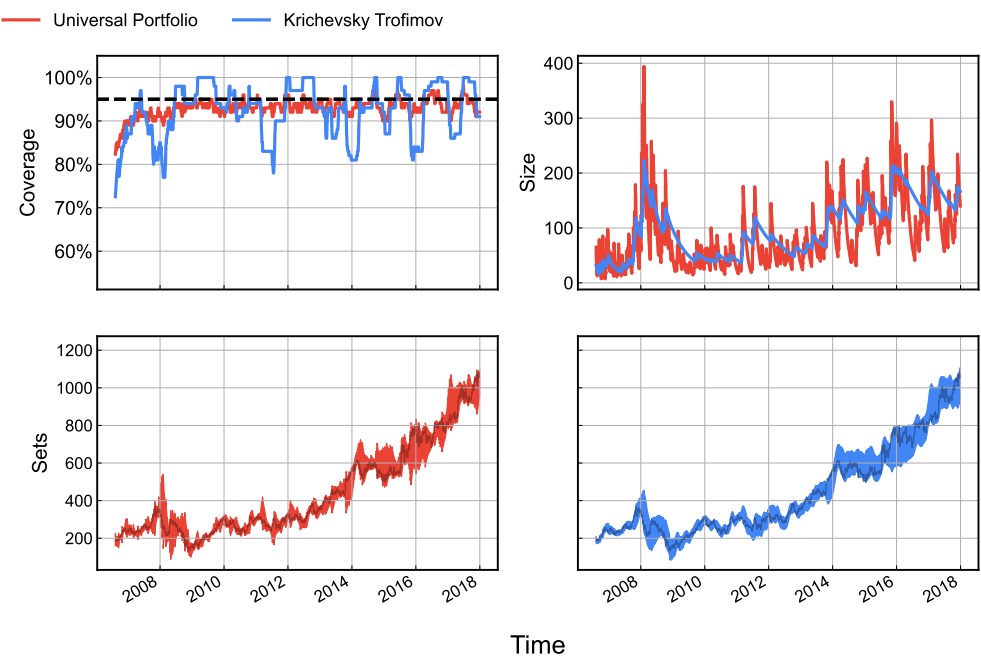

*Figure 34.* As in Figure 33, UP-OCP versus KT.

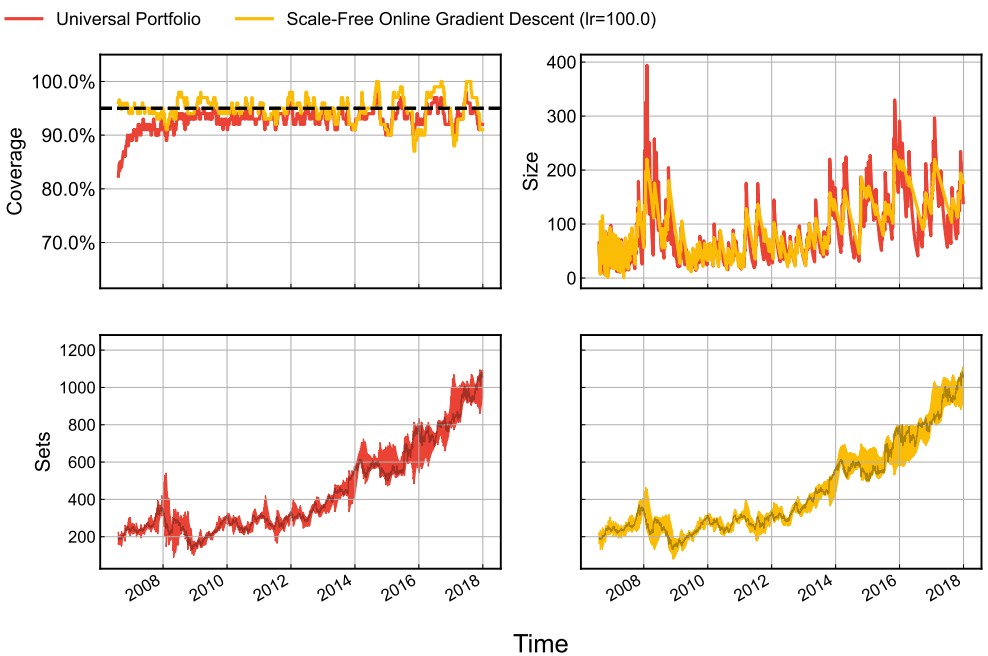

*Figure 35.* As in Figure 33, UP-OCP versus SFOGD (lr=100).

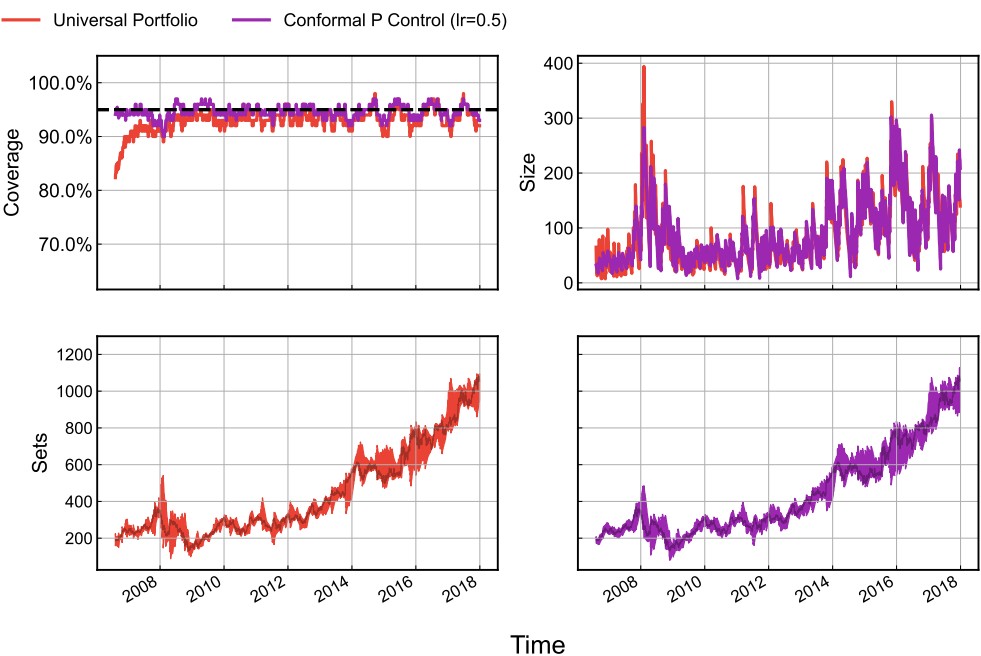

*Figure 36.* As in Figure 33, UP-OCP versus P Ctrl (lr=0.5).

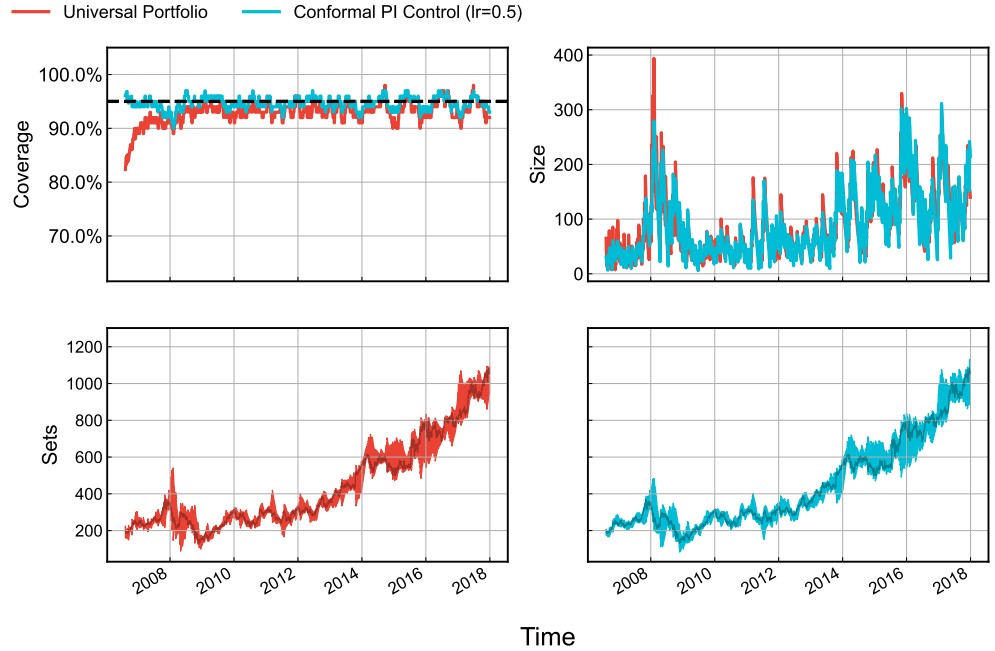

*Figure 37.* As in Figure 33, UP-OCP versus PI Ctrl (lr=0.5).

*Table 6.* Quantitative comparison on the GOOGL dataset.

|                      | UP    | KT    | DtACI    | SFOGD (lr=100) | P Ctrl (lr=0.5) | PI Ctrl (lr=0.5) |
|----------------------|-------|-------|----------|----------------|-----------------|------------------|
| Marginal coverage    | 0.932 | 0.925 | 0.952    | 0.948          | 0.946           | 0.946            |
| Longest err sequence | **2** | 17    | 5        | 4              | **2**           | **2**            |
| Average set size     | **86.7** | 98.5 | $\infty$ | 95.8        | 91.8            | 90.4             |
| Median set size      | **67.8** | 90.6 | 79.2   | 86.1           | 74.4            | 72               |
| 75% quantile set size | **121** | 137 | 133     | 132            | 124             | 126              |
| 90% quantile set size | **173** | 169 | $\infty$ | **171**       | 176             | 178              |
| 95% quantile set size | 204   | 188   | $\infty$ | 193            | 207             | 211              |

**More Pareto Frontiers and Target-level Tracking.**

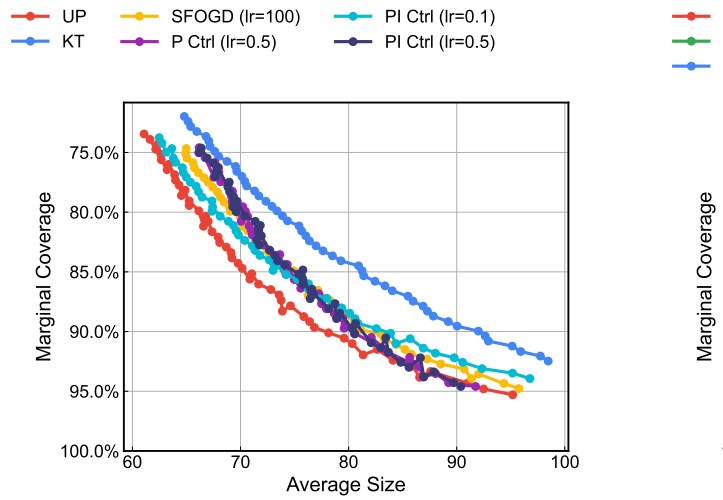

*Figure 38.* Mean prediction set sizes on GOOGL.

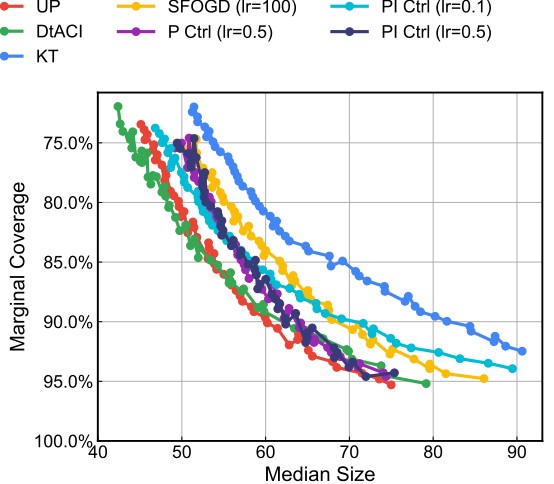

*Figure 39.* Median prediction set sizes on GOOGL.

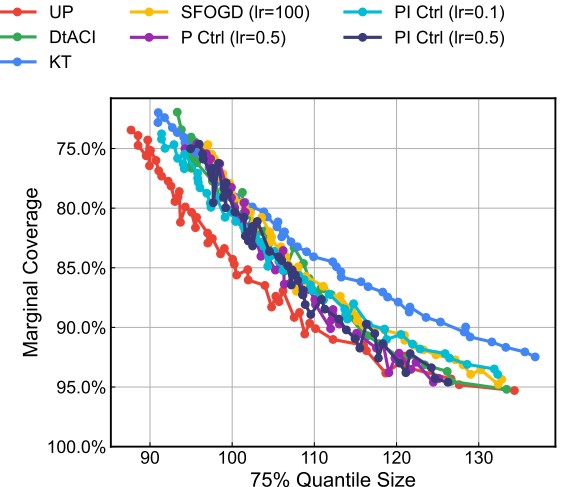

*Figure 40.* 75% quantile prediction set sizes on GOOGL.

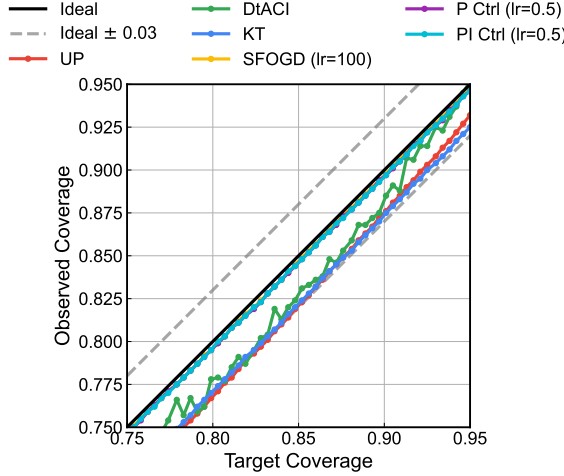

*Figure 41.* Realized versus target coverage on GOOGL data. Most methods track the diagonal within a small tolerance ($\pm$ 0.03).

## N.4. Electricity Demand Dataset

This dataset measures electricity demand in New South Wales collected at half-hour increments from May 7, 1996 to December 5, 1998 (we zoom in on the first 2000 time points).

**Local Adaptivity.**

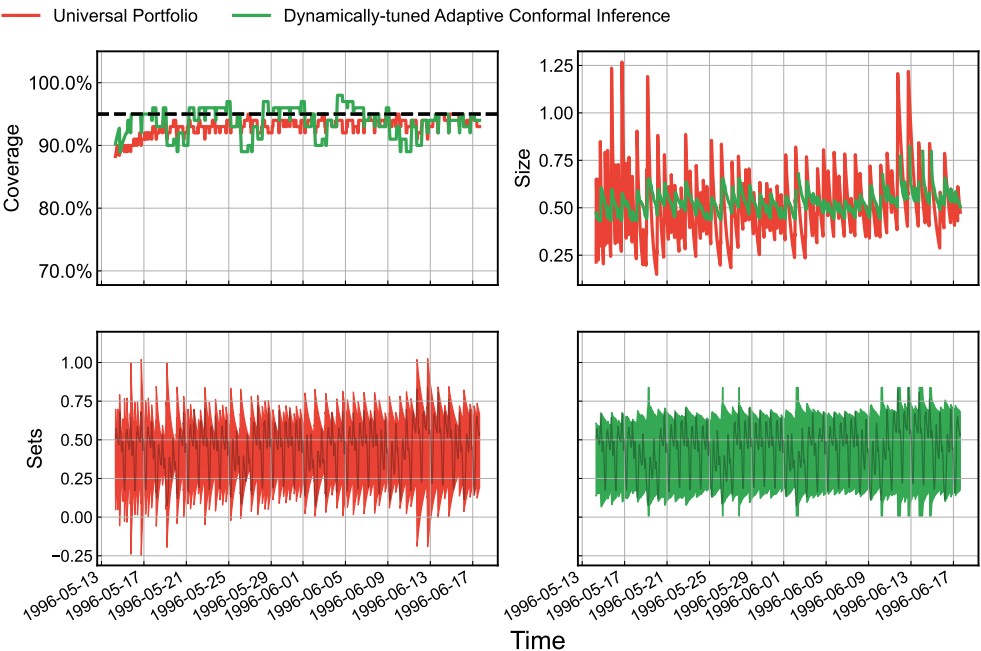

*Figure 42.* UP-OCP versus DtACI for forecasting electricity demand.

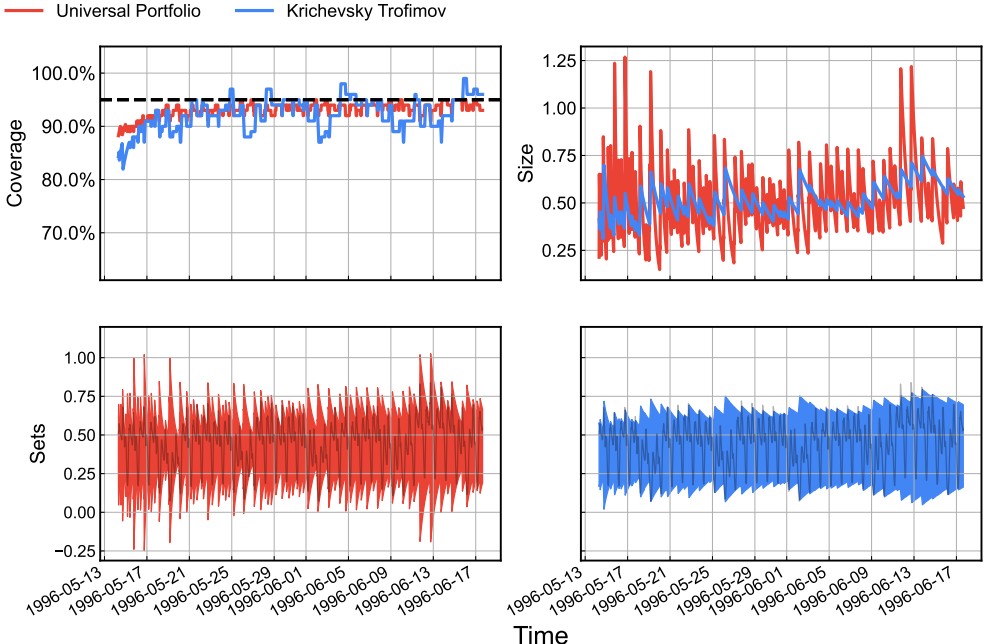

*Figure 43.* As in Figure 42, UP-OCP versus KT.

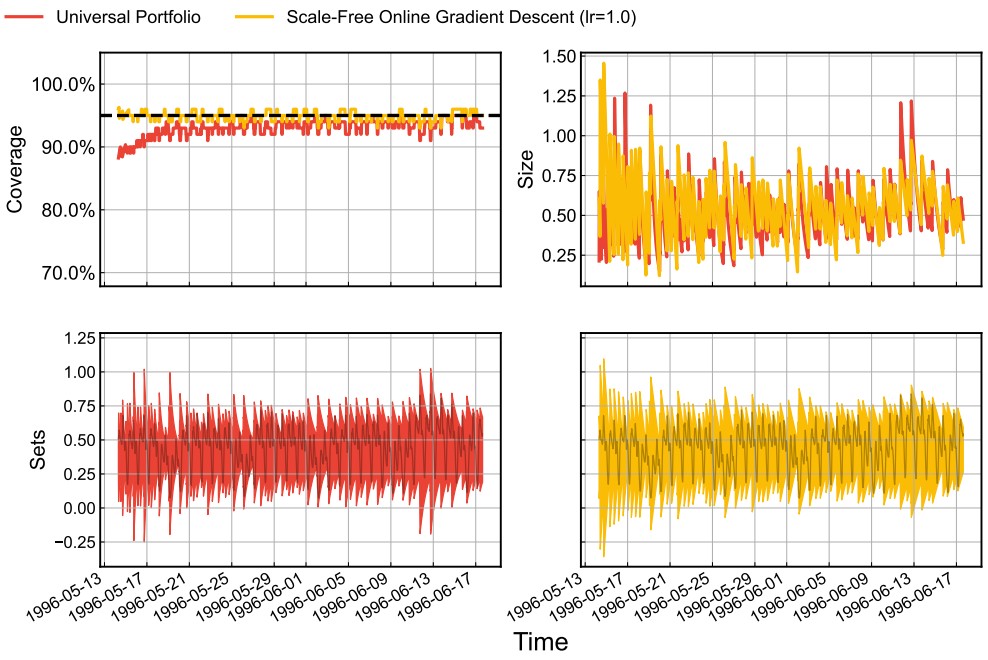

*Figure 44.* As in Figure 42, UP-OCP versus SFOGD (lr=1.0).

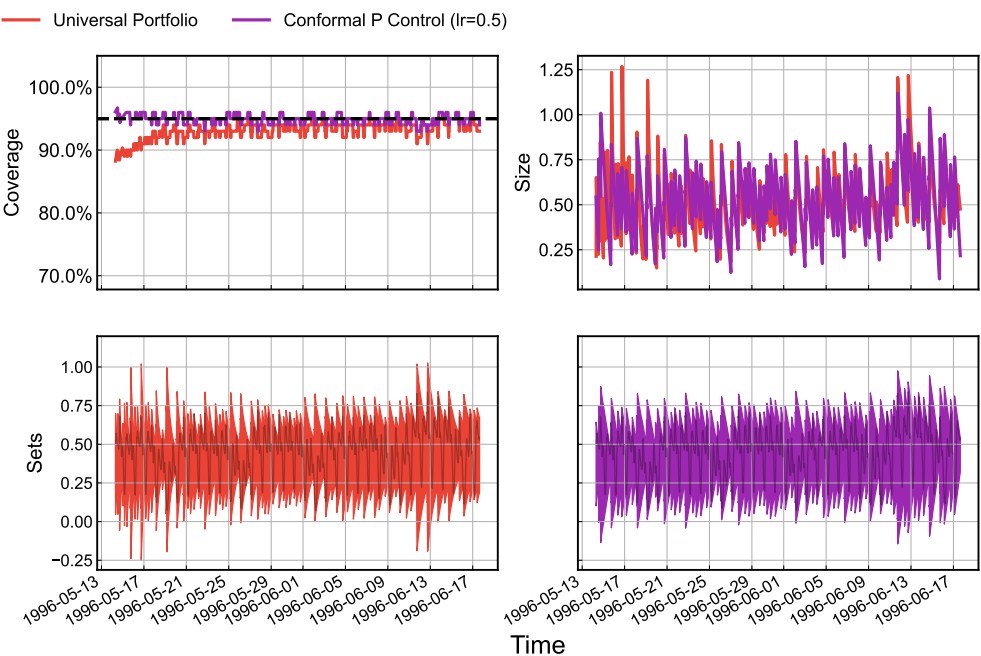

*Figure 45.* As in Figure 42, UP-OCP versus P Ctrl (lr=0.5).

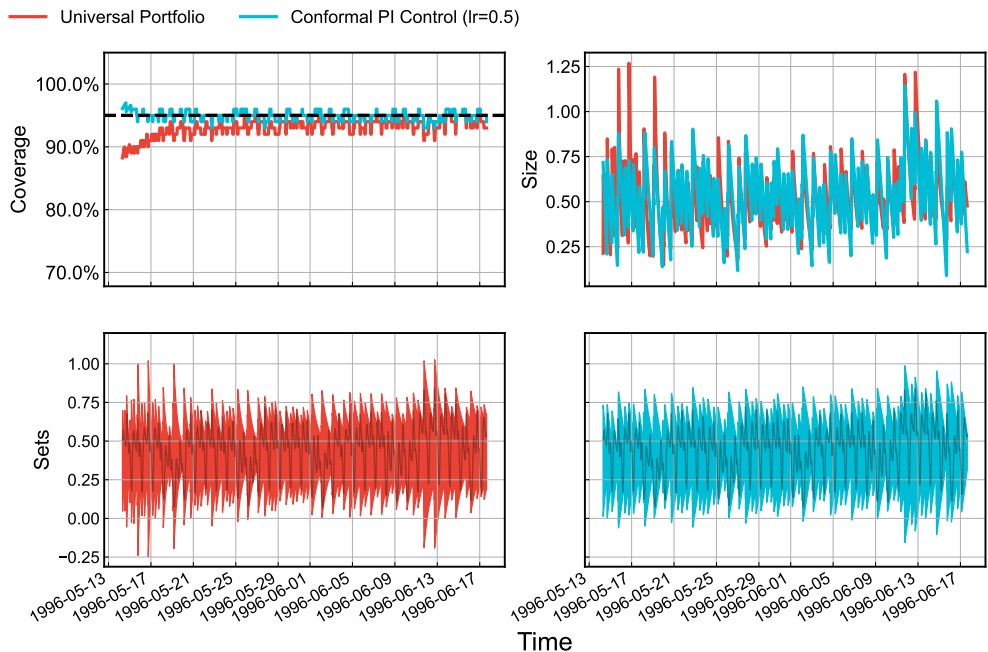

*Figure 46.* As in Figure 42, UP-OCP versus PI Ctrl (lr=0.5).

*Table 7.* Quantitative comparison on the electricity demand dataset.

|  | UP | KT | DtACI | SFOGD (lr=1.0) | P Ctrl (lr=0.5) | PI Ctrl (lr=0.5) |
|---|---|---|---|---|---|---|
| Marginal coverage | 0.933 | 0.927 | 0.939 | 0.95 | 0.95 | 0.95 |
| Longest err sequence | **3** | 7 | 6 | **2** | **1** | **2** |
| Average set size | **0.507** | 0.518 | $\infty$ | 0.561 | 0.528 | 0.533 |
| Median set size | **0.487** | 0.506 | 0.53 | 0.552 | 0.524 | 0.525 |
| 75% quantile set size | 0.593 | 0.572 | 0.563 | 0.664 | 0.637 | 0.65 |
| 90% quantile set size | 0.715 | 0.635 | 0.619 | 0.772 | 0.742 | 0.748 |
| 95% quantile set size | 0.793 | 0.664 | 0.671 | 0.862 | 0.806 | 0.821 |

**More Pareto Frontiers and Target-level Tracking.**

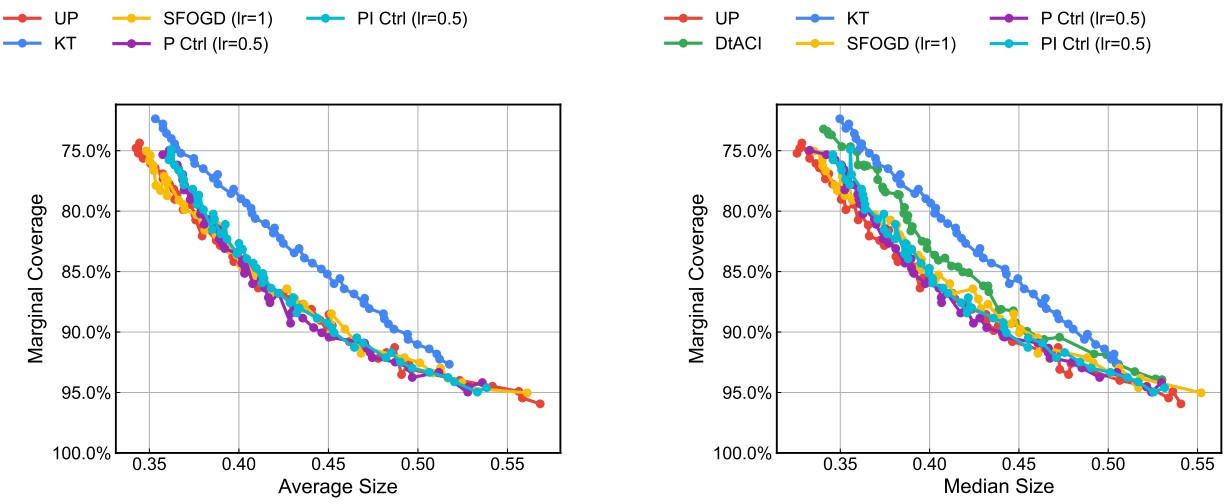

*Figure 47.* Mean prediction set sizes on electricity demand.

*Figure 48.* Median prediction set sizes on electricity demand.

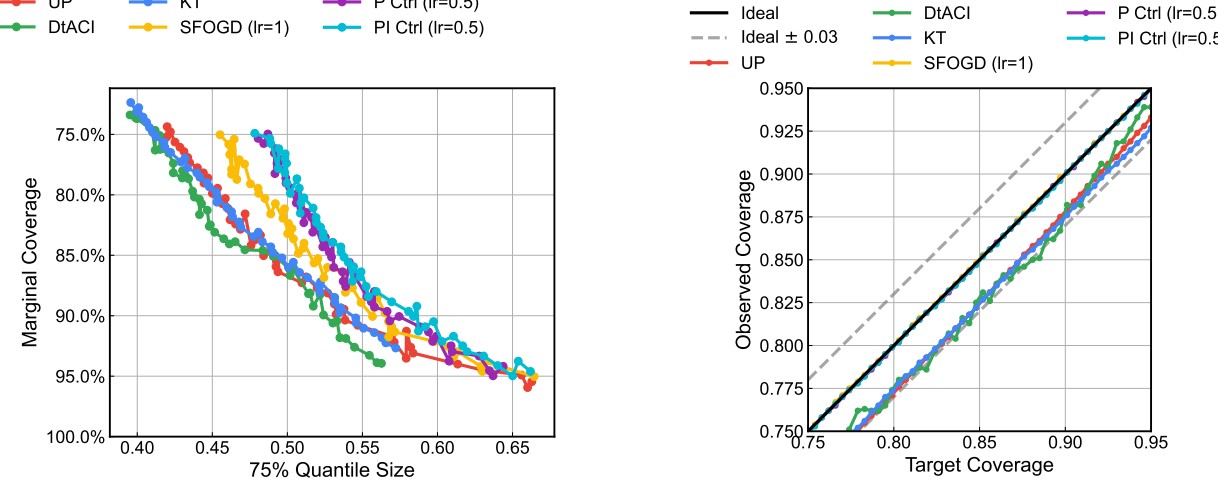

*Figure 49.* 75% quantile prediction set sizes on electricity demand.

*Figure 50.* Realized versus target coverage on electricity demand. Most methods track the diagonal within a small tolerance ($\pm$ 0.03).

## N.5. Synthetic Sinusoid

We generate the nonconformity scores $S_t$ as a sinusoidal wave distorted by Gaussian noise. Formally, for $t = 1, \ldots, T$,

$$S_t = \max\left(0, \left[\sin\left(\frac{2\pi t}{P}\right) + 0.5\right] S_{\text{mag}} + S_{\text{min}} + \varepsilon_t\right),$$

where $\varepsilon_t \overset{\text{i.i.d.}}{\sim} \mathcal{N}(0, \sigma^2)$. We fix the period $P = 200$, the magnitude scale $S_{\text{mag}} = 10$, and the minimum offset $S_{\text{min}} = 2$. The noise scale is set to $\sigma = 0.3$. The total sequence length is $T = 3000$.

**Local Adaptivity.**

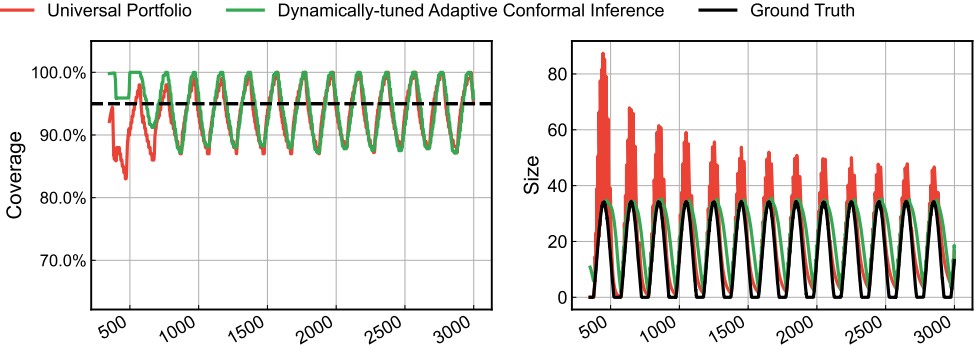

*Figure 51.* UP-OCP versus DtACI for forecasting synthetic sinusoid data.

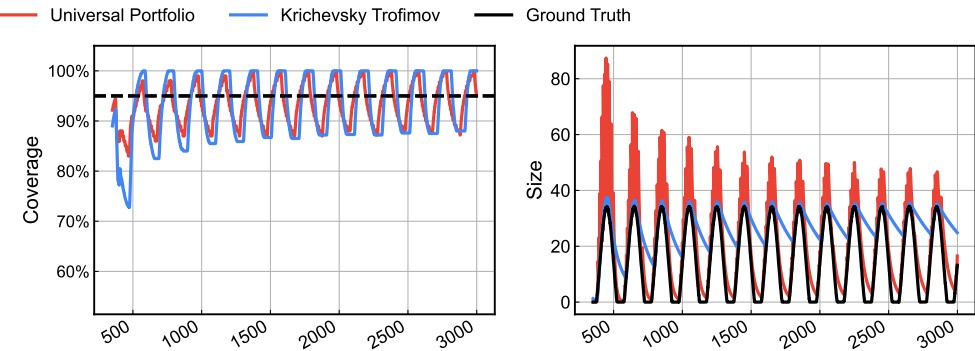

*Figure 52.* As in Figure 51, UP-OCP versus KT.

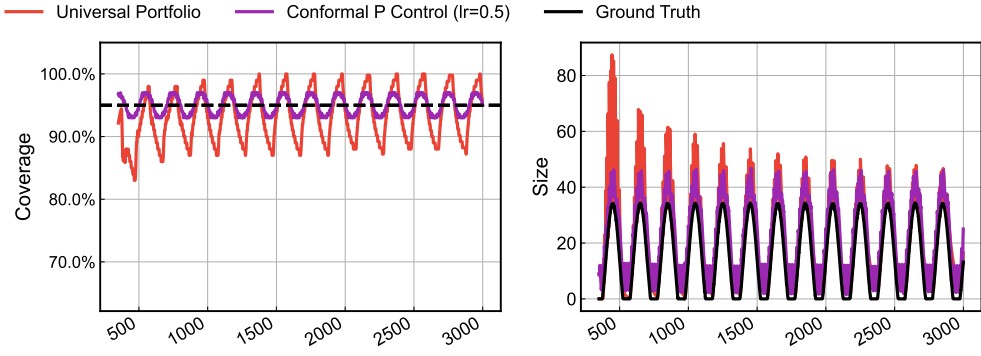

*Figure 53.* As in Figure 51, UP-OCP versus P Ctrl (lr=0.5).

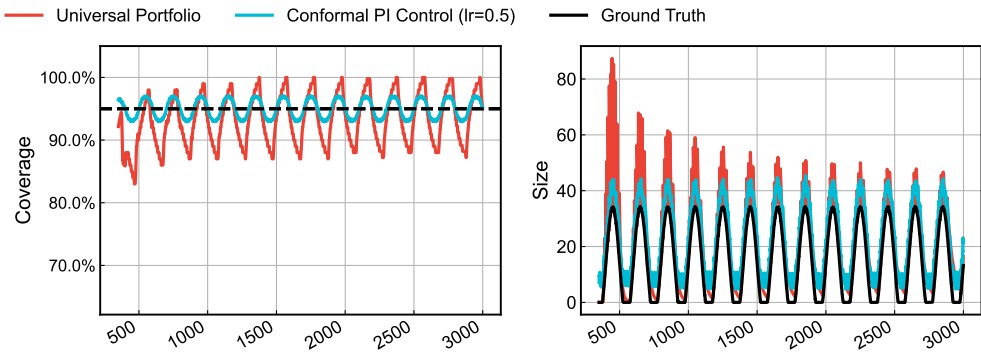

*Figure 54.* As in Figure 51, UP-OCP versus PI Ctrl (lr=0.5).

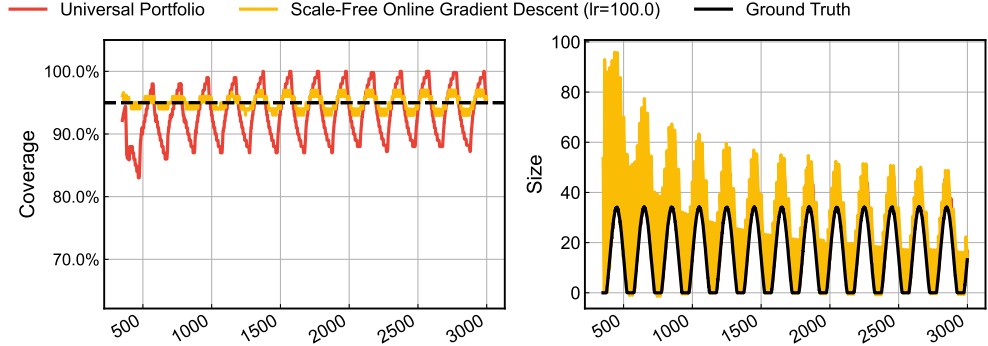

*Figure 55.* As in Figure 51, UP-OCP versus SFOGD (lr=100).

*Table 8.* Quantitative comparison on the sinusoid dataset (synthetic).

|  | UP | KT | DtACI | SFOGD (lr=100) | P Ctrl (lr=0.5) | PI Ctrl (lr=0.5) |
|---|---|---|---|---|---|---|
| Marginal coverage | 0.931 | 0.928 | 0.942 | 0.95 | 0.95 | 0.95 |
| Average set size | **21.1** | 26.9 | $\infty$ | 28.6 | 22.8 | 22.8 |
| Median set size | **17.9** | 27.6 | 26.4 | 27.7 | 21.9 | 21.5 |
| 75% quantile set size | 34.1 | 32.1 | 33.2 | 40 | 34.9 | 34.9 |
| 90% quantile set size | 43.2 | 34.5 | $\infty$ | 50.4 | 41.8 | 41.9 |
| 95% quantile set size | 48.3 | 35.3 | $\infty$ | 57.8 | 45 | 45.1 |

**More Pareto Frontiers and Target-level Tracking.**

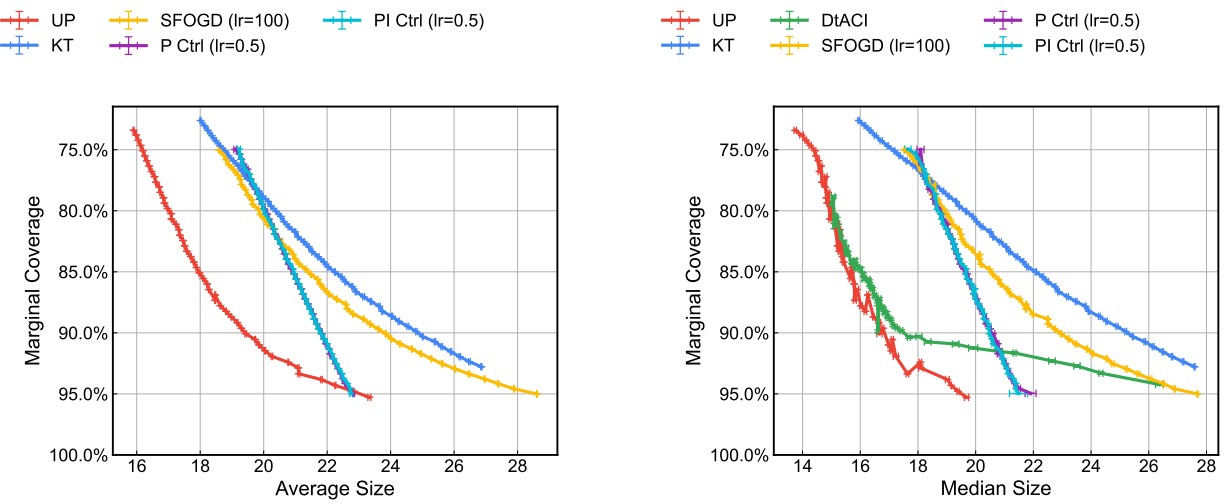

*Figure 56.* Mean prediction set sizes on synthetic sinusoid data.

*Figure 57.* Median prediction set sizes on synthetic sinusoid data.

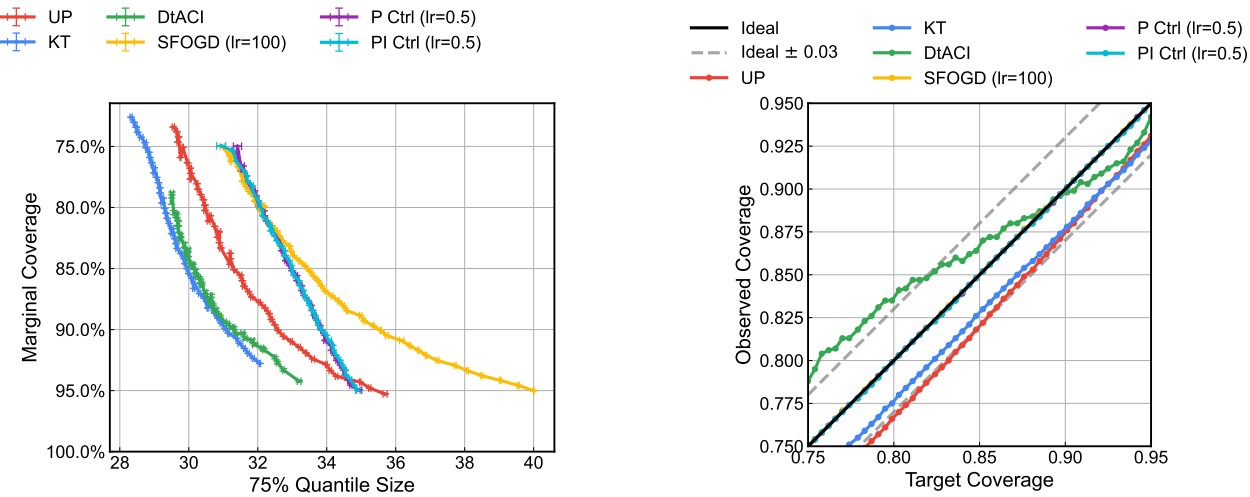

*Figure 58.* 75% quantile prediction set sizes on synthetic sinusoid data.

*Figure 59.* Realized versus target coverage on synthetic sinusoid data. Most methods track the diagonal within a small tolerance ($\pm$ 0.03).

## N.6. Stationary Trend with Random Waves

We examine robustness against sparse, heavy-tailed random waves, in comparison with the fixed periodic wave positions in the sinusoid setting. The nonconformity scores $S_t$ are generated via a process involving a constant baseline, sparse exponential noise, and a rolling window to create wave structures.

Formally, for $t = 1, \ldots, T$, we define a constant baseline $C = 10$. We sample a sparsity mask $B_t$ and a noise magnitude $E_t$ as:

$$B_t \sim \text{Bernoulli}(p),$$
$$E_t \sim \text{Exponential}(1/\sigma),$$

where $p = 0.1$ represents the spike probability and $\sigma = 10$ is the scale parameter. We first compute an intermediate score $\tilde{S}_t$ by applying multiplicative noise only when the mask is active:

$$\tilde{S}_t = C(1 + B_t E_t).$$

To simulate locally correlated volatility rather than isolated point outliers, the final score $S_t$ is obtained by applying a centered rolling max-filter of window size $W = 25$:

$$S_t = \max_{\tau \in [t - \lfloor W/2 \rfloor, t + \lfloor W/2 \rfloor]} \tilde{S}_\tau.$$

The total sequence length is $T = 3000$.

**Local Adaptivity.**

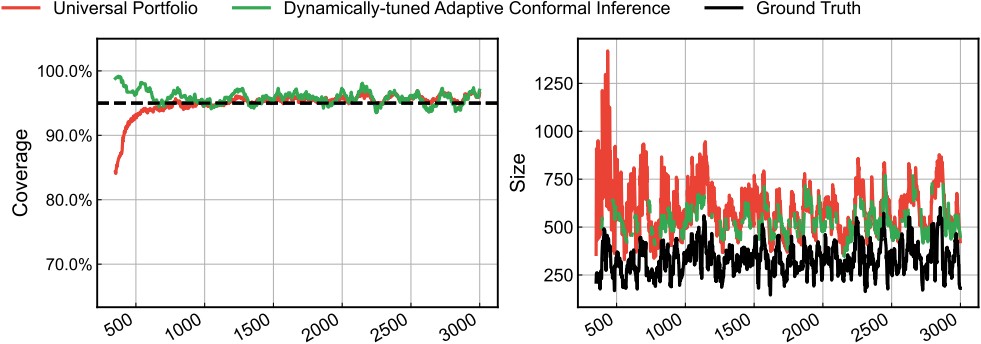

*Figure 60.* UP-OCP versus DtACI for forecasting stationary synthetic data with random waves.

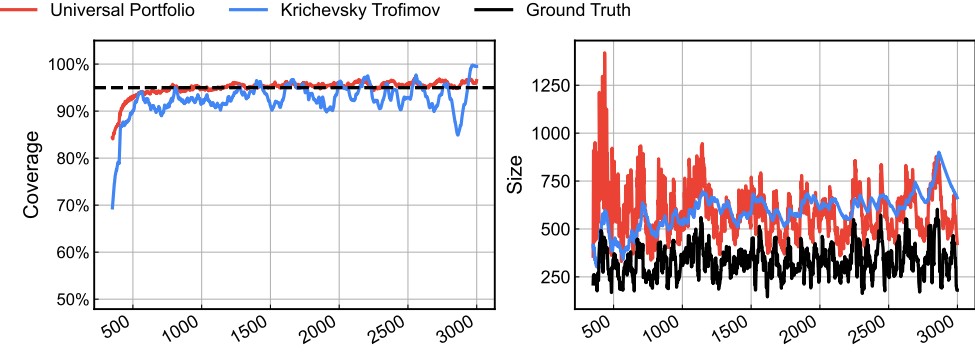

*Figure 61.* As in Figure 60, UP-OCP versus KT.

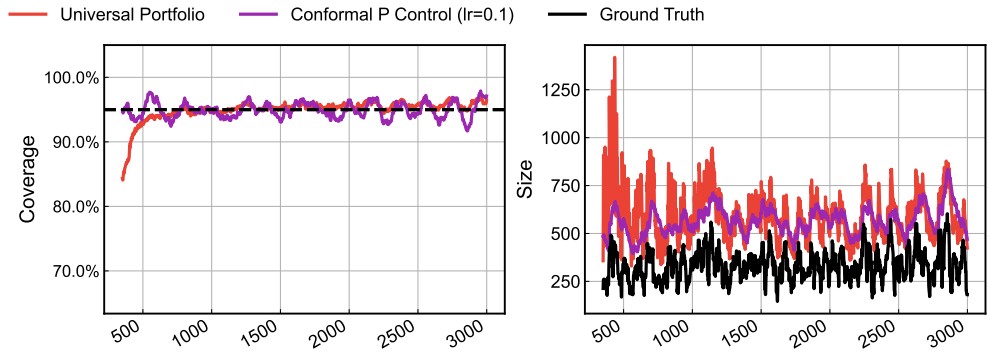

*Figure 62.* As in Figure 60, UP-OCP versus P Ctrl (lr=0.1).

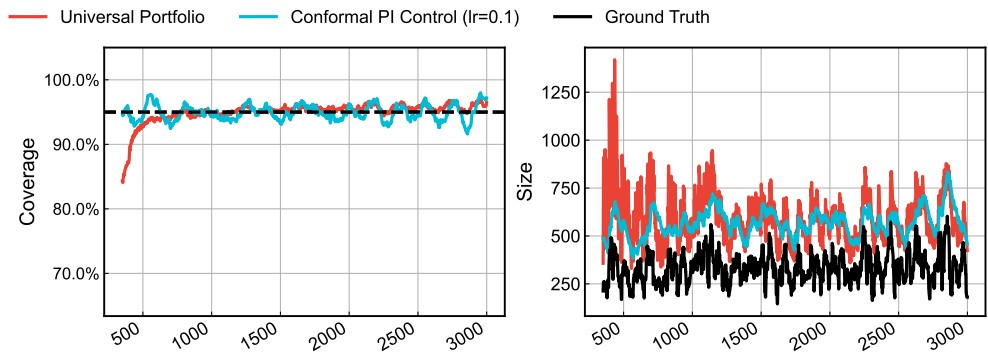

*Figure 63.* As in Figure 60, UP-OCP versus PI Ctrl (lr=0.1).

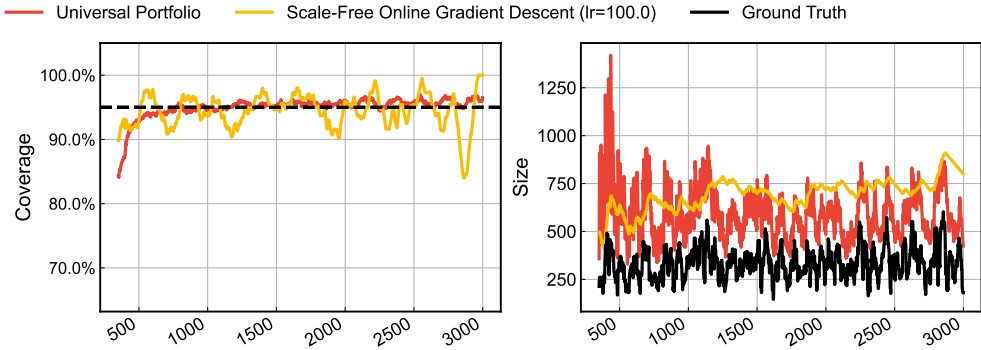

*Figure 64.* As in Figure 60, UP-OCP versus SFOGD (lr=100).

*Table 9.* Quantitative comparison on the stationary dataset (synthetic).

|  | UP | KT | DtACI | SFOGD (lr=100) | P Ctrl (lr=0.1) | PI Ctrl (lr=0.1) |
|---|---|---|---|---|---|---|
| Marginal coverage | **0.952** | 0.93 | 0.958 | 0.946 | 0.949 | 0.949 |
| Average set size | 592 | 596 | $\infty$ | 694 | 566 | 567 |
| Median set size | **500** | 574 | 494 | 691 | 531 | 534 |
| 75% quantile set size | 746 | 724 | 665 | 841 | 692 | 694 |
| 90% quantile set size | 1070 | 892 | $\infty$ | 962 | 882 | 880 |
| 95% quantile set size | 1320 | 994 | $\infty$ | 1050 | 1000 | 1010 |

**More Pareto Frontiers and Target-level Tracking.**

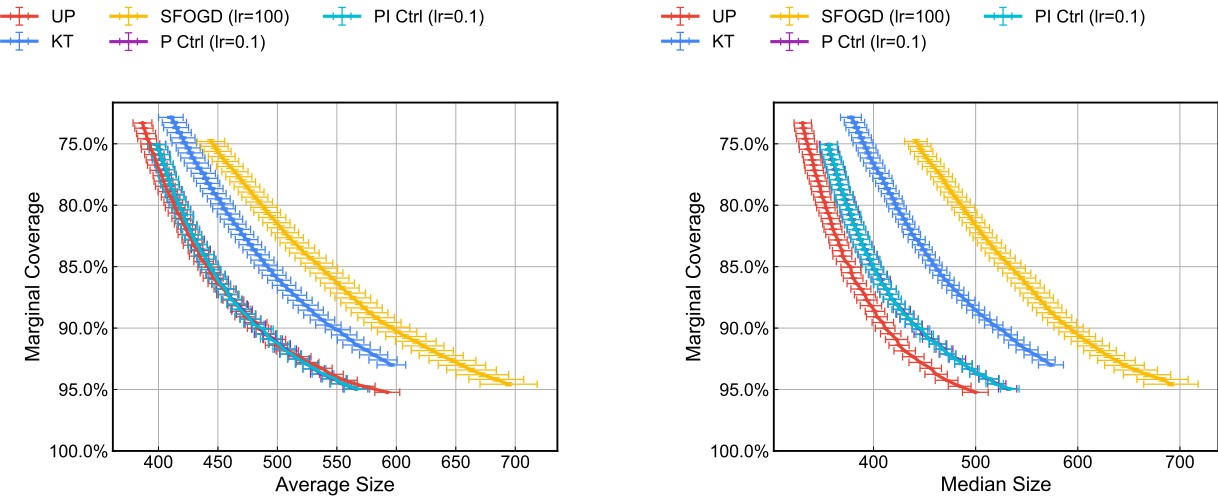

*Figure 65.* Mean prediction set sizes on synthetic stationary data with random waves.

*Figure 66.* Median prediction set sizes on synthetic stationary data with random waves.

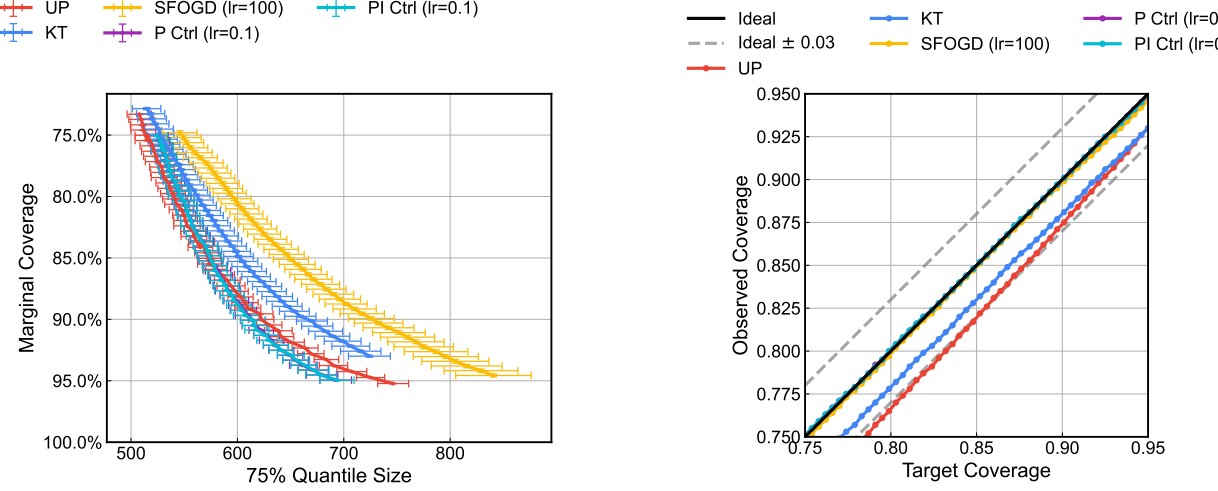

*Figure 67.* 75% quantile prediction set sizes on synthetic stationary data with random waves.

*Figure 68.* Realized versus target coverage on synthetic stationary data with random waves. Most methods track the diagonal within a small tolerance ($\pm$ 0.03).

## N.7. Quadratic Trend with Random Waves

Finally, we evaluate the algorithms on quadratic drift, which combines the sparse noise structure of the stationary regime with a non-stationary, monotonically increasing trend. This tests the ability of the algorithms to track a drifting baseline while remaining robust to random waves.

Formally, the underlying trend $T_t$ follows a quadratic trajectory starting at $0$ and ending at $20$ over $T = 3000$ steps:

$$T_t = \frac{20}{T^2} t^2. \tag{47}$$

The noise generation follows the same multiplicative, sparse structure as the stationary wavelet experiment. We sample a Bernoulli mask $B_t \sim \text{Bernoulli}(0.1)$ and exponential noise $E_t \sim \text{Exponential}(1/10)$. The raw score $\tilde{S}_t$ applies this noise to the drifting baseline:

$$\tilde{S}_t = T_t(1 + B_t E_t). \tag{48}$$

Finally, the observed score $S_t$ is the result of a rolling max-pooling operation with window size $W = 25$, creating locally correlated volatility structures around the drift.

**Local Adaptivity.**

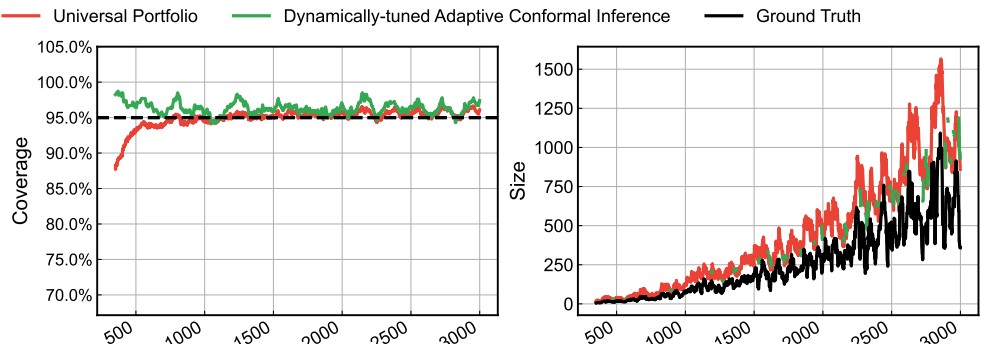

*Figure 69.* UP-OCP versus DtACI for forecasting synthetic data with quadratic trend and random waves.

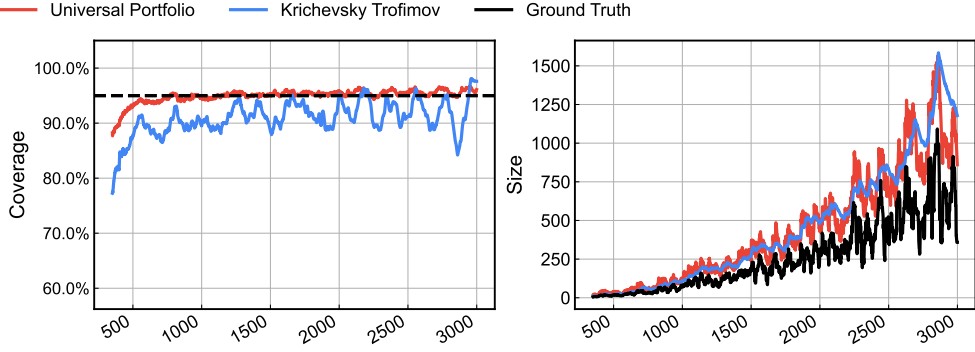

*Figure 70.* As in Figure 69, UP-OCP versus KT.

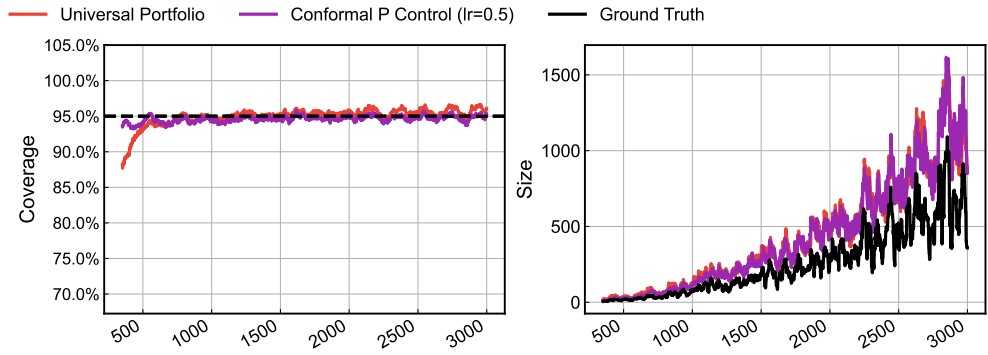

*Figure 71.* As in Figure 69, UP-OCP versus P Ctrl (lr=0.5).

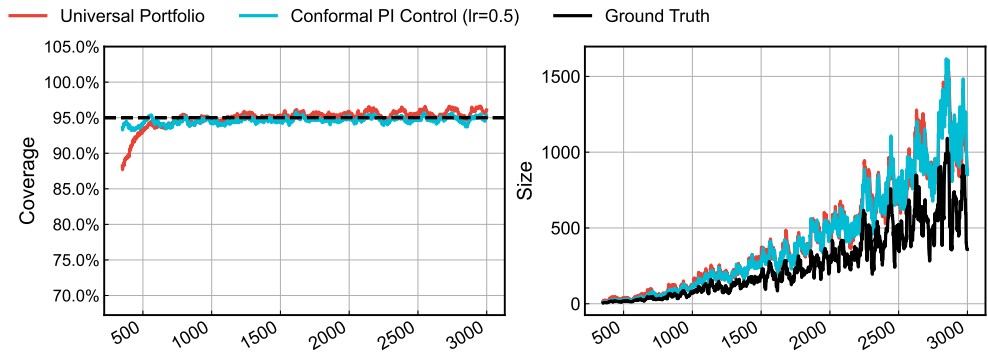

*Figure 72.* As in Figure 69, UP-OCP versus PI Ctrl (lr=0.5).

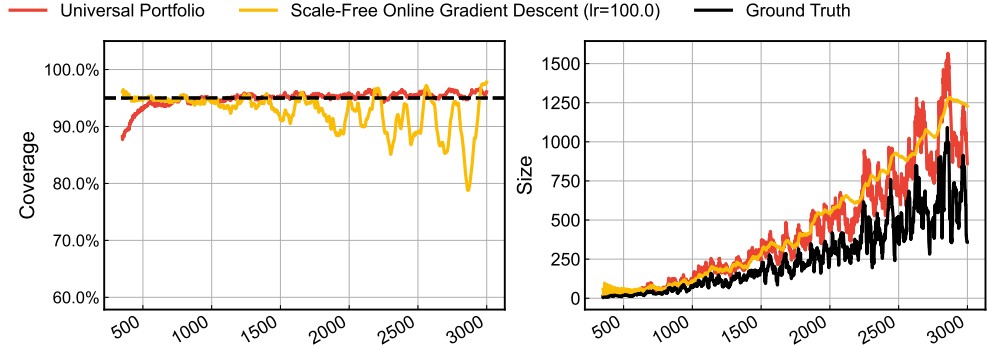

*Figure 73.* As in Figure 69, UP-OCP versus SFOGD (lr=100).

*Table 10.* Quantitative comparison on the quadratic-trend dataset (synthetic).

|  | UP | KT | DtACI | SFOGD (lr=100) | P Ctrl (lr=0.5) | PI Ctrl (lr=0.5) |
|---|---|---|---|---|---|---|
| Marginal coverage | **0.951** | 0.913 | 0.963 | 0.927 | 0.947 | 0.947 |
| Average set size | **433** | 451 | $\infty$ | 467 | **429** | **429** |
| Median set size | **292** | 308 | 406 | 324 | **285** | **285** |
| 75% quantile set size | **624** | 692 | $\infty$ | 790 | **626** | **626** |
| 90% quantile set size | **1020** | 1040 | $\infty$ | 1090 | **1020** | **1020** |
| 95% quantile set size | 1320 | 1360 | $\infty$ | 1260 | 1310 | 1310 |

## More Pareto Frontiers and Target-level Tracking.

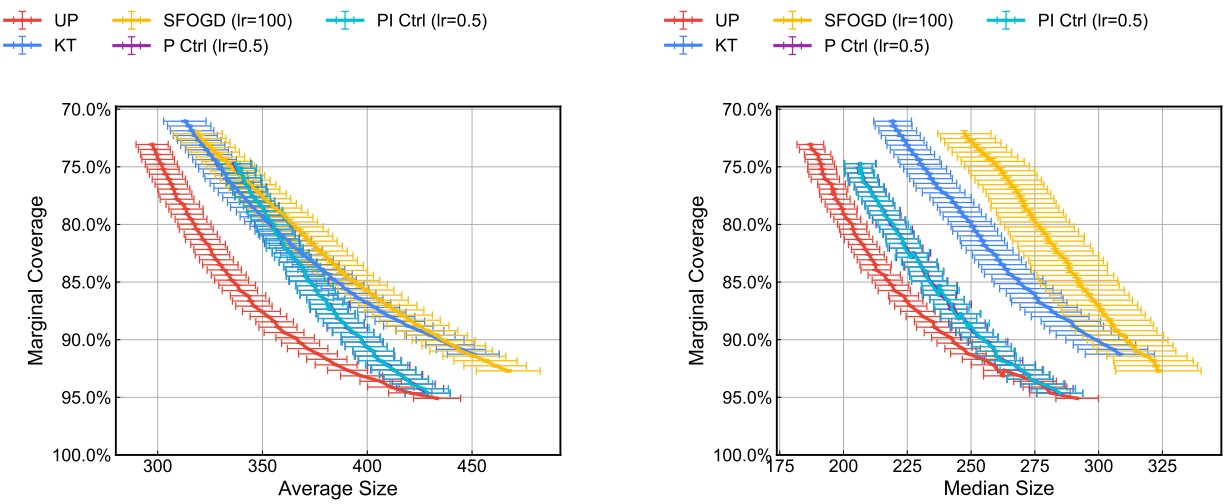

*Figure 74.* Mean prediction set sizes on synthetic data with quadratic trend and random waves.

*Figure 75.* Median prediction set sizes on synthetic data with quadratic trend and random waves.

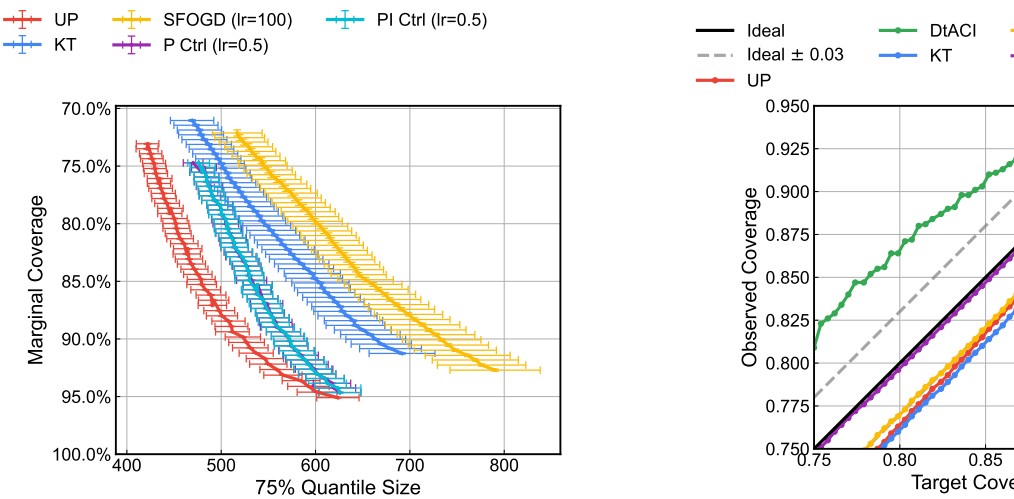

*Figure 76.* 75% quantile prediction set sizes on synthetic data with quadratic trend and random waves.

*Figure 77.* Realized versus target coverage on synthetic data with quadratic trend and random waves. Most methods track the diagonal within a small tolerance ($\pm$ 0.03).

## O. Sensitivity of Parameterized Baselines to Hyperparameters

In this section, we empirically demonstrate the sensitivity of parameterized OCP methods (SF-OGD, P/PI Control) to hyperparameter choices. While tuned baselines can achieve competitive performance (as shown in Section 5), selecting these parameters requires an oracle or grid search that is not feasible in a true online setting. We give three such examples of failure below. These examples underscore that parameterized methods cannot be naively plugged in; they require careful tuning. UP-OCP avoids these divergence modes by design without requiring manual tuning.

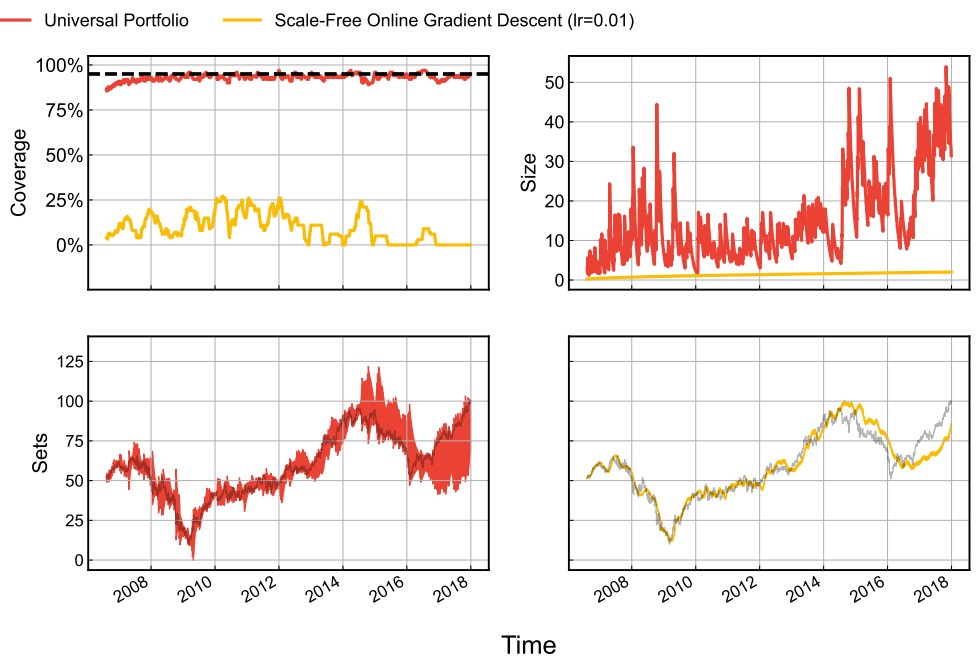

*Figure 78.* UP-OCP versus SF-OGD (lr=0.01) on the AXP dataset; SF-OGD (lr=0.01) fails to expand the prediction sets sufficiently, resulting in intervals that are consistently too narrow (right panel). The marginal coverage (yellow, left panel) collapses to nearly 0%, far below the 95% target.

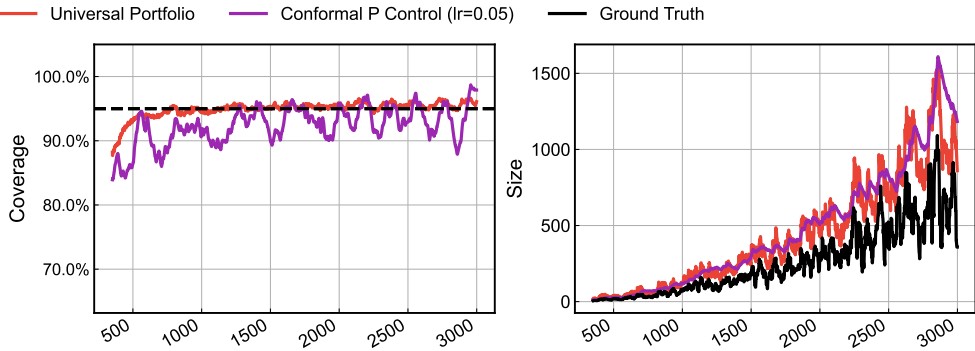

*Figure 79.* UP-OCP versus P Ctrl (lr=0.05) on synthetic data with quadratic trend and random waves; while the parameter-free UP-OCP (red) maintains stable coverage near the target, the P controller (purple) exhibits significant oscillation. Intuitively, this indicates that the controller is overreacting to single data points.

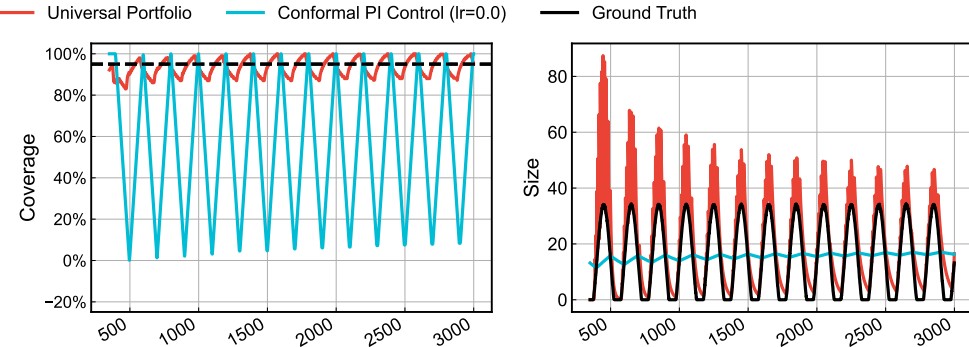

*Figure 80.* UP-OCP versus PI Ctrl (lr=0.0) on synthetic sinusoid data; the PI controller fails to adapt to periodicity in the ground truth (black). The prediction set sizes (cyan, right panel) remain effectively constant. The coverage (left panel) oscillates deterministically between 0% and 100% as the ground truth noise wave passes in and out. In contrast, UP-OCP (red) correctly modulates the interval width to track the sinusoidal pattern, maintaining valid coverage.

