# OpenReview forum: "Online Conformal Prediction via Universal Portfolio Algorithms"
_ICML.cc/2026/Conference — ICML 2026 spotlight_

### Official Review · Reviewer_UHNT · 2026-02-15

**Soundness:** 4
**Presentation:** 4
**Significance:** 3
**Originality:** 3
**Overall Recommendation:** 6
**Confidence:** 4

**Summary:**

The paper provides a new fundamental result on the connection between gradient equilibrium (online conformal prediction) and linear regret through the Fenchel conjugate of a bound on the linear regret.  This motivates the study of universal portfolio schemes for nline conformal prediction, yielding parameter-free protocols with improved theoretical guarantees on convergence as compared to the state of the art.

**Compliance With Llm Reviewing Policy:**

Affirmed.

**Final Justification:**

The paper provides a new fundamental result on the connection between gradient equilibrium (online conformal prediction) and linear regret through the Fenchel conjugate of a bound on the linear regret.  This motivates the study of universal portfolio schemes for nline conformal prediction, yielding parameter-free protocols with improved theoretical guarantees on convergence as compared to the state of the art.

**Key Questions For Authors:**

What are the practical limitations of the approach?

In what directions can the main result on the linear regret be extended to other notions of regret, such as the strongly adaptive regret?

**Limitations:**

Yes.

**Strengths And Weaknesses:**

Strengths

The paper is technically sound, and provides an important reference in the line of work on online conformal prediction.

Weaknesses

Please describe the relationship between the upper bound on the regret, the Fenchel dual and gradient in (7) via a figure. This is briefly discussed in words in the first paragaph of Section 4).

The experimental results are rather limited, and may not provide a full understanding of the limitations of the scheme.

Comment on possible extensions towards strongly adaptive regret following Bhatnagar et al 2023.

---

> ### Author Rebuttal · Authors · 2026-03-30
>
> We thank the Reviewer for the strong support and for recognizing the fundamental nature of our regret-to-coverage connection via Fenchel duality.
>
> ---
>
> ### Response to Key Questions:
>
> > On practical limitations:
>
> 1. **Marginal coverage only.** UP-OCP controls the *time-averaged* miscoverage rate. It does not provide *conditional* coverage (e.g., calibration conditional on threshold values and subgroups of the feature space). Combining UP-OCP with approaches from multi-validity [1] could be worth investigating in future work.
>
> 2. **Finite-time under-coverage.** As discussed in our response to Reviewer iEFj, the slight under-coverage is consistent with a fundamental coverage-vs-size trade-off. UP-OCP navigates this Pareto frontier by producing tighter sets at a small coverage cost. The $\alpha$-correction in Appendix F effectively closes this gap (see our response to Reviewer iEFj for details and an empirical demonstration). In future work, it would be valuable to provide theoretical support for an optimal calibration scheme.
>
> We will add a paragraph discussing these limitations and potential solutions in the revised version.
>
> > On extensions to strongly adaptive regret:
>
> Our framework naturally extends to sub-intervals. Since Theorem 3.1 relies on a pointwise bound $-g_t b_t \leq (1-\alpha) S_t$ (Lemma A.1), which holds for every round $t$ independently, the Fenchel conjugate argument can be applied to any sub-interval $[\tau, \tau+k-1] \subseteq [T]$: if an algorithm satisfies $\sum_{t=\tau}^{\tau+k-1} g_t(b_t - u) \leq F_k(u)$ for all $u$ and all intervals of length $k$, then $F_k^*(-\sum_{t=\tau}^{\tau+k-1} g_t) \leq (1-\alpha) \sum_{t=\tau}^{\tau+k-1} S_t$, yielding local coverage guarantees on every interval.
>
> The difficulty is at the meta-algorithm level. SAOCP [2] achieves strongly adaptive *standard* regret. While each expert (SF-OGD) controls linearized regret, the meta-algorithm aggregates experts via $\hat s_t = \sum_i p_{i,t} \hat{s}_{i,t}$ and decomposes meta-regret using convexity. This does *not* carry over to linearized regret, because $g_t = \nabla \ell_t(\hat{s}_t)$ is evaluated at the aggregate prediction, not at individual experts'. SAOCP [2] addresses this by switching to a randomized variant (sampling an expert $i \sim p_t$), making linearized regret decomposition work.
>
> A promising direction is to use UP-OCP instances as the experts within the SAOCP meta-algorithm [2]. Each expert would be a UP-OCP instance operating on its own active interval, inheriting our tighter $\alpha$-dependence in the regret bound. The theoretical challenge is to obtain a strongly adaptive *linearized* regret guarantee for the meta-algorithm that preserves the improved dependence on $\alpha$. The SAOCP coverage rate with SF-OGD experts is $\mathcal{O}(\alpha^{-2} \ln T / T^{1/4})$, substantially worse than UP-OCP's $\mathcal{O}(\sqrt{\alpha(1-\alpha) \ln T / T})$, so there is significant room for improvement by combining the two approaches. We will highlight this as a future direction.
>
> ---
>
> ### Response to Weaknesses
>
> > On the relationship between regret, Fenchel dual, and gradients in Theorem 3.1:
>
> In our response to Reviewer PTj8, we provide a formal derivation showing that a tighter regret bound $F_T$ shrinks the feasible set for the sum of subgradients, via the inverse ordering of Fenchel conjugates. We will include an illustrative figure in the revised version.
>
> > On experimental evaluation:
>
> While the main text only focuses on two representative datasets due to space constraints, the appendix contains a comprehensive experimental evaluation:
>
> - **8 datasets in total:** 4 stock price series (AXP, AAPL, AMZN, GOOGL), 1 electricity demand dataset (NSW), and 3 synthetic environments (Appendices K and L).
> - **50 target levels:** Pareto frontiers across $\alpha \in [0.05, 0.25]$, giving a picture of the coverage-size trade-off.
> - **Multiple metrics:** We report mean, median, 75%, 90%, and 95% quantiles of the prediction set size, as well as longest error sequence and marginal coverage.
> - **Sensitivity analysis (Appendix M):** We demonstrate that parameterized baselines (SF-OGD, P/PI Control) are highly sensitive to hyperparameter choice, and can fail catastrophically without careful tuning.
> - **Score growth demonstration (Appendix I):** We validate empirically that the polynomial growth model $S_t \leq D t^q$ is a more realistic representation of real-world score dynamics than the static boundedness assumption.
>
> If the Reviewer has additional specific datasets or experiments in mind, we would be very happy to run them and include the results in the revised version.
>
> ---
>
> [1] Bastani, O., Gupta, V., Jung, C., Noarov, G., Ramalingam, R., and Roth, A. Practical adversarial multivalid conformal prediction. NeurIPS, 2022.
>
> [2] Bhatnagar, A., Wang, H., Xiong, C., and Bai, Y. Improved online conformal prediction via strongly adaptive online learning. ICML, PMLR, 2023.

---

> > ### Author Rebuttal · Reviewer_UHNT · 2026-04-01
> >
> > My concerns have been adequately addressed, and I had already selected the maximum score.

---

### Official Review · Reviewer_U4YL · 2026-02-26

**Soundness:** 3
**Presentation:** 2
**Significance:** 3
**Originality:** 3
**Overall Recommendation:** 4
**Confidence:** 2

**Summary:**

This paper studies the problem of online conformal prediction (OCP), especially its relationship with other problems. The paper first shows the relation between miscoverage and linearized regret, showing that any online algorithm with guarantee of linearized regret could be applied to solve OCP problem. The paper further pushes this forward and shows that linearized regret minimization with pinball loss could be reduced to portfolio selection problem, which enjoys a benefit when $\alpha$ is close to 0 or 1. Experiments show that the proposed portfolio-algorithm-based OCP outperforms considered baselines.

**Compliance With Llm Reviewing Policy:**

Affirmed.

**Final Justification:**

This paper studies online conformal prediction and presents several reductions from other problems. The paper demonstrates that these reductions leads to improved guarantees for online conformal prediction, revealing the connection between different problems.

While the paper introduces several reduction, the sharpness of the reductions are not sufficiently discussed, therefore the connection between the discussed problems remains from fully characterized in this paper. Given the pros and cons listed above, I would like to maintain my evaluation as weak accept.

**Key Questions For Authors:**

My questions are listed below

1. How is the linearized regret differs from standard regret in terms of algorithmic results in literature? Can algorithm that minimizes standard regret automatically minimize linearized regret, or does this holds vise versa? If neither induces the other, what are the sota algorithm for both setting?

2. Can the results (derived from portfolio selection) in this paper induces better linearized regret in online learning where the loss function are not necessarily symmetry?

**Limitations:**

Yes

**Strengths And Weaknesses:**

The strengths and weaknesses of this paper is summarized as follows

Strengths:

1. The paper makes several interesting reduction, revealing the relationship between OCP, linearized regret minimization and wealth maximization in conformal market.

2. The reduction introduces new algorithms to OCP, which is superior in the scenarion of $\alpha$ close to 0 or 1.

Weaknesses:

1. While the paper introduces two reductions, it is not discussed whether such reduction is sharp. For example, as remarked in the paper, Lemma A.1 is loose when applying OSD to OCP. It is unknown whether other reduction (e.g., Theorem 4.3) incoporates such looseness

2. The reduction-incurred bound heavily depends on Frechel Duality, for example in Theorem 3.1 and Theorem 4.3, therefore it is not straightforward to see how dependencies on problem parameters (e.g., T, $\alpha$) translates. The paper could provide some more examples to demonstrate these scalings (e.g, for Theorem 3.1, provide some algorithms with their rate of regret and show how the regret translates to miscoverage).

3. The comparison to previous results are not clear enough. A better way might be summarizing the miscoverage rate (and other notable features) of OSD, KT and other relevant results in a table. Currently, the relevant rate is scattered over the paper and not convenient for comparison.

---

> ### Author Rebuttal · Authors · 2026-03-31
>
> We thank the Reviewer for all the constructive suggestions.
>
> ---
>
> ### Response to Key Questions:
>
> > On relationship between standard regret and linearized regret:
>
> We kindly refer the Reviewer to the first three paragraphs and Remark 3.2 of Section 3. We summarize the key points here:
>
> The standard regret is defined as $\text{Regret}_T(u) = \sum_t (\ell_t(b_t) - \ell_t(u))$, while the linearized regret is $\text{LinRegret}_T(u) = \sum_t g_t(b_t - u)$, where $g_t \in \partial \ell_t(b_t)$. By definition of subgradients, $\text{Regret}_T(u) \leq \text{LinRegret}_T(u)$ for all $u$.
> Thus, controlling linearized regret is *harder*: any linearized-regret bound automatically implies a standard-regret bound, but not vice versa.
>
> Remark 3.2 gives a counterexample for the reverse implication, where standard regret is $\mathcal{O}(\sqrt{T})$ but linearized regret grows as $\mathcal{O}(T)$. This shows sublinear standard regret does *not* imply coverage, whereas sublinear linearized regret does.
>
> In practice, most online learning algorithms use first-order oracles, so they inherently minimize linearized regret, with standard regret bounds as corollaries.
>
> > On extensions to asymmetric losses:
>
> We are a bit unsure how to interpret the terminology "symmetric loss functions." The pinball loss itself is already asymmetric (not an even function), and our algorithm applies. For now, we consider two interpretations below.
>
> If "symmetric" means subgradients with binary outcomes $g_t \in \{-(1-\alpha), \alpha\}$, then our portfolio reduction indeed extends to general losses with continuous subgradient range $g_t \in [a, b]$ ($a < 0 < b$). The conformal market generalizes via:
> $$
>     w_{t,1} = 1 - \frac{g_t}{b}, \quad w_{t,2} = 1 + \frac{g_t}{|a|}
> $$
> These returns remain non-negative. Although the explicit update rule in Section 4.2 no longer applies, Equation (10) is still valid. Applying Universal Portfolio to this generalized market induces tighter, variance-adaptive linearized regret bounds. This idea has been well-studied and used to generate tight confidence sequences in [1], with a regret bound given by Theorem 9.
>
> If "symmetric" refers to prediction sets, we refer to our response to Reviewer PTj8, where we show how to handle asymmetric intervals via two independent UP-OCP instances.
>
> We would be grateful if the Reviewer could help clarify so that we can make sure to answer their question to the highest level of precision.
>
> ---
>
> ### Response to Weaknesses
>
> > On sharpness of the two reductions:
>
> **Theorem 3.1 (Linearized regret $\Rightarrow$ Coverage):** This reduction has two ingredients: (i) Lemma A.1: $-g_t b_t \leq (1-\alpha)S_t$, and (ii) Fenchel duality. Lemma A.1 can be loose for specific algorithms like OSD. However, as we show in Appendix B (Lemma B.1), this looseness is *not inherent to the framework* but due to the generality of Lemma A.1. When algorithm-specific structure is available (e.g., the bounded iterates of OSD), a tighter bound on $-\sum_t g_t b_t$ recovers the sharp rate. As we have shown in Appendix A (proof of Theorem 3.1), the Fenchel duality step itself is tight: it is an equality when the supremum in the conjugate definition is attained. However, establishing lower bounds for the online miscoverage rate remains an open question.
>
> **Theorem 4.3 (Portfolio $\Rightarrow$ Linearized Regret):**  The reduction from the regret of a portfolio algorithm to the regret of an online linear optimization algorithm is tight, in the sense that one can prove that this is an *equivalence* rather than an implication; see Theorem 10.6 in [2].
> Some looseness can occur in the Fenchel conjugate inversion of the KL-based wealth lower bound (Appendix E.4), but the dominant terms are still correct.
>
> We will discuss the sharpness of both reductions in the revised version.
>
> > On examples and comparison to previous results:
>
> We will include a comparison table after Theorem 3.1 to show leading-order dependencies on $T$ and $\alpha$. Consider the following worked examples for Theorem 3.1, assuming $S_t \leq D t^q$:
>
> | Algorithm | Regret Bound | Miscoverage Rate | Parameter-Free? | Handles $S_t \to \infty$? | Improves as $\alpha \to 0$? |
> | :--- | :--- | :--- | :--- | :--- | :--- |
> | OSD | $\frac{u^2}{2\eta} + \frac{\eta T}{2}$ | $\mathcal{O}\left(\frac{\sqrt{D T^q}}{\sqrt{T}}\right)$ | No | No (rate degrades with $q$) | No |
> | KT | $\|u\|\sqrt{T \ln(24T^2 u^2 + 1)} + 1$ | $\mathcal{O}\left(\sqrt{\frac{\ln[(1-\alpha) D T^{q+3/2}]}{T}}\right)$ | Yes | Yes (log dependence on $q$) | No |
> | **UP-OCP** | **See Theorem 4.3.** | $\mathcal{O}\left(\sqrt{\frac{\alpha(1-\alpha)\ln[(1-\alpha) D T^{q+1}]}{T}}+ \frac{\ln[(1-\alpha) D T^{q+1}]}{T}\right)$ | **Yes** | **Yes** | **Yes** ($\to \frac{\ln T}{T}$) |
>
> ---
>
> [1] Orabona, F. and Jun, K.-S. Tight concentrations and confidence sequences from the regret of universal portfolio. IEEE Trans. Inf. Theory, 2023.
>
> [2] Orabona, F. A modern introduction to online learning. arXiv preprint, 2019.

---

> > ### Author Rebuttal · Reviewer_U4YL · 2026-03-31
> >
> > I appreciate the responses from the authors. While most of my concerns are addressed, I have some remaining questions as follows
> >
> > 1. On extensions to asymmetric losses: I am refering to your reference of symmetry in line 199, right column. My question is basically about: If the loss of some linear regret minimization problem is asymmetric, can you derive some algorithm that admits a better regret guarantee?
> >
> > 2. On extensions to asymmetric losses: Thank you for the answer! I recommend the author to add this discuss in the revision.
> >
> > 3. On examples and comparison to previous results: Thank you for the answer! Please also consider incorporate the result overview in the revision

---

> > > ### Author Response · Authors · 2026-04-01
> > >
> > > > On extensions to asymmetric losses:
> > >
> > > Thank you for the clarification. The asymmetry we exploit in our work is very specific to the structure of the pinball (quantile) loss used in online conformal prediction. The piecewise linear structure of the pinball loss produces binary subgradients $g_t \in \\{-(1-\alpha), \alpha\\}$. When $\alpha$ is small (the practically relevant regime, e.g., $\alpha = 0.05$), these outcomes are highly skewed. This particular structure is what allows UP-OCP via the universal portfolio reduction to achieve a variance-adaptive regret bound, compared to generic/symmetric algorithms like OSD and KT.
> > >
> > > For general regret minimization problems, however, asymmetry in the loss function is typically not the critical property for achieving sublinear regret (oftentimes not properly defined). What matters more are the regularity properties of the functional class, such as Lipschitz continuity, smoothness, or strong convexity. For these standard settings, minimax-optimal regret bounds are very well-studied (see, e.g., [1]), and the algorithms achieving them (e.g., Krichevsky-Trofimov bettor for Lipschitz functions) are well-known and do not specifically leverage asymmetry. In other words, asymmetry alone does not give faster rates beyond the established minimax rate for the given function class.
> > >
> > > > On adding the asymmetric loss discussion to the revision:
> > >
> > > Thank you for the suggestion. We will incorporate the discussion on asymmetric extensions (from our earlier response) into Section 4 of the revised version.
> > >
> > > > On incorporating the result overview in the revision:
> > >
> > > Absolutely. We will include the comparison table in Section 3, immediately after Theorem 3.1, so that readers can see the dependencies on $T$ and $\alpha$ at a glance.
> > >
> > > ---
> > >
> > > [1] Orabona, F. A modern introduction to online learning. arXiv preprint, 2019.

---

### Official Review · Reviewer_PTj8 · 2026-03-12

**Soundness:** 4
**Presentation:** 3
**Significance:** 3
**Originality:** 3
**Overall Recommendation:** 5
**Confidence:** 5

**Summary:**

In this paper, authors showed that online coverage in conformal prediction can be achieved by any online algorithm with small linearized regret. Then authors proposed a new algorithm that minimizes the linearized regret after reducing the problem to a portfolio selection problem. This algorithm particularly handled the asymmetry inherent in this problem. Lastly authors empirically verified that their algorithm is better at balancing coverage and size of the prediction set than other baselines.

**Compliance With Llm Reviewing Policy:**

Affirmed.

**Final Justification:**

My concerns are resolved and I would like to keep my score.

**Key Questions For Authors:**

How much would the current analysis framework change if the prediction sets are no longer symmetric intervals? Is it possible that some general notion of linearized regret could be relevant?

**Limitations:**

Yes

**Strengths And Weaknesses:**

**Strengths:**
1. Given prior results that no-regret alone doesn't imply online coverage, the observation that no-linearized-regret does imply online coverage is novel and insightful.
2. The reduction from linearized regret minimization to portfolio selection is novel and the observation that the problem is asymmetric is crucial for designing the right algorithm.
3. There is both theoretical and empirical evidence that the proposed algorithm dominates other baseline algorithms like OGD.

**Weaknesses:**
1. Minor issues: Does $\mu(\lambda)$ in equation (10) and later stand for the density of $d\mu$ wrt the Lebesgue measure?
2. From equation (5) it's clear that the mis-coverage rate is small if the sum of sub-gradients has small absolute value. It would be better if authors can explain how Theorem 3.1 implies that "small linearized regret implies small sum of sub-gradients".

---

> ### Author Rebuttal · Authors · 2026-03-30
>
> We thank the Reviewer for the careful reading and recognition of the novelty of our contributions, particularly, linearized regret as the right notion for coverage.
>
> ---
>
> ### Response to Key Questions:
>
> > How much would the current analysis framework change if the prediction sets are no longer symmetric intervals? Is it possible that some general notion of linearized regret could be relevant?
>
> This is a great question that points to important practical directions for future work.
> After some thought, we can observe that Theorem 3.1 by itself does not assume symmetric intervals. Its proof only uses:
>
> (i) the correspondence between pinball-loss subgradient $g_t$ and the coverage event $\boldsymbol{1}\\{b_t \geq S_t\\}$ and
> (ii) a bound on $-g_t b_t$ (Lemma A.1).
>
> It never references the shape of the prediction set.  If we can encode asymmetry into the nonconformity score, the online learning layer remains the same, and leads to the same results.
>
> For instance, we explain below how to extend our framework to learn independent adjustments for the left and right tails (by using signed residual or CQR score [1]).
> We track two independent thresholds $b_t^-$ and $b_t^+$, to generate prediction sets of form $C_t = [\hat{Y}_t^- - b_t^-, \hat{Y}_t^+ + b_t^+]$.
> Now, the miscoverage event can be decomposed into two mutually exclusive events:
> $$
>     P\left(Y_t \notin C_t\right)=P\left(Y_t<\hat{Y}_t^- - b_t^{-}\right)+P\left(Y_t>\hat{Y}_t^+ + b_t^{+}\right).
> $$
>
> This shows that we can split the target miscoverage $\alpha = \alpha^- + \alpha^+$ (usually taking both as $\alpha/2$), and track two separate signed scores: $S_t^- = \hat{Y}_t^- - Y_t$ and $S_t^+ = Y_t - \hat{Y}_t^+$. Assuming each score is bounded (non-negativity is not an issue),  Theorem 3.1 (and hence UP-OCP) apply independently to each component. This immediately gives separate control of upper- and lower-tail miscoverage.
>
> Thus, we run two *independent* instances of an online algorithm (with sublinear regret) in parallel:
> - One instance minimizes the linearized regret on $S_t^-$ to update $b_t^-$, bounded by $F_T(u^-)$.
> - Another minimizes the linearized regret on $S_t^+$ to update $b_t^+$, bounded by $F_T(u^+)$.
>
> We will include a discussion of this asymmetric extension in the revised version.
>
> While this approach provides a compelling solution, in future work, it would be of interest to study additional questions such as:
>
> (i) how to split the budget $\alpha = \alpha_1 + \alpha_2$, and
> (ii) whether a *coupled* multi-asset market can exploit cross-tail structure for tighter joint bounds.
>
> ---
>
> ### Response to Weaknesses
>
> > Minor issues: In equation (10) and later, does $\mu(\lambda)$ stand for the density of $\mu$ with respect to the Lebesgue measure $d\mu$?
>
> Yes. Here $\mu$ is the prior measure on $[0,1]$, and when we write $\mu(\lambda)$, we are referring to its density with respect to Lebesgue measure, i.e.,
> $$
>     d\mu(\lambda) = \frac{1}{\pi \sqrt{\lambda (1 - \lambda)}} \\, d\lambda.
> $$
>
> In the revised version, we will make this explicit.
>
> > From equation (5) it's clear that the mis-coverage rate is small if the sum of sub-gradients has small absolute value. It would be better if authors can explain how Theorem 3.1 implies that "small linearized regret implies small sum of sub-gradients".
>
> Thank you for pointing this out. We present a detailed mathematical derivation below.
>
> Let $F_T^A$ and $F_T^B$ be two valid linearized regret bounds such that $F_T^A(u) \leq F_T^B(u)$ for all $u \in \mathbb{R}$ and some $T \in \mathbb{N}^+$ (i.e., algorithm A has tighter regret). By definition, the corresponding Fenchel conjugates are ordered inversely:
> $$
>     (F_T^A)^\star(z) = \sup_u (zu - F_T^A(u)) \geq \sup_u (zu - F_T^B(u)) = (F_T^B)^\star(z) \quad \forall z \in \mathbb{R}.
> $$
>
> Now, let $G_T = -\sum_{t=1}^T g_t$ and let $D_T = (1-\alpha) \sum_{t=1}^T S_t$. By Theorem 3.1, any algorithm with a regret bound $F_T$ must satisfy $F_T^\star(G_T) \leq D_T$, which defines a feasible set for the sum of subgradients: $\mathcal{Z}(F_T) = \{z \in \mathbb{R} : F_T^\star(z) \leq D_T\}$. For any $z \in \mathcal{Z}(F_T^A)$, we have
> $$
> (F_T^B)^\star(z) \leq (F_T^A)^\star(z) \leq D_T \\
> \implies \mathcal{Z}(F_T^A) \subseteq \mathcal{Z}(F_T^B).
> $$
>
> This set inclusion implies that a tighter regret bound shrinks the maximum possible absolute sum of subgradients, i.e., $\left|\sum_{t=1}^T g_t\right|$, and equivalently, a smaller miscoverage rate.
>
> We will complement the intuitive description with the formal proof in the appendix.
>
> ---
>
> [1] Romano, Y., Patterson, E., and Candes, E. Conformalized quantile regression. NeurIPS, 2019.

---

> > ### Author Rebuttal · Reviewer_PTj8 · 2026-04-03
> >
> > I appreciate responses from authors and would like to keep my score.

---

### Official Review · Reviewer_iEFj · 2026-03-13

**Soundness:** 3
**Presentation:** 3
**Significance:** 3
**Originality:** 3
**Overall Recommendation:** 5
**Confidence:** 3

**Summary:**

This paper studies online conformal prediction (OCP). First, the authors show that the general notion of linearized regret directly implies coverage. Built on that, they propose UP-OCP, a new parameter-free method for OCP, and derives finite-time miscoverage bounds. Experiments show that UP-OCP delivers consistently better size/coverage trade-offs than prior baselines.

**Compliance With Llm Reviewing Policy:**

Affirmed.

**Final Justification:**

My questions have been resolved, and I have no further concerns. I will maintain my positive score.

**Key Questions For Authors:**

The empirical results show that UP-OCP achieves coverage slightly below the target level, at around 93%. What explains this gap, and is there a way to fix it?

**Limitations:**

yes

**Strengths And Weaknesses:**

Strengths
1. The linearized regret perspective is conceptually clean and potentially broadly useful. Theorem 3.1 gives a fairly general black-box route from online-learning guarantees to OCP coverage, rather than relying on algorithm-specific analyses.
2. The reduction to an asymmetric two-asset conformal market is interesting, and the resulting UP-OCP update is parameter-free and simple.
3. This paper presents extensive experiments on both synthetic data and real-world time series.

Weaknesses
1. Claims such as “optimal” and “best known finite-time coverage guarantee” need more clarification, since the paper mainly presents upper bounds.
2. The paper does not include proof sketches, which makes the theoretical results harder to understand.

---

> ### Author Rebuttal · Authors · 2026-03-31
>
> We thank the Reviewer for a clear summary of our work. Below we address each point raised.
>
> ---
>
> ### Response to Key Questions:
>
> > On under coverage in finite samples:
>
> **Why under-coverage arises.** There is a fundamental trade-off between marginal coverage and prediction set size in OCP. A trivial predictor achieves near-perfect coverage (Appendix H.5). Recent lower bounds [1] prove that any algorithm whose prediction sets have size at most $\mu$ times the optimal must incur $\mathcal{O}\left(\log(1/\varepsilon_{\min})/\log \mu \cdot\alpha T\right)$ coverage errors, where $\varepsilon_{\min}$ is the smallest relevant scale. So, the observed ~93% coverage for UP-OCP reflects its navigation of this Pareto frontier: it produces tighter prediction sets at the cost of slightly more miscoverage events than the target. Yet, it is guaranteed to reach the target coverage asymptotically.
>
> This gap is not specific to UP-OCP. Similar examples include KT, which achieves only ~92% in the AXP dataset, and DtACI, which overshoots at ~96% coverage in the quadratic dataset (while UP-OCP reaches almost the exact target of 95%).
> The tuned baselines (e.g., SF-OGD with oracle-selected learning rate) incur a smaller miscoverage error precisely *because* they benefit from ex-post choice, which is unavailable in a practical setting.
>
> We will add a remark in the revised version to clarify this fundamental coverage-size trade-off and its implications for finite-sample performance.
>
> **How to fix it.** We propose two approaches:
>
> 1. **$\alpha$-Correction (Appendix F).** We kindly refer the Reviewer to Appendix F, where we describe a principled heuristic: one runs UP-OCP with a slightly inflated target $\alpha' = \alpha + 1/\sqrt{T}$. Since the coverage converges at rate $\mathcal{O}(\ln T / T)$, this inflation is a lower-order perturbation that does not degrade the asymptotic convergence rate. Moreover, the Pareto frontier does not change — we are simply shifting along it. Empirically, this correction effectively closes the gap observed in finite samples. As an illustrative example, we show below the performance on the AXP dataset after applying this correction.
>
> | Metric | UP-OCP | KT | DtACI | SF-OGD |
> | :--- | :--- | :--- | :--- | :--- |
> | Marginal Coverage | **0.953** | 0.920 | 0.956 | 0.948 |
> | Longest Err. Seq. | **3** | 15 | 6 | **3** |
> | Avg. Set Size | **16.4** | 16.9 | $\infty$ | **16.4** |
> | Median Set Size | **13.0** | 12.9 | 12.6 | 13.8 |
> | 75% Quantile Size | **20.6** | 24.9 | 21.8 | 21.5 |
> | 90% Quantile Size | 34.3 | 32.5 | $\infty$ | 32.3 |
> | 95% Quantile Size | 41.7 | 36.1 | $\infty$ | 36.1 |
>
> 2. **Multi-Level Calibration.** Post-processing methods like the Multi-level Quantile Tracker [2] can enforce calibration across multiple quantile levels without crossings, via isotonic regression. Integrating such multi-level calibration techniques with UP-OCP is an interesting future direction.
>
> ---
>
> ### Response to Weaknesses
>
> > On claiming "best known finite-time coverage guarantee" and "optimal":
>
> We appreciate this observation and will revise the language in the revised version to be more precise.
>
> By "best known finite-time coverage guarantee," we mean that among the *upper bounds* established for existing OCP algorithms, UP-OCP achieves the tightest dependency on both $\alpha$ and $T$. Concretely:
> - **OSD**: $\mathcal{O}(T^{q-1})$.
> - **KT**: $\mathcal{O}(\sqrt{\ln [(1 - \alpha) T^{q + 3/2}]/ T})$, no improvement as $\alpha \to 0$.
> - **UP-OCP**: $\mathcal{O}(\sqrt{\alpha (1 - \alpha) \ln [(1 - \alpha) T^{q + 1}]/ T} + \ln [(1 - \alpha) T^{q + 1}]/ T)$ in general, reduced to $\mathcal{O}(\ln T^{q + 1}/ T)$ as $\alpha \to 0$.
>
> "Optimal" refers to the fact that the universal portfolio algorithm achieves the *minimax-optimal* regret for bounded Online Linear Optimization problems, as established by [3]. Through our reduction (Theorem 4.3), this regret translates to a miscoverage bound via Theorem 3.1, so any method based on our linearized-regret-to-coverage framework cannot improve beyond this rate. However, we do not claim a matching lower bound for the OCP problem *itself*. Establishing such lower bounds is an important open question that we will highlight more explicitly.
> Hence, we will make sure that "optimal" is carefully explained in the revised version.
>
> > On proof sketches:
>
> We apologize for this omission, which was due to space limitations in the initial submission. We will add proof sketches of:
>
> (i) Theorem 3.1 (Linearized regret $\Rightarrow$ Coverage) and (ii) Theorem 4.3 (Portfolio $\Rightarrow$ Linearized regret) in the revised version to improve readability.
>
> ---
>
> [1] Srinivas, V. Online conformal prediction with efficiency guarantees. SODA, SIAM, 2026.
>
> [2] Ding, T., Gibbs, I., and Tibshirani, R. J. Calibrated multi-level quantile forecasting. arXiv preprint, 2025.
>
> [3] Cover, T. M. and Ordentlich, E. Universal portfolios with side information. IEEE Transactions on Information Theory, 2002.

---

> > ### Author Rebuttal · Reviewer_iEFj · 2026-04-01
> >
> > Thank you to the authors for their responses. My questions have been resolved, and I have no further concerns. I will maintain my positive score.

---

### Decision · Program_Chairs · 2026-04-30

**Decision:**

Accept (spotlight)

**Comment:**

This paper introduces UP-OCP, a parameter-free algorithm for online conformal prediction, built on a novel connection to portfolio optimisation. Reviewers were broadly positive, praising the theoretical novelty, the parameter-free design, and the empirical evaluation.

Minor concerns were raised by reviewers: the algorithm slightly undershoots the target coverage in finite samples; the optimality claims are imprecise; proof sketches are missing; and prior work comparison is limited. All four reviewers confirmed their concerns were resolved.
The revision should incorporate the promised comparison table, proof sketches, and discussion of limitations and extensions.